# Circumpolar polynya regions and ice production in the Arctic: Results from MODIS thermal infrared imagery for 2002/2003 to 2014/2015 with a regional focus on the Laptev Sea

Andreas Preußer[1], Günther Heinemann[1], Sascha Willmes[1], and Stephan Paul[2]

[1]Department of Environmental Meteorology, Fac. of Regional and Environmental Sciences, University of Trier, Behringstr. 21, Trier, D-54296, Germany.
[2]Alfred Wegener Institute, Helmholtz Centre for Polar and Marine Research, Am Handelshafen 12, 27570 Bremerhaven, Germany.

*Correspondence to:* Andreas Preußer (preusser@uni-trier.de)

**Abstract.** High-resolution MODIS thermal infrared satellite data are used to infer spatial and temporal characteristics of 17 prominent coastal polynya regions over the entire Arctic basin. Thin-ice thickness distributions ($\leq 20\,\mathrm{cm}$) are calculated from MODIS ice-surface temperatures, combined with ECMWF ERA-Interim atmospheric reanalysis data in an energy balance model for 13 winter-seasons (2002/2003 to 2014/2015; November to March). From all available MODIS swath-data, (quasi-) daily thin-ice thickness composites are computed in order to derive quantities such as polynya area and total thermodynamic (i.e. potential) ice production. A gap-filling approach is applied to account for cloud and data gaps in the MODIS composites. All polynya regions combined cover an average thin-ice area of $226.6 \pm 36.1\ \mathrm{x10^3 km^2}$ in winter. This allows for an average total wintertime accumulated ice production of about $1811 \pm 293\ \mathrm{km^3}$, whereby the Kara Sea region and the North Water polynya (both 15%), polynyas at the western side of Novaya Zemlya (20%) as well as scattered smaller polynyas in the Canadian Arctic Archipelago (all combined 12%) are the main contributors. Other well-known sites of polynya formation (Laptev Sea, Chukchi Sea) show smaller contributions and range between 2 and 5%. We notice distinct differences to earlier studies on pan-Arctic polynya characteristics, originating to some part from the use of high-resolution MODIS data, as the capability to resolve small scale (> 2km) polynyas and also large leads is increased. Despite the short record of 13 winter-seasons, positive trends in ice production are detected for several regions of the eastern Arctic (most significant in the Laptev Sea region with an increase of $6.8\ \mathrm{km^3/yr}$) and the North Water polynya, while other polynyas in the western Arctic show a more pronounced variability with varying trends. We emphasize the role of the Laptev Sea polynyas as being a major influence on Transpolar Drift characteristics through a distinct relation between increasing ice production and ice area export. Overall, our study presents a spatially highly accurate characterization of circumpolar polynya dynamics and ice production which should be valuable for future modeling efforts on atmosphere- sea ice - ocean interactions in the Arctic.

## 1 Introduction

The sea ice cover in the Arctic is subject to continuous changes through a variety of thermodynamic and dynamic processes, which are driven by atmosphere and ocean dynamics. Areas of open water and thin ice, i.e. polynyas and leads, are characteristic

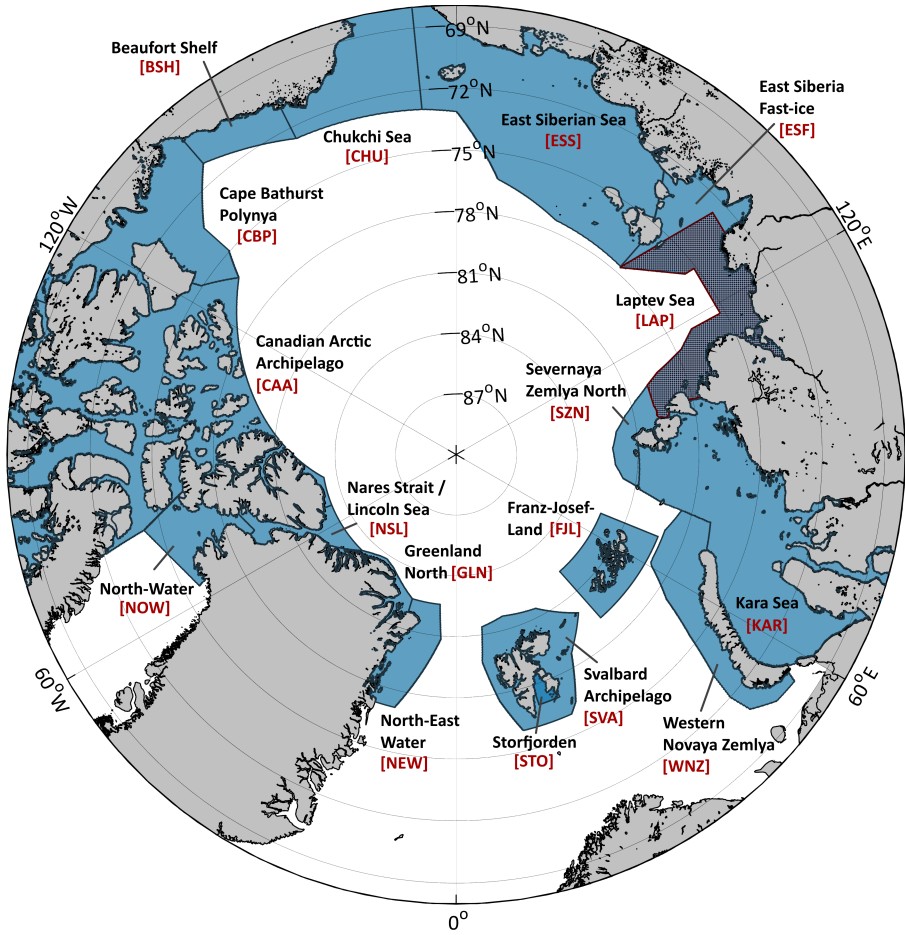

**Figure 1.** Map of all investigated areas of interest located in the Arctic, north of 68° N. Except for the Laptev Sea (red frame), all other applied polynya masks are marked in blue and enclose the typical location of each polynya in wintertime.

features in this ice scape with a huge influence on local physical, biological and chemical processes at the interface between the atmosphere and the ocean (Barber and Massom, 2007).

Especially during wintertime, the presence of open water and thin ice leads to increased ocean to atmosphere heat fluxes, thereby allowing for new ice production and brine release as well as generally strong modifications of both the atmospheric boundary layer and upper ocean layers (Ebner et al., 2011; Gutjahr et al., 2016). Hence, an accurate assessment of wintertime sea-ice production in the Arctic is of vital interest for the understanding of Arctic sea-ice dynamics, the annual sea ice mass balance and, in general, for the verification of climate and ocean models. In case of the Arctic, it is widely considered that the main mechanism for polynya and lead openings are divergent ice motions caused by wind-induced stress (Smith et al., 1990). Therefore, most Arctic polynyas can be found adjacent to or in proximity of a fixed obstacle such as the coastline, attached land-fast ice or ice bridges under offshore-wind conditions (Williams et al., 2007). While the time of formation, the duration

and the spatial extent of a polynya can be highly variable from year to year, their location of formation is generally rather stable (Morales-Maqueda et al., 2004). Leads are, in contrast, by far more variable both in space and time (Willmes and Heinemann, 2016). A regular monitoring of these open water and thin-ice areas with a high spatial accuracy is therefore a crucial step to be able to detect long-term changes, potential linkages and feedbacks to other environmental compartments as well as spatial and
temporal patterns.

Based on the inventory of Barber and Massom (2007), we here define a total of 17 individual polynya regions in the Arctic north of $68°$ N (Fig.1). Some of these areas are designed to match reference areas in previous studies (e.g. Kern, 2008). The areal extent, i.e. the total ocean area, of each sub-region is depicted in Tab. 1. The vast majority of polynyas of our study is located around the Arctic shelf areas, with the largest fraction in the Siberian shelf region (East Siberian Sea (ESS), Laptev
Sea (LAP), Severnaya Zemlya North (SZN), Kara Sea (KAR), Western Novaya Zemlya (WNZ)). Other well-known sites of polynya formation are the North Water (NOW) Polynya in northern Baffin Bay, several other frequently appearing thin-ice zones around northern Greenland (Nares Strait / Lincoln Sea (NSL), Greenland North (GLN), North-East Water (NEW) polynya), the Storfjorden (STO) polynya in the Svalbard Archipelago (SVA) and a number of smaller polynya locations around Franz-Josef Land (FJL), the Canadian Arctic Archipelago (CAA) as well as the Beaufort (BSH and CBP) and Chukchi (CHU)
Seas. The marginal ice zone (MIZ) in Fram Strait and northern Barents Sea is mostly excluded in our investigations due to a variety of potential ambiguities originating from ocean heat fluxes and a high interannual variability of the MIZ in terms of location and extent. However, in order to ensure consistency to previous studies, the MIZ is to some extent included in the CHU, STO, NOW, WNZ and KAR areas. For those regions, this implies that the here derived characteristics may contain periods with extensive ice-free conditions, first and foremost in early winter.

Pan-Arctic estimations of daily thin-ice thicknesses and ice production in polynyas were previously published by Tamura and Ohshima (2011) and Iwamoto et al. (2014), who both presented newly developed empirical thin-ice algorithms. Therein, commonly used passive microwave remote sensing data from the Special Sensor Microwave / Imager (SSM/I) and Advanced Microwave Scanning Radiometer - EOS (AMSR-E) satellite sensors is related to reference thin-ice thicknesses from Advanced Very High Resolution Radiometer (AVHRR) and Moderate Resolution Imaging Spectroradiometer (MODIS) thermal infrared
data, based on a characteristic inverse relationship between the surface brine volume fraction and the thickness of sea ice (Iwamoto et al., 2014). In both studies, the advantages of passive microwave systems (complete daily coverage in the Arctic, almost no influence of clouds) come at the cost of quite coarse spatial resolutions (6.25 - 25 km) which strongly limit the ability to resolve small and/or narrow thin-ice areas in close proximity to coastlines or along fast-ice edges (Preußer et al., 2015a).

According to Willmes et al. (2011), a retrieval of long-term ice production is challenging for several reasons. The derivation
of polynya area needs to be addressed with spatial and temporal resolutions that are sufficient to capture the seasonal and regional dynamics of polynya events (Winsor and Björk, 2000; Morales-Maqueda et al., 2004; Tamura et al., 2008; Willmes et al., 2010). Further, the heat loss over the polynya has to be calculated, which requires detailed information about the fraction of open water, the ice thickness and its distribution within the polynya. The distribution of thin-ice largely affects the heat loss by providing feedback on the ice surface temperature, thereby altering the vertical temperature gradients both through the ice
as well as towards the lower atmospheric boundary layer. Not less important, an accurate calculation of heat loss requires a

state-of-the-art approach regarding the parametrization of the surface energy balance, turbulent fluxes of latent and sensible heat and the conductive heat flux through the ice. Thus, detailed (i.e. region-specific and ideally highly resolved) information on meteorological quantities and correct formulations for the turbulent exchange coefficient for heat ($C_H$) are of particular importance (Gutjahr et al., 2016).

In order to address those challenges, the prime focus of this study is aimed towards the derivation of (quasi-) daily spatial thin-ice thickness distributions, which allows for a pan-Arctic retrieval of associated quantities like polynya area and thermo-dynamic ice production. We make use of a high-resolution and long-term record of thermal-infrared data from MODIS, as measured ice-surface temperatures can be combined with atmospheric reanalysis data in a 1-D energy balance model (Adams et al., 2013) to obtain ice thicknesses up to 50 cm (Sect. 3.1). Based on these daily distributions and taking into account a

necessary compensation for inherent cloud- and data gaps (Sect. 3.2), the amount of new sea-ice formation can be determined (Sect. 3.3). In Sect. 4, our achieved results will be presented and discussed, before closing this paper with final conclusions and prospects for further investigations. In recent studies using the same methodology based on MODIS data, we focused on the Storfjorden polynya (Preußer et al., 2015b) and NOW polynya (Preußer et al., 2015a). The present study has a strong focus on the Laptev Sea region due to significant changes over the last decade (Sect. 4.2) as well as its role as a central component of

the Transpolar Drift, a large-scale drift system where ice from the Siberian coastal regions is advected across the Arctic Ocean and through Fram Strait.

## 2   Data

### 2.1   MODIS ice-surface temperatures

The MOD/MYD29 Collection 5 sea-ice product (Hall et al., 2004; Riggs et al., 2006) is used to derive thin-ice thickness (TIT)

from MODIS satellite data. It features swath data of ice-surface temperatures (ISTs) from both MODIS instruments on board the Terra and Aqua polar-orbiting satellite platforms. All swath data offer a spatial resolution of 1x1 $\mathrm{km}^2$ at nadir and include a basic cloud-screening procedure using the MODIS cloud mask (MOD/MYD35; Ackerman et al. (2010)). In general, the accuracy of the MOD/MYD29 ISTs is given with 1–3 K (Hall et al., 2004). All IST swaths covering the Arctic Ocean and adjacent seas were extracted using meta data information for each MODIS swath. Subsequently, single swaths are mapped

onto a common equirectangular (reference-) grid covering all areas north of $68°$ N , with the output resolution set at 2 km in order to account for the decreasing spatial resolution of the MODIS sensor off-nadir. For our analysis between 2002/2003 and 2014/2015 (November to March), we used a total of 143,000 MODIS swaths for the complete Arctic domain, averaging at 73 scenes per day. The average amount of MODIS scenes per day and polynya region is additionally listed in Tab. 1.

### 2.2   ERA-Interim atmospheric reanalysis data

In order to provide the necessary atmospheric input for the applied surface energy balance model, the following variables from the European Center for Medium-Range Weather Forecasts (ECMWF) ERA-Interim reanalysis product (Dee et al.,

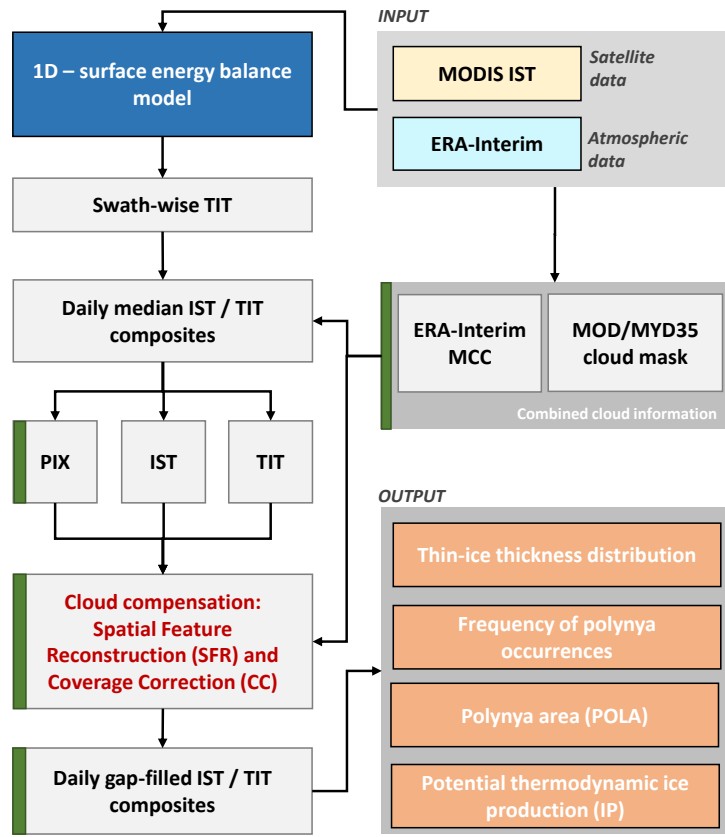

**Figure 2.** Schematic overview on the current version of the MODIS thin-ice thickness (TIT) retrieval scheme, based on Paul et al. (2015b) and Preußer et al. (2015a). The most recent updates are marked in green and are mainly aimed towards an additional cloud-cover treatment. Besides indicated abbreviations, 'IST', 'MCC' and 'PIX' denote the ice surface temperature, medium cloud cover and persistence index, respectively.

2011) are used: 2m-temperature, 2m-dew point temperature, 10m-wind speed and mean sea-level pressure. As the use of the MOD/MYD35 cloud mask during nighttime often contains misclassifications and ambiguities from undetected clouds and sea smoke, we additionally utilize ERA-Interim medium cloud cover (MCC) information. The study of Liu and Key (2014) demonstrated that the ERA-Interim MCC fields correspond closely to the MODIS derived cloud patterns throughout the seasons and can therefore be used as an additional quality control during the TIT retrieval. The temporal resolution of all variables is 6 h, so that each single MODIS swath can be linked to the closest time step of the atmospheric fields from ERA-Interim for the calculation of thin-ice thickness, with the overall average time difference being 90 ± 52 minutes (max. 180 minutes). The data set is provided by ECMWF at a spatial resolution of 0.75° (approx. 80 km). All ERA-Interim data fields are linearly interpolated and projected on the common reference grid in order to match the higher spatial resolution of MODIS data.

## 3 Methodology

### 3.1 MODIS thin-ice thickness retrieval using a surface energy balance model

We derive daily TIT distributions up to 50 cm by using an approach that follows the work of Yu and Rothrock (1996), Yu and Lindsay (2003) and Drucker et al. (2003). The core of this approach is an one-dimensional energy balance model, in which ice surface temperature (IST) and the thin-ice thickness are related to atmospheric radiation fluxes and turbulent heat fluxes. The original method of Yu and Rothrock (1996) was first improved and modified by Willmes et al. (2010) and Adams et al. (2013). More recently, the latest modifications of the algorithm are described in detail in Preußer et al. (2015b, a) and Paul et al. (2015b), together with comprehensive information on applied parametrization schemes that are used to calculate atmospheric radiation fluxes and turbulent heat fluxes. A complete overview on the currently used data-processing chain is given in Fig. 2.

There are certain limitations and simplifications attached to this procedure to derive TIT. First, it is only applicable to clear sky conditions, as clouds and sea smoke would strongly alter the recorded IST (Riggs et al., 2006). Second, we only use nighttime scenes to avoid potential ambiguities from incident short-wave radiation (Yu and Lindsay, 2003; Adams et al., 2013). Furthermore, newly formed ice is assumed to be free of snow and the temperature profile between the surface (IST) and the lower boundary of the ice (constant; freezing point of sea water) is linear. Consequently and following this assumption, the approach does not explicitly discriminate between different ice types within a polynya, as TIT are solely derived from calculating the heat conduction in/through an assumed layer of ice (aside from subsequent gap-filling; see Sect. 3.2).

The study of Adams et al. (2013) presented a sensitivity analysis of the TIT retrieval, which revealed average uncertainties of $\pm$ 1.0 cm, $\pm$ 2.1 cm and $\pm$ 5.3 cm for TIT classes 0–5 cm, 5–10 cm and 10–20 cm, respectively. Between 20–50 cm, the uncertainty increases considerably. Therefore, we constrain our analysis accordingly as a thickness range of TIT $\leq$ 0.2 m is widely regarded as a threshold for polynya areas and for estimates of thermodynamic ice production in polynyas (Yu and Rothrock, 1996; Adams et al., 2013; Haid et al., 2015).

### 3.2 Calculation of daily TIT composites and correction of cloud- and data gaps

Because of the restriction to nighttime scenes, a less frequent MODIS coverage is present especially towards the end (February to March) of each winter-season. In order to increase the MODIS coverage for all considered areas (Fig. 1), we derive daily composites of IST and TIT from the total number of available MODIS swaths covering the Arctic domain on a given day (compare Sect. 2.1). Following the procedure described in Sect. 3.1, the TIT is first calculated from each single swath on its own. Subsequently, the daily median TIT per pixel is calculated and stored alongside its corresponding IST value and daily median energy-balance components. The median is preferred over a simple average in order to reduce the potential risk of erroneously high or low values in single swaths, originating e.g. from unidentified clouds.

As described in Sect. 2.2, we additionally make use of ERA-Interim MCC fields as an indicator for potential cloud-coverage during the generation of daily TIT composites. Previous studies showed that a threshold of 75 % cloud-cover in the MCC-fields is quite effective in identifying and filtering/removing potentially cloud-affected areas (Paul et al., 2015b; Preußer et al., 2015a). The combined MODIS and ERA-Interim cloud information allows for the assignment of four different quality-classes for each

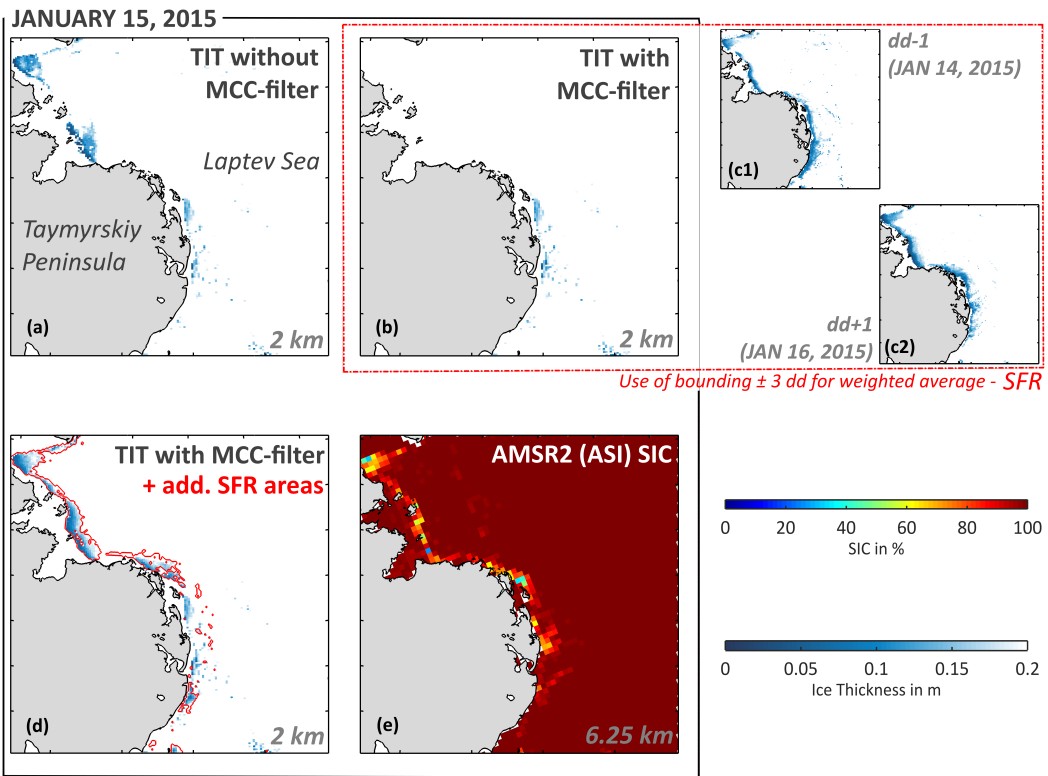

**Figure 3.** Different stages in the MODIS thin-ice thickness (TIT up to 0.2 m) processing chain for a single exemplary day (January 15, 2015). Sub-panels (a), (b), (c1/c2) and (d) all feature a subset (north-western Laptev Sea) from daily pan-Arctic TIT composites, with (a) showing the daily TIT without any cloud-treatment besides the MOD35 cloud mask and (b) the resulting TIT distribution after applying the ERA-Interim medium cloud cover (MCC) filter. Two bounding days with a better coverage of TIT are featured in panels (c1) and (c2) as a reference for the highest relative contribution in the spatial feature reconstruction (SFR) algorithm. The resulting spatial distribution of TIT after application of SFR is shown in panel (d), with new additional / reconstructed areas (up to 20 cm) marked in red. A comparison with Advanced Microwave Scanning Radiometer-2 (AMSR2) ASI sea-ice concentrations (Spreen et al. (2008); Beitsch et al. (2014); University of Bremen) from the same date is given in (e). The respective grid-resolution is given in the lower right corner of each sub-panel.

pixel in the daily composites: (1) confident clear-sky pixels ('ccs'; clear-sky MODIS and ERA-Interim), (2) mixed-covered pixels ('mcp'; ratio between clear-sky input swaths and the total number of input swaths per pixel), (3) definitive cloud-covered pixels ('dcc'; both in MODIS and ERA-Interim) and (4) completely uncovered pixels ('ucp').

Paul et al. (2015b) introduced an additional cloud-cover check based on the daily persistence of each pixel that is classified as thin ice (TIT $\leq$ 0.2 m). Misclassified thin-ice detections (i.e. clouds) are generally associated with low persistence-values due to their more mobile nature and displacements on sub-daily time scales. In contrast, polynyas show a higher spatial and temporal persistence due to their distinct formation mechanisms (Sect. 1). Leads, however, may be discarded by this criteria, since they generally have a low persistence due to their short lifetime and sea-ice drift caused by wind, ocean currents and tides. Based on these simple but distinct relations, we use a pixel-wise persistence index (PIX), defined as the ratio between the total number of MODIS swaths that feature thin-ice at a given pixel-location and the total number of swaths that feature clear-sky conditions at the same pixel-position.

All derived quality attributes (MCC-filter, cloud-cover information, PIX) are utilized in the Spatial Feature Reconstruction (SFR) algorithm (Paul et al., 2015a), which was recently successfully applied on a regional scale in both the Antarctic and Arctic to increase the information about otherwise cloud-covered areas (Paul et al., 2015b; Preußer et al., 2015a). The basic principle is that cloud-induced gaps in the daily TIT composites are compared with the TIT of the surrounding six days. In doing so, a probability of thin-ice occurrence is derived using a weighted composite of the days surrounding an initial day of interest (DOI). As in previous studies, we applied the following set of weights: $w_3 = 0.02$ (DOI $\pm$ 3), $w_2 = 0.16$ (DOI $\pm$ 2) and $w_1 = 0.32$ (DOI $\pm$ 1). The probability threshold remains fixed at $th$ = 0.34 and needs to be surpassed in order to assign 'new' probable polynya pixels. Paul et al. (2015a) showed that this combination is less restrictive in terms of missing MODIS coverage in close proximity of the initial DOI. The procedure is applied on all areas with identified low-quality data (low persistence, cloud-covered), so that indicated gaps can be filled with new information on potential thin-ice occurrences. For these areas, new TIT and IST values are pixel-wise allocated using a weighted average (same set of weights $w_1$ to $w_3$ applied) of the surrounding six days (Paul et al., 2015b; Preußer et al., 2015a). Table 1 gives an overview on the achieved MODIS coverage before and after application of the SFR algorithm. On a pan-Arctic level, the average (2002/2003 to 2014/2015) MODIS coverage is increased from around 0.75 (confident clear-sky and high-quality mixed-cover pixels featuring clear-sky conditions in more than 50% of all daily input swaths) to 0.93 (including SFR areas), with certain regions performing better (e.g. CBP, LAP, NEW, SZN) and some other regions noticeably worse (CHU, GLN, WNZ).

A total of 66 case studies in the Brunt Ice Shelf region of Antarctica demonstrated the generally good performance of the algorithm in comparison to more intelligible approaches by realistically reproducing artificially cloud covered thin-ice areas with an average spatial correlation of 0.83 and a RMSE of 1904 km$^2$ (Paul et al., 2015a). When compared to reference runs based on equally-weighted and in some cases shorter time intervals, the SFR procedure featuring above listed weights $w_1$ to $w_3$ (DOI $\pm$ 3 days) yielded superior results both in spatial correlation and reconstructed polynya extent, regardless of the temporal polynya evolution (e.g. opening/closing events). As an additional example from the Arctic (north-western Laptev Sea), Fig. 3 visualizes the basic principle of the SFR algorithm, together with a qualitative comparison of Advanced Microwave Scanning Radiometer 2 (AMSR2) sea-ice concentration (SIC) data (Spreen et al., 2008; Beitsch et al., 2014). As a first step, the MCC-

**Table 1.** Areal extents (i.e. total ocean area) of all applied polynya masks in $km^2$. Further, the interannual average amount of MODIS swaths that could be used for calculating daily composites in a given region is indicated, together with the interannual average daily MODIS coverage (decimal cover fraction ranging from 0 to 1 with their respective standard deviations) before (COV2) and after (COV4) application of the Spatial Feature Reconstruction (SFR) for each polynya region from 2002/2003 to 2014/2015 (November to March). The abbreviation 'ccs' denotes to confident clear-sky coverage, while 'HQ mcp' are high-quality mixed-cover pixels where either MODIS or ERA-Interim medium cloud cover feature cloud signals in the daily composites. In addition, the average thin-ice thickness (TIT, in cm) inside each polynya region (for all TIT $\leq 0.2$ m) is given together with its standard deviation. An overview on all applied predefined polynya masks is given in Fig. 1.

| Region | Total ocean area | Avg. number of MODIS swaths | COV2 | COV4 | Avg. TIT |
|---|---|---|---|---|---|
| | $(10^3 km^2)$ | $(d^{-1})$ | (ccs, HQ mcp) | (ccs, HQ mcp, SFR) | (cm) |
| Beaufort Shelf (BSH) | 91.6 | 6 | $0.76 \pm 0.03$ | $0.97 \pm 0.02$ | $14.0 \pm 0.5$ |
| Canadian Arctic Archipelago (CAA) | 719.6 | 14 | $0.82 \pm 0.03$ | $0.96 \pm 0.01$ | $13.7 \pm 0.2$ |
| Cape Bathurst (CBP) | 311.6 | 10 | $0.81 \pm 0.03$ | $0.98 \pm 0.01$ | $14.1 \pm 0.4$ |
| Chukchi Sea (CHU) | 286.0 | 5 | $0.55 \pm 0.04$ | $0.79 \pm 0.03$ | $12.8 \pm 0.4$ |
| East Siberian Fast-Ice (ESF) | 110.1 | 8 | $0.77 \pm 0.04$ | $0.96 \pm 0.01$ | $14.3 \pm 0.3$ |
| East Siberian Sea (ESS) | 904.1 | 9 | $0.70 \pm 0.03$ | $0.92 \pm 0.01$ | $14.0 \pm 0.3$ |
| Franz-Josef-Land (FJL) | 140.1 | 13 | $0.79 \pm 0.04$ | $0.97 \pm 0.02$ | $11.7 \pm 0.8$ |
| Greenland North (GLN) | 33.8 | 13 | $0.67 \pm 0.04$ | $0.81 \pm 0.04$ | $16.3 \pm 0.4$ |
| Kara Sea (KAR) | 725.5 | 14 | $0.75 \pm 0.04$ | $0.95 \pm 0.02$ | $11.7 \pm 1.1$ |
| Laptev Sea (LAP) | 281.1 | 12 | $0.80 \pm 0.03$ | $0.98 \pm 0.01$ | $13.5 \pm 0.5$ |
| North East Water (NEW) | 112.0 | 13 | $0.81 \pm 0.03$ | $0.98 \pm 0.01$ | $13.7 \pm 0.5$ |
| North Water (NOW) | 110.1 | 13 | $0.85 \pm 0.04$ | $0.97 \pm 0.01$ | $11.5 \pm 0.5$ |
| Nares Strait / Lincoln Sea (NSL) | 55.5 | 14 | $0.83 \pm 0.03$ | $0.96 \pm 0.01$ | $13.6 \pm 1.0$ |
| Storfjorden (STO) | 11.7 | 9 | $0.75 \pm 0.04$ | $0.95 \pm 0.03$ | $9.2 \pm 1.7$ |
| Svalbard Archipelago (SVA+STO) | 204.3 | 13 | $0.68 \pm 0.06$ | $0.90 \pm 0.04$ | $7.2 \pm 1.0$ |
| Severnaya Zemlya North (SZN) | 65.3 | 12 | $0.80 \pm 0.04$ | $0.98 \pm 0.01$ | $13.5 \pm 0.6$ |
| Western Novaya Zemlya (WNZ) | 211.6 | 12 | $0.60 \pm 0.07$ | $0.83 \pm 0.04$ | $6.8 \pm 1.6$ |
| Total | 436.2 | $73^a$ | $0.75 \pm 0.04$ | $0.93 \pm 0.02$ | $12.7 \pm 0.6$ |

a Not the sum of all regions, as single MODIS swaths may cover multiple regions at the same time.

filter eliminates potentially cloud-influenced areas which are in this case located north of the Taymyr Peninsula (Laptev Sea, Russia). One could argue that this filtering is a bit harsh, but we choose a more conservative threshold to minimize the risk of 'false' thin-ice pixels. Afterwards, the SFR algorithm is applied and a new gap-filled TIT composite (Fig. 3 (d)) is produced. In this particular example from January 15, 2015, the reconstructed TIT distribution compares well with locations of lower SIC

from AMSR2 (Fig. 3 (e)) while maintaining the increased spatial detail at the same time. Based on this example and successful applications in previous works by the authors (Paul et al., 2015a, b; Preußer et al., 2015a), we conclude that the applied schemes to compensate and correct cloud-effects work reasonably well on a pan-Arctic scale and allow for a fair comparison to other commonly used remote sensing approaches to infer polynya characteristics, with limitations regarding the reconstruction of leads.

### 3.3 Derivation of ice production and polynya area

Ice production rates are derived by assuming that the entire heat loss at the ice surface to the overlying atmosphere contributes to new ice formation (Tamura et al., 2007, 2008; Willmes et al., 2011). Components for the following equation (Eq. 1) can be taken from calculated and gap-filled daily MODIS composites.

$$\frac{\partial h}{\partial t} = \frac{-\bar{Q}_{ice}}{\rho_{ice} * L_f} \tag{1}$$

Therein, $\frac{\partial h}{\partial t}$ denotes to the ice production rate, $\bar{Q}_{ice}$ is the daily mean conductive heat flux through the ice, $\rho_{ice}$ is the density of the ice ($\rho_{ice} = 910 \text{ kg/m}^3$; Timco and Frederking (1996)) and $L_f$ is the latent heat of fusion of sea ice ($L_f = 0.334 \text{ MJ/kg}$; Tamura and Ohshima (2011)). Concerning $L_f$, Tamura and Ohshima (2011) noted that an accurate value for areas of high ice production is not known so far. Following the work of Martin (1981), Tamura and Ohshima (2011) argued that frazil ice consists of freshwater ice crystals enclosed with a thin saline layer and that frazil ice production rates are of similar magnitudes

as freshwater ice production rates. Consequently, we also use fixed values for $\rho_{ice}$ and $L_f$ in order to ensure comparability with earlier studies focusing on sea-ice production in (Arctic) polynyas (e.g. Willmes et al., 2011; Tamura and Ohshima, 2011; Iwamoto et al., 2014). However, this simplification may introduce an additional error source in our estimates due to spatially and temporally varying conditions for ice formation. Note that the negative sign in Eq. 1 implies that the atmospheric heat flux is positive when the surface gains energy, and at the same time it assures that ice production only takes place when there is a

net energy loss from the surface. According to the surface energy balance, the heat flux $\bar{Q}_{ice}$ is equal to the total atmospheric heat loss (sum of net radiation, turbulent latent and sensible heat flux). We do not consider an ocean heat flux, although it might potentially reduce thermodynamic ice growth in certain areas of the Chukchi Sea (Hirano et al., 2016), the Canadian Arctic Archipelago (Hannah et al., 2009; Melling et al., 2015) and northern Baffin Bay (e.g. Steffen, 1985; Yao and Tang, 2003) by as much as 23-27% in case of the NOW polynya (Tamura and Ohshima, 2011; Iwamoto et al., 2014). The volume ice production

rate $\frac{\partial V}{\partial t}$ (IP) is calculated by multiplying $\frac{\partial h}{\partial t}$ with the areal extent of each pixel in the regarded region. Ice production rates are calculated for each pixel with a TIT $\leq 0.2$ m and afterwards extrapolated to daily rates. However, it has to be noted that daily IP rates inhibit a positive bias due to the exclusive use of both nighttime and clear-sky MODIS scenes. The former is mainly

of concern during the late autumn / early spring period when polar night conditions are absent, while the latter circumstance is unavoidable throughout each winter when relying on optical and infrared remote sensing data. Since low-level clouds reduce the net radiative loss by about 50 $\mathrm{W/m^2}$ in polar regions (Heinemann and Rose, 1990; König-Langlo and Augstein, 1994), the restriction to cloud-free conditions in the daily composites results in a positive bias in IP. Considering an average clear-sky

fraction of $73 \pm 8$ % per pixel from all input swaths in a given daily composite and assuming that not all clouds are low-level, the overestimation of net energy loss by our method can be estimated to be less than 10 $\mathrm{W/m^2}$, which corresponds to less than 0.4 m IP per winter.

The daily polynya area (POLA, in $\mathrm{km^2}$) in each polynya mask (Fig. 1) is defined as the accumulated total area of all thin-ice pixels with a TIT $\leq 0.2$ m. Remaining MODIS coverage gaps after the application of the SFR approach (e.g. prolonged

periods of persistent cloud cover, i.e. no coverage on more than 3 consecutive days) are handled by additionally applying an extrapolation approach (coverage-correction; CC) on calculated POLA and IP estimates, which yields daily values with an error-margin of 5 to 6% (Preußer et al., 2015b). In case of very persistent cloud cover inside the respective reference areas and a resulting daily MODIS coverage below 50% (i.e. COV4 < 0.5), both daily POLA and IP are linearly interpolated from bounding days.

The complete period from November to March each winter is considered for the calculation of POLA / IP, which implies that the here derived values are potentially influenced by shifts in the timing of freeze onset during the early freezing season (November / December). For potentially MIZ-influenced regions (CHU, SVA, NOW, WNZ, KAR), this has to be considered when comparing metrics derived for the full winter period (November to March).

For topographically complex regions like Greenland and Arctic fjords, recent studies revealed shortcomings of the coarse-

resolution ERA-Interim data regarding the representation of mesoscale spatial features in the wind field, such as tip-jets, channeling effects or other topography-induced phenomena related to locally increased wind speeds (e.g. Moore et al., 2016). Thus, ERA-Interim shows a tendency to underestimate peak wind speeds (Moore et al., 2016) which might in some cases induce a negative bias (lower heat fluxes/ smaller POLA / less IP) in regions where polynya formation is strongly influenced by the local topography (e.g. CAA, NOW, NEW, SZN). In our study, the usage of ERA-Interim is motivated by ensuring

comparability to similar studies (e.g. Iwamoto et al., 2014) as well as the constraint that higher-resolution atmospheric data sets such as the Arctic System Reanalysis (ASRv1 – 30km; Bromwich et al., 2015) are currently not available for the complete time period from 2002 to 2015.

## 4   Results and Discussion

### 4.1   Thin-ice dynamics, polynya area and thermodynamic ice production in the Arctic for 2002/2003–2014/2015

Interannual average values for TIT $\leq 0.2$ m are listed in Tab. 1 for each polynya region. They range between 6.8 cm (WNZ) and 16.3 cm (GLN), with an overall average of about $12.7 \pm 0.6$ cm. The underlying long-term time series of average wintertime TIT within each polynya (not shown) reveal a tendency towards decreasing thin-ice thicknesses in almost every region (e.g. up to 2.5 cm per decade in the Storfjorden polynya), with the only exceptions being the CAA, GLN and NEW.

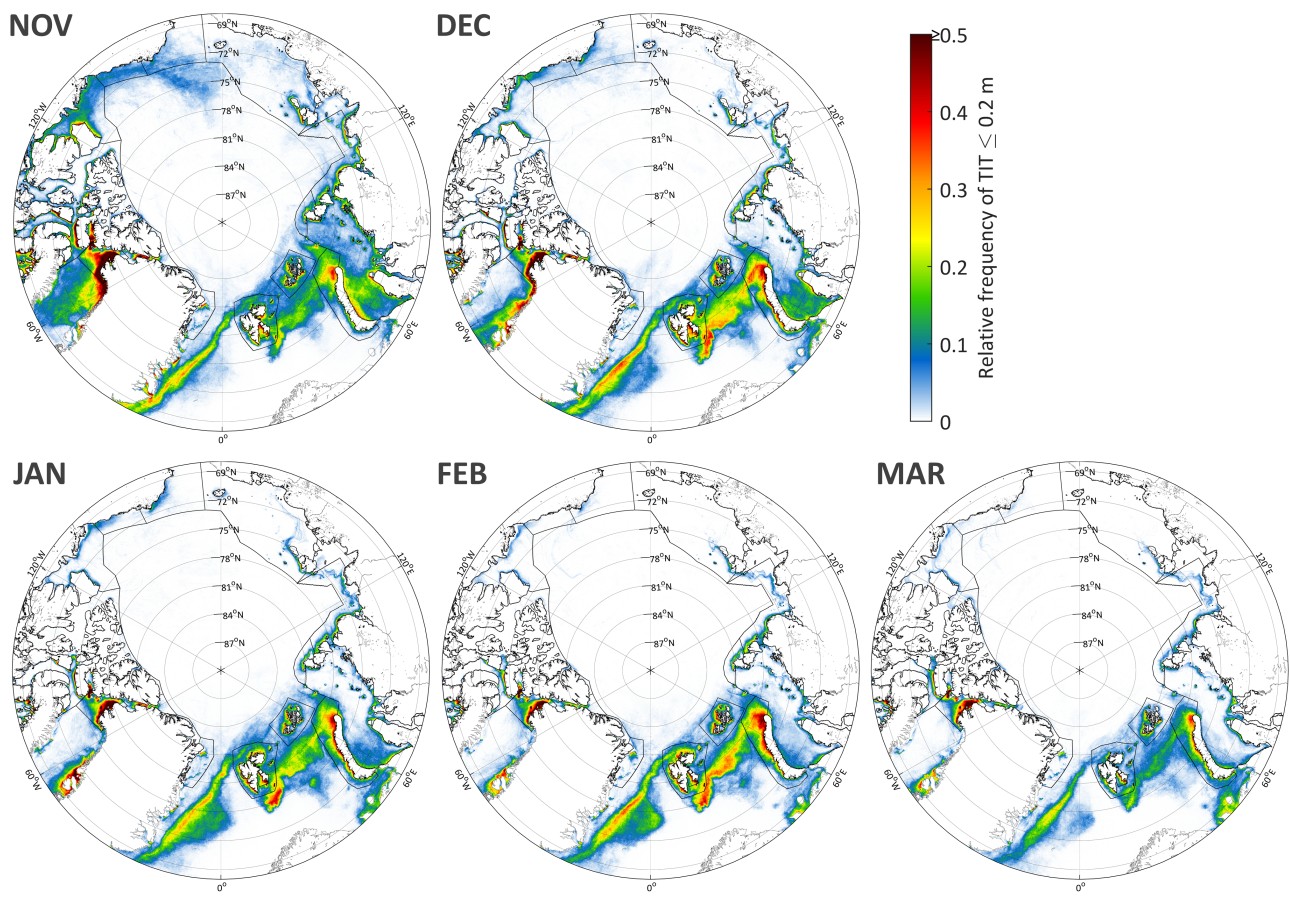

**Figure 4.** Average wintertime (November to March) frequencies of TIT $\leq 0.2$ m in the Arctic between winters 2002/2003 and 2014/2015. For each month, frequencies are calculated per pixel as the fraction of days with a TIT $\leq 0.2$ m relative to the 13-yr investigation period. Note that only thin-ice areas within the margins of a given polynya mask (dashed black lines; compare Fig. 1) are used for further analysis, while all other areas are discarded. Hence, areas with high TIT frequencies in the marginal ice zone (MIZ) around Fram Strait and northern Barents Sea are excluded from further analysis due to potential ambiguities originating from ocean heat fluxes and a high interannual variability of the MIZ in terms of location and extent.

Monthly thin-ice frequencies, calculated per pixel as the fraction of days with a TIT $\leq 0.2$ m relative to the 13-yr investigation period, are presented in Fig. 4. Frequencies of larger than 0.5 are primarily found around the Canadian Arctic, first and foremost in the North Water (NOW) polynya and the eastern CAA. More specifically, coastal areas around Devon Island and south-eastern Ellesmere Island (Hells Gate / Cardigan Strait) and larger areas at the eastern exits of Lancaster Sound and Jones

Sound are well visible and have previously been related to tidal currents and slightly increased ocean heat fluxes (Hannah et al., 2009; Melling et al., 2015). Other areas with similar magnitudes include the Storfjorden polynya and coastal areas (north-)west of Novaya Zemlya. Besides, elongated thin-ice areas along the Siberian shelf (Laptev and Kara Sea; frequencies around 0.05 to 0.35 each month) are well delineated. Locations of frequent thin-ice occurrences in the Kara Sea are in accordance with results from the study of Kern (2008). The northern Barents Sea, Franz-Josef-Land and the Svalbard archipelago also feature

quite high appearance rates of around 0.1 to 0.3. Contrary to earlier reports (Barber and Massom, 2007), the North-East Water (NEW) polynya in north-eastern Greenland (approx. 81° N, 13° W) neither shows any sign of disappearance, nor is it limited to the spring to late autumn period. With average frequencies of around 0.1 to 0.25 each month in winter, it has more likely to be categorized as a regularly forming polynya. Comparatively low frequencies below 0.15 (especially from January to March) are primarily found in the Beaufort and Chukchi Sea as well as in the East Siberian Sea. Vast fast-ice areas, e.g. along the

Siberian coast, can be detected from monthly TIT frequencies, as these areas usually appear at fixed locations attached to the shore and TIT frequencies tend towards zero as the ice quickly thickens by congelation ice growth. Hence, our 13-year record of monthly TIT-occurrence rates offers the potential to further develop optimized automatic methods for a regular Arctic-wide mapping of monthly fast-ice extents and could thereby complement currently existing approaches from earlier studies (e.g. Yu et al., 2014; Mahoney et al., 2014; Selyuzhenok et al., 2015).

Compared to the study of Willmes and Heinemann (2016), leads are only weakly visible in these long-term averages (frequencies below 0.05–0.1). In Fig. 4, leads are mainly located in the area of the Beaufort Sea and north of Greenland (shear zones) which can be mainly attributed to their relatively high spatial and temporal persistence. Frequent lead occurrences in e.g. the East-Siberian Sea found by Willmes and Heinemann (2016) are not reflected in our study. In some regions, however, the influence of (shelf-) bathymetry and associated ocean currents on the spatial distribution of polynya and lead occurrences

is also visible in our here derived TIT frequencies (e.g. eastern exit Vilkitsky Strait, Hanna Shoal / northern Chukchi Shelf, northern ESS).

In Fig. 5 and Fig. 6, the interannual variability of the average POLA (in $\mathrm{km}^2$) and accumulated IP (in $\mathrm{km}^3$) are presented for all examined polynya regions, respectively. In both figures, the difference between the beginning (November to December) and end (January to March) of the freezing (winter) period is additionally highlighted. Concerning POLA, it shows that the largest

average wintertime extents are found in the NOW, WNZ and KAR areas. The study of Preußer et al. (2015a) demonstrated that the large POLA values in the NOW-region are part of a (non-significant) long-term increase of average polynya extents between 1978 and 2015. In case of polynyas in proximity of Novaya Zemlya, the here derived average wintertime value for POLA of around 42 x $10^3$ $\mathrm{km}^2$ is fairly close to the respective value by Iwamoto et al. (2014) (49 x $10^3$ $\mathrm{km}^2$), despite the circumstances that their study covers an extended winter period from September to May and features a different mask area, which stretches

over some part of the western Kara Sea. As mentioned earlier, Kern (2008) presented POLA values for the Kara Sea. His

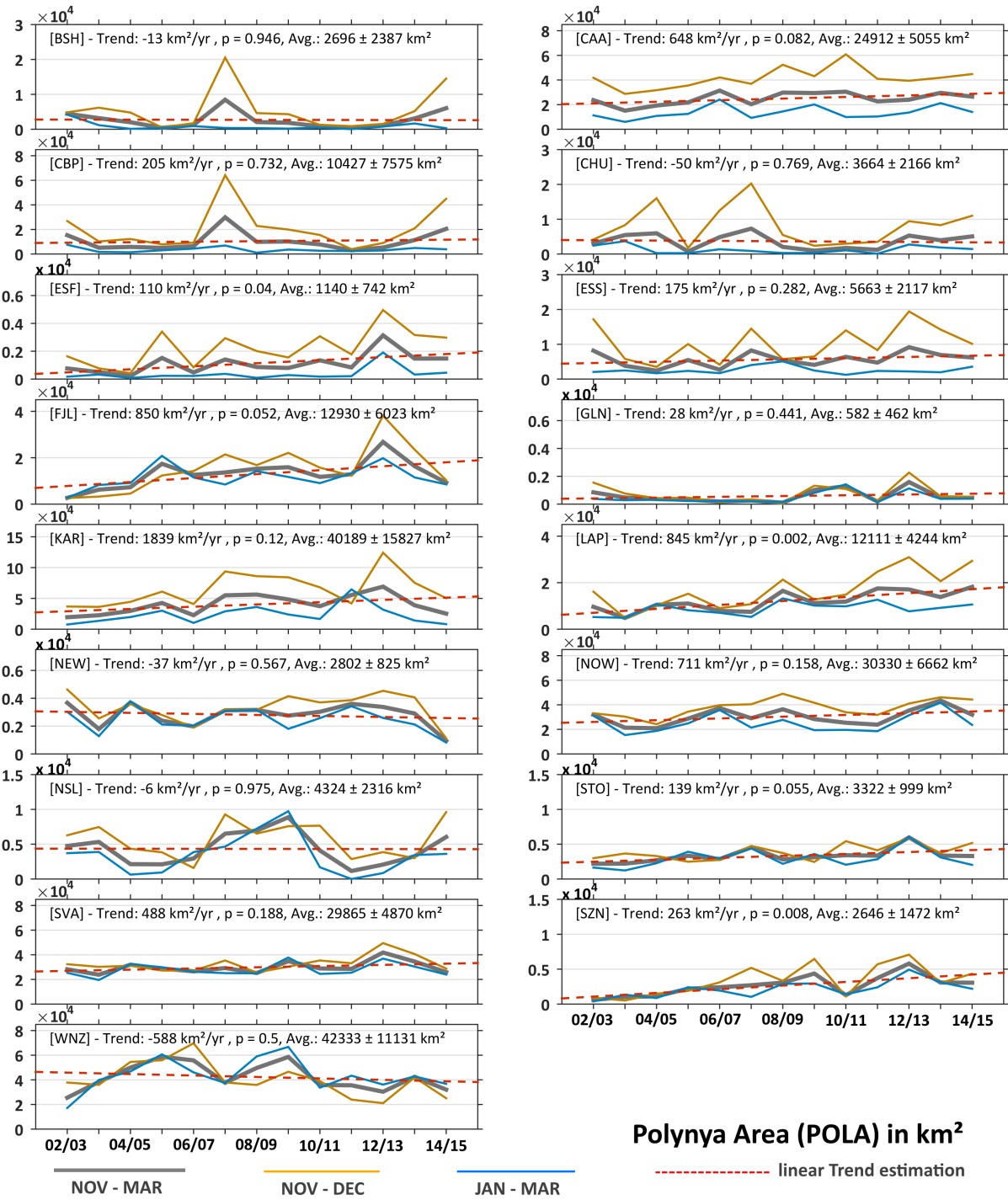

**Figure 5.** Regional time series of the annual average polynya area (POLA; TIT $\leq 0.2$ m) in $km^2$ for 2002/2003 to 2014/2015, together with a seasonal comparison (November to December vs. January to March) and a linear trend estimation. The estimated linear trend (in $km^2$/yr), its p-value and the interannual average POLA (in $km^2$) are additionally listed in each sub-panel. Please note the varying scale on each y-axis.

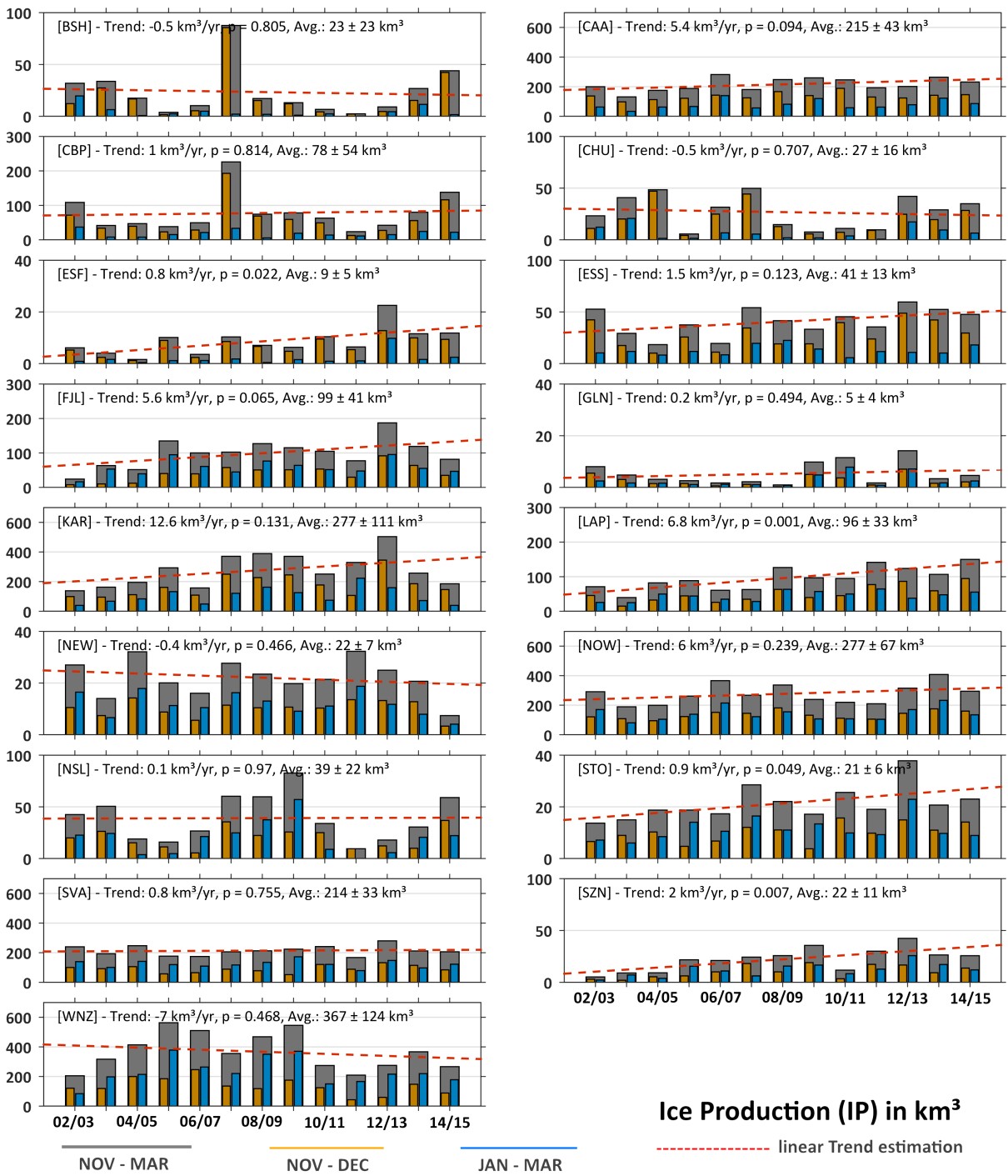

**Figure 6.** Regional time series of the annually accumulated ice production (IP) in $\mathrm{km}^3$ for 2002/2003 to 2014/2015, together with a seasonal comparison (November to December vs. January to March) and a linear trend estimation. The estimated linear trend (in $\mathrm{km}^3/\mathrm{yr}$), its p-value and the interannual average IP (in $\mathrm{km}^3$) are additionally listed in each sub-panel. Please note the varying scale on each y-axis.

**Table 2.** Average polynya area (POLA) in $km^2$ in each polynya region between 2002/2003 and 2014/2015 (SFR cloud-cover correction applied). Besides being based on the available winter period from November to March, it is further separated between the early freezing season (November to December) and the late freezing season (January to March). All values are derived from daily MODIS TIT composites after application of the predefined polynya masks (Fig. 1). Trends are additionally given, where underlined, bold and bold italic numbers denote statistical significance (two-sided t-test) at the 90, 95 and 99 % level, respectively.

| Region | November to March | | November to December | | January to March | |
|---|---|---|---|---|---|---|
| | *Avg. POLA* $(10^3 km^2)$ | *Trend POLA* $(km^2/yr)$ | *Avg. POLA* $(10^3 km^2)$ | *Trend POLA* $(km^2/yr)$ | *Avg. POLA* $(10^3 km^2)$ | *Trend POLA* $(km^2/yr)$ |
| Beaufort Shelf (BSH) | $2.7 \pm 2.4$ | -13 | $5.5 \pm 5.8$ | 132 | $0.8 \pm 1.1$ | -111 |
| Canadian Arctic Archipelago (CAA) | $24.9 \pm 5.1$ | 648 | $41.5 \pm 8.3$ | 958 | $13.7 \pm 5.2$ | 438 |
| Cape Bathurst (CBP) | $10.4 \pm 7.6$ | 205 | $20.6 \pm 16.9$ | 581 | $3.6 \pm 2.0$ | -49 |
| Chukchi Sea (CHU) | $3.7 \pm 2.2$ | -50 | $8.2 \pm 5.7$ | -95 | $1.3 \pm 1.1$ | -37 |
| East Siberian Fast-Ice (ESF) | $1.1 \pm 0.7$ | **110** | $2.3 \pm 1.3$ | **200** | $0.4 \pm 0.5$ | 48 |
| East Siberian Sea (ESS) | $5.7 \pm 2.1$ | 175 | $10.3 \pm 5.2$ | 385 | $2.5 \pm 1.1$ | 33 |
| Franz-Josef-Land (FJL) | $12.9 \pm 6.0$ | 850 | $15.2 \pm 9.7$ | **1565** | $11.5 \pm 4.9$ | 380 |
| Greenland North (GLN) | $0.6 \pm 0.5$ | 28 | $0.8 \pm 0.6$ | 14 | $0.5 \pm 0.4$ | 37 |
| Kara Sea (KAR) | $40.2 \pm 15.8$ | 1839 | $64.9 \pm 26.8$ | 3209 | $23.5 \pm 15.5$ | 904 |
| Laptev Sea (LAP) | $12.1 \pm 4.2$ | ***845*** | $17.0 \pm 8.0$ | ***1559*** | $8.8 \pm 2.7$ | 362 |
| North East Water (NEW) | $2.8 \pm 0.8$ | -37 | $3.3 \pm 1.0$ | -10 | $2.4 \pm 0.9$ | -55 |
| North Water (NOW) | $30.3 \pm 6.7$ | 711 | $37.7 \pm 7.1$ | **1072** | $25.4 \pm 7.8$ | 464 |
| Nares Strait / Lincoln Sea (NSL) | $4.3 \pm 2.3$ | -6 | $5.7 \pm 2.6$ | 18 | $3.4 \pm 2.8$ | -22 |
| Storfjorden (STO) | $3.3 \pm 1.0$ | 139 | $3.9 \pm 1.1$ | **175** | $2.9 \pm 1.3$ | 115 |
| Svalbard Archipelago (SVA+STO) | $29.9 \pm 4.9$ | 488 | $32.8 \pm 6.5$ | 732 | $27.9 \pm 5.3$ | 327 |
| Severnaya Zemlya North (SZN) | $2.6 \pm 1.5$ | ***263*** | $3.4 \pm 2.2$ | **353** | $2.1 \pm 1.2$ | **203** |
| Western Novaya Zemlya (WNZ) | $42.3 \pm 11.1$ | -588 | $40.4 \pm 13.8$ | -1806 | $43.6 \pm 13.0$ | 230 |
| Total | $226.6 \pm 36.1$ | **5468** | $309.4 \pm 62.6$ | 8864 | $171.3 \pm 32.6$ | 3151 |

retrievals are based on approximately the same reference area (Fig. 1), which in this case allows for a fair comparison to the here presented numbers. It shows that the average POLA in the late freezing season reveals similar magnitudes in recent years. During the period from 1979 and 2004, the average POLA (in Kern (2008): January to April) ranged between 1 to 5 x $10^4$ $km^2$ (except for 1995: around 6 x $10^4$ $km^2$), which is close to the range of our here presented results for January to March

5 (Fig.5; Tab. 2). Although the estimated positive trend in POLA remains non-significant for the Kara Sea as in Kern (2008), the magnitude of the trend in the late freezing period (January to March; around 9000 $km^2$/decade) seems to have increased from 2400 $km^2$/decade (Kern, 2008) to around 9000 $km^2$/decade over the last 13 years. The interannual POLA variability in all regions is generally pronounced, but is especially large for smaller polynyas / thin-ice regions such as the NSL, NEW and ESS. Concerning seasonal differences, it appears that some regions (e.g. NEW, GLN, LAP, SZN) have the tendency towards larger

10 thin-ice areas during the freeze-up period since approximately 2006/2007 to 2007/2008. About 8 to 10 polynya regions show distinct positive trends of up to 18,390 $km^2$ per decade (KAR), with only the LAP, ESF and SZN regions being significant (two-sided t-test) with $p \leq 0.05$. Interestingly, sub-regions located in proximity of the Beaufort Gyre (BSH and CBP) indicate very large thin-ice areas between November and December 2007, shortly after the 2nd lowest September sea-ice extent since

**Table 3.** Average accumulated ice production (IP) in $km^3$ in each polynya region between 2002/2003 and 2014/2015 (SFR cloud-cover correction applied). Besides being based on the available winter period from November to March, it is further separated between the early freezing season (November to December) and the late freezing season (January to March). All values are derived from daily MODIS TIT composites after application of the predefined polynya masks (Fig. 1). Trends are additionally given, where underlined, bold and bold italic numbers denote statistical significance (two-sided t-test) at the 90, 95 and 99 % level, respectively.

| Region | November to March | | November to December | | January to March | |
| --- | --- | --- | --- | --- | --- | --- |
| | *Acc. IP* ($km^3$) | *Trend IP* ($km^3/yr$) | *Acc. IP* ($km^3$) | *Trend IP* ($km^3/yr$) | *Acc. IP* ($km^3$) | *Trend IP* ($km^3/yr$) |
| Beaufort Shelf (BSH) | $23 \pm 23$ | -0.5 | $19 \pm 23$ | 0.0 | $5 \pm 5$ | -0.4 |
| Canadian Arctic Archipelago (CAA) | $215 \pm 43$ | 5.4 | $136 \pm 23$ | 2.5 | $79 \pm 31$ | 2.9 |
| Cape Bathurst (CBP) | $78 \pm 54$ | 1.0 | $60 \pm 48$ | 1.1 | $18 \pm 10$ | -0.1 |
| Chukchi Sea (CHU) | $27 \pm 16$ | -0.5 | $20 \pm 14$ | -0.3 | $7 \pm 6$ | -0.2 |
| East Siberian Fast-Ice (ESF) | $9 \pm 5$ | **0.8** | $7 \pm 3$ | **0.6** | $2 \pm 2$ | 0.3 |
| East Siberian Sea (ESS) | $41 \pm 13$ | 1.5 | $28 \pm 13$ | 1.3 | $13 \pm 5$ | 0.2 |
| Franz-Josef-Land (FJL) | $99 \pm 41$ | 5.6 | $42 \pm 24$ | **4.1** | $57 \pm 22$ | 1.5 |
| Greenland North (GLN) | $5 \pm 4$ | 0.2 | $3 \pm 2$ | 0.0 | $3 \pm 2$ | 0.2 |
| Kara Sea (KAR) | $277 \pm 111$ | 12.6 | $174 \pm 76$ | 9.0 | $104 \pm 56$ | 3.6 |
| Laptev Sea (LAP) | $96 \pm 33$ | ***6.8*** | $51 \pm 24$ | ***4.8*** | $45 \pm 14$ | **2.0** |
| North East Water (NEW) | $22 \pm 7$ | -0.4 | $10 \pm 3$ | 0.0 | $12 \pm 4$ | -0.4 |
| North Water (NOW) | $277 \pm 67$ | 6.0 | $135 \pm 27$ | 3.4 | $145 \pm 45$ | 2.6 |
| Nares Strait / Lincoln Sea (NSL) | $39 \pm 22$ | 0.1 | $20 \pm 10$ | 0.2 | $20 \pm 16$ | -0.1 |
| Storfjorden (STO) | $21 \pm 6$ | **0.9** | $10 \pm 4$ | **0.5** | $11 \pm 4$ | 0.4 |
| Svalbard Archipelago (SVA+STO) | $214 \pm 33$ | 0.8 | $91 \pm 24$ | 1.6 | $123 \pm 24$ | -0.8 |
| Severnaya Zemlya North (SZN) | $22 \pm 11$ | ***2.0*** | $10 \pm 6$ | **0.9** | $12 \pm 6$ | **1.1** |
| Western Novaya Zemlya (WNZ) | $367 \pm 124$ | -7.0 | $136 \pm 56$ | -6.8 | $231 \pm 88$ | -0.2 |
| Total | $1811 \pm 293$ | 34.5 | $940 \pm 178$ | 22.4 | $871 \pm 175$ | 12.1 |

1979 (approx. 4.7 million $km^2$; Parkinson and Comiso (2013)). This did not appear in a similar way in 2012 (record low of approx. 3.4 million $km^2$). A detailed investigation shows that the freeze-up in the Beaufort Sea area was much slower in 2007 and extended until mid-December, while in 2012 the same area was ice-covered by November 10. The study of Timmermans (2015) linked this significant delay in ice growth to upward mixing processes of ocean heat in the Canada Basin, originating from the release of stored solar heat input following summer 2007. This resulted in large areas with very thin ice (around 170,000 $km^2$) in November to December and consequently allowed for huge amounts of latent and sensible heat to be released from the ocean, leading to extraordinary high IP values in these areas (Fig. 6).

Regarding IP, many of the above described features are also visible in the regional time series of Fig. 6. Contrary to Tamura and Ohshima (2011), the majority of polynya regions shows overall positive (up to 126 $km^3$ per decade (KAR)) or no trends in wintertime ice production, and only four regions indicate a slight, yet insignificant decrease over the last 13 years (BSH, CHU, NEW, WNZ). Complete overviews on calculated average POLA and IP values per region, together with their respective trends, are given in Tab. 2 and Tab. 3, respectively. These overviews highlight that seasonal differences (November to December vs. January to March) have a huge effect on calculated average values and trends for the complete winter period from November

to March. Consequently, the here discussed numbers should be regarded as winter integrals with potentially inherent effects originating from the timing of freeze-up onset. In case of e.g. the Kara Sea, Franz-Josef-Land, the Chukchi Sea and the Canadian Arctic Archipelago, large thin-ice and potential open-water areas during the early freezing period in November and December imprint on the total winter averages as well as derived trends of POLA and IP, especially from 2007/2008 onwards.

While the majority of polynyas also feature positive trends in the late freezing season from January to March, these trends are for the most part not significant.

The average total ice production in Arctic polynyas sums up to $1811 \pm 293$ km$^3$ per winter. Thus, it lies in between previously determined average values of $2940 \pm 373$ km$^3$ (Tamura and Ohshima, 2011; 1992/1993 - 2007/2008) and $1178 \pm 65$ km$^3$ (Iwamoto et al., 2014; 2002/2003 - 2010/2011) per winter. We expect that the MODIS-derived quantities offer

a valuable increase in both spatial and quantitative accuracy due to the use of high-resolution and gap-filled daily fields of thin-ice thicknesses. A shortening of the averaging interval to the period 2002/2003 - 2010/2011 (as in Iwamoto et al. (2014), but not accounting for differences in covered winter period) reduces the here derived average total ice production marginally by about 1-2%. In order to assess apparent differences between our here derived data-set and the passive microwave data set by Iwamoto et al. (2014), a more direct comparison based on identical reference areas and the same winter period would be

necessary.

A spatial overview of the average (2002/2003 to 2014/2015) accumulated ice production per winter (November to March) is presented in Fig. 7. Likewise to Fig. 4, the NOW polynya stands out at first glance due to its high average ice production of up to 14 m per winter. However, smaller polynyas in the Canadian Arctic (around Devon Island) feature comparatively high values for ice production. Most other areas in the Arctic produce on average between 1-3 m of ice per winter, with a few

noticeable exceptions like Franz Josef Land (about 4-5 m per winter), the northern tip of Novaya Zemlya (5-7 m per winter) and some coastal areas in the Kara Sea (1-4 m per winter). While the core areas of high ice production show a high resemblance to Iwamoto et al. (2014) with marginal differences in absolute numbers, MODIS is capable to provide enhanced spatial detail. This is especially valuable concerning the narrow thin-ice areas along the coast and fast-ice edges in the eastern part of the Arctic (Kara Sea, Laptev Sea, East Siberian Sea), as these areas are not resolved by the coarse-resolution passive microwave data

(6.25 km; Iwamoto et al., 2014). This striking advantage is also reflected in the comparatively narrow fjords and bays/sounds around Greenland and the Canadian Archipelago, where a high ice production of up to 3 meters per winter is found. While these observations, mostly related to differences in spatial resolution, could explain the above described discrepancy in average accumulated numbers to some extent (compare Preußer et al. (2015a)), the net effect of a lower grid size cannot be quantified here.

Spatial trends between the winter seasons 2002/2003 and 2014/2015 (November to March) can be calculated by applying a linear regression on the annual accumulated IP per pixel. The resulting map is shown in Fig. 8 (a). Besides many interesting small-scale patterns, two main conclusions can be drawn from this spatial overview: (1) While the trends identified in the western Arctic show no consistent pattern, large areas of the eastern Arctic are characterized by significant (two-sided t-test; significance levels indicated in Fig. 8 (b)) positive trends that can exceed 2 meters per decade and (2) we observe opposing

negative / positive IP-trends along the coasts of the Laptev and Kara Seas which could be due to changes in fast-ice extent

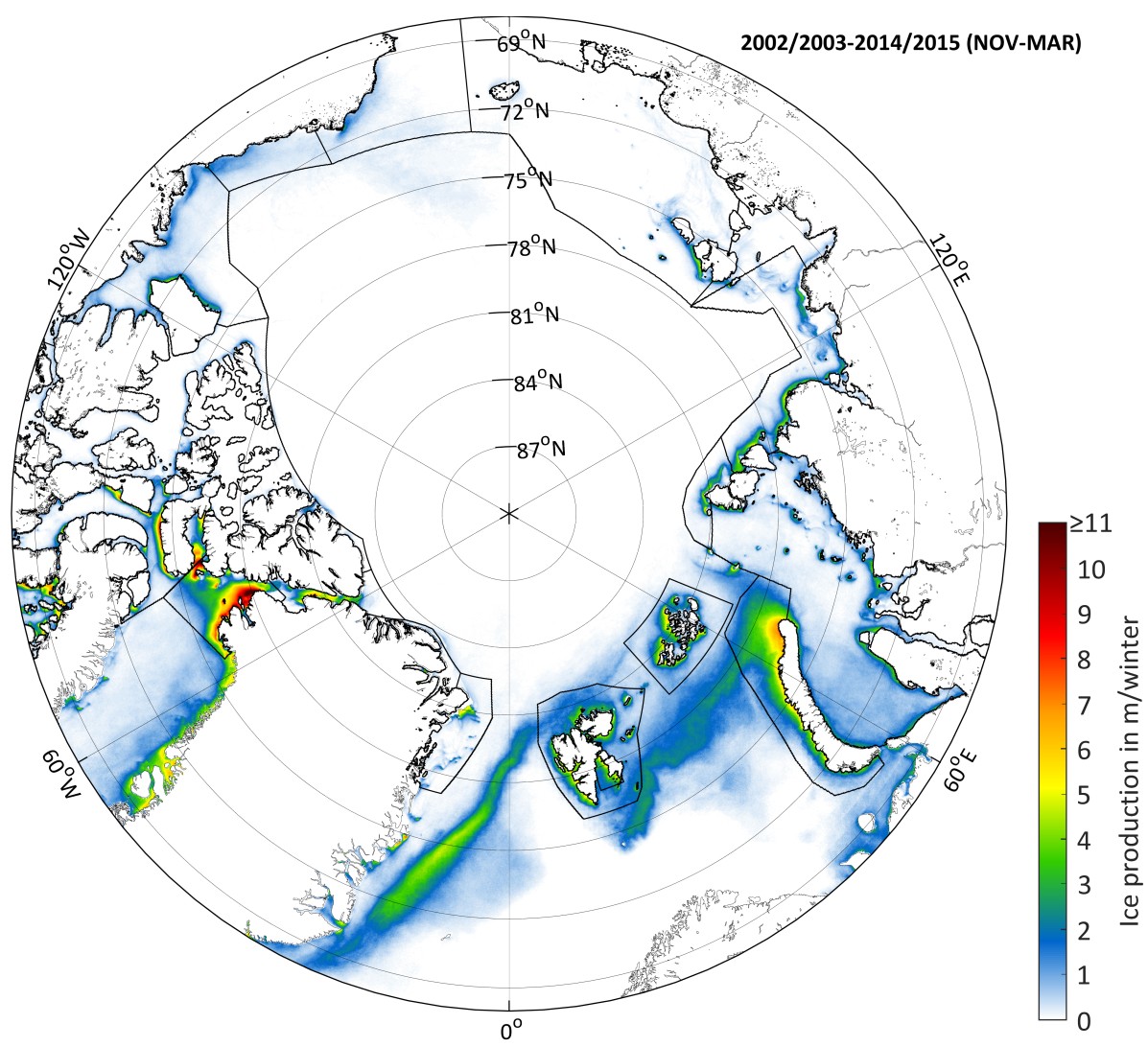

**Figure 7.** Average (2002/2003 to 2014/2015) accumulated ice production (m per winter) during winter (November to March) in the Arctic, north of 68° N. The margins of applied polynya masks (Fig. 1) are shown in black dashed lines.

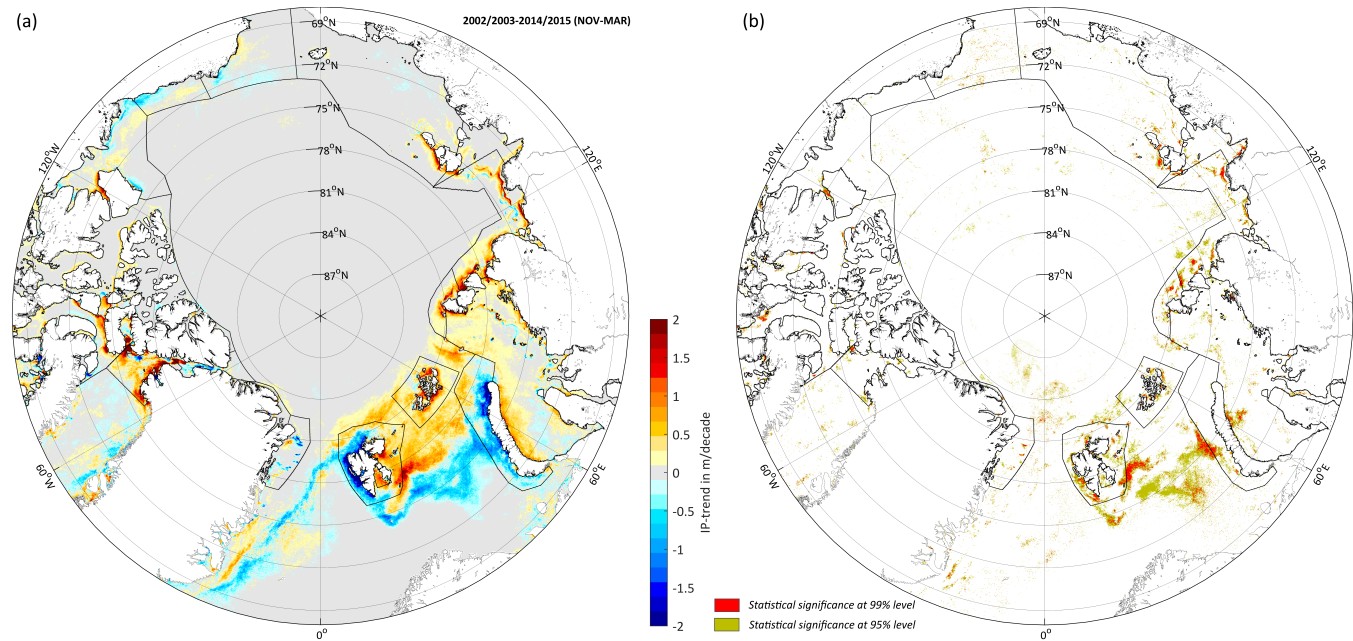

**Figure 8.** (a) Decadal trends (m per decade) of wintertime (November to March) ice production in the Arctic, north of $68°$ N. Trends are calculated by applying a linear regression on the annual accumulated IP per pixel for the period 2002/2003 to 2014/2015. Areas with statistical significance (based on a two-sided t-test) at the 95% and 99% level are depicted in (b). The margins of applied polynya masks (Fig. 1) are shown in black dashed lines.

over the 13-year period. Decreasing fast-ice extents and durations in the eastern Arctic between 1976 and 2007 were recently described by Yu et al. (2014). In addition, Selyuzhenok et al. (2015) analyzed the fast ice in the south-eastern Laptev Sea in more detail (1999 to 2013). While their study showed that the winter maximum fast-ice extent (March/April) as well as the shape and location of the fast-ice edge did not vary significantly over the regarded time period, they likewise presented an

5 overall decrease in the fast-ice season (-2.8 $\mathrm{d/yr^{-1}}$) due to a later formation and earlier break-up. These described changes regarding the timing of fast-ice formation in early winter could explain the observed structures of positive / negative trends in proximity of fast-ice areas.

In order to put these observations into context, we suppose that this characteristic pattern of opposing trends in the western and eastern Arctic as well as the apparently fast-ice related structures in the Laptev Sea and Kara Sea could be connected to

10 an overall later appearing fall freeze-up (Markus et al., 2009; Stroeve et al., 2014) in recent years, which itself is thought to result from a complex mixture/interplay of steadily and year-round increasing (2m-) air temperatures (e.g. Cohen et al., 2014), distinct large-scale atmospheric patterns (e.g. Rigor et al., 2002) and the overall downward trend of total sea-ice extent and volume in the Arctic (e.g. Schweiger et al., 2011; Laxon et al., 2013). The latter implies a tendency towards a more fragile, thus mobile, sea-ice cover in the Arctic, with a potentially increased sensitivity for external forcing mechanisms (i.e. strong winds

15 and/or ocean currents) that are responsible for thin-ice formation in polynyas and leads. As being one of the main regions with

highly pronounced and significant positive trends in both POLA and IP throughout the complete winter period, the following section will take a closer look on polynya dynamics in the Laptev Sea.

## 4.2 Regional focus - Laptev Sea

One main advantage of the high-resolution MODIS data is the ability to perform detailed investigations on a regional scale across the Arctic. The grid spacing of 2 km allows for the detection of relatively fine and delineated polynya structures and for more accurate statements about areas of high ice production, as it was previously possible.

The Laptev Sea was previously described as a key region to investigate climatic changes in the Arctic shelf seas (ACIA, 2005), as it is one of the major source area for sea ice export into the Transpolar Drift system (Dethleff et al., 1998). As can be seen in Fig. 9, the Laptev Sea is located between the Severnaya Zemlya at the western boundary, the Lena Delta at the southern edge and the New Siberian Islands, which serve as the boundary in the East (approximately 70-80°N, 100-140°E). The water-mass composition in the Laptev Sea is temporarily quite variable, as there is a huge freshwater inflow during the summer and autumn period (around 750 km$^3$ per year; Rigor and Colony (1997)) and strong ice-formation accompanied with brine rejection in polynyas during winter (Bauch et al., 2012). These processes significantly alter the stratification of the upper ocean layers as well as the salinity levels in the annual cycle. These and other recurring features of the sea ice and ocean environments have recently been illustrated and updated by Janout et al. (2016).

During the freezing period (roughly October to June), fast ice forms along the coastlines of the Laptev Sea, which usually reaches its maximum areal extent by April. The approximate location of the fast-ice edge at the end of March is depicted in Fig. 9. For drifting sea ice, the fast-ice edge forms an advanced coast line with heavy ridging occurring along this edge during onshore wind events (Rigor and Colony, 1997). The combination of this fast-ice edge and off-shore components of the mean wind-patterns enable the formation of several flaw-lead polynyas across the Laptev Sea which can reach widths of up to 200 km (Bareiss and Görgen, 2005; Martin and Cavalieri, 1989; Ernsdorf et al., 2011; Adams et al., 2013).

When comparing previous studies dealing with ice production rates in the Laptev Sea (Dethleff et al., 1998; Winsor and Björk, 2000; Dmitrenko et al., 2009; Willmes et al., 2011; Tamura and Ohshima, 2011; Bauer et al., 2013; Iwamoto et al., 2014; Gutjahr et al., 2016), it gets clear that there are large differences depending on the applied methods and various different data-sets. In these studies, values for the accumulated ice production during an average winter season ('extended' winter-period from November to April) are ranging between 55 km$^3$ (Willmes et al., 2011) for an approach using microwave and thermal infrared remote sensing data in combination with atmospheric reanalysis data, and 258 km$^3$ (Dethleff et al., 1998), who used a simple relationship between wind direction/speed and polynya area. Estimated average values (September to May) from Tamura and Ohshima (2011) (152 km$^3$) and Iwamoto et al. (2014) (77 km$^3$) range in between. Although derived for different time periods and slightly varying reference areas, these large discrepancies highlight the relevance of applying improved and high-resolution approaches to quantify sea-ice production.

In order to give an overview on the long-term development of thin-ice areas (TIT $\leq$ 0.2 m) in the Laptev Sea, the daily POLA is presented in Fig. 10. It is evident that the largest areas of thin-ice appear on average in November and more recently also in December (compare Tab. 2). A tendency towards an increased duration of these polynya-events can be observed. In

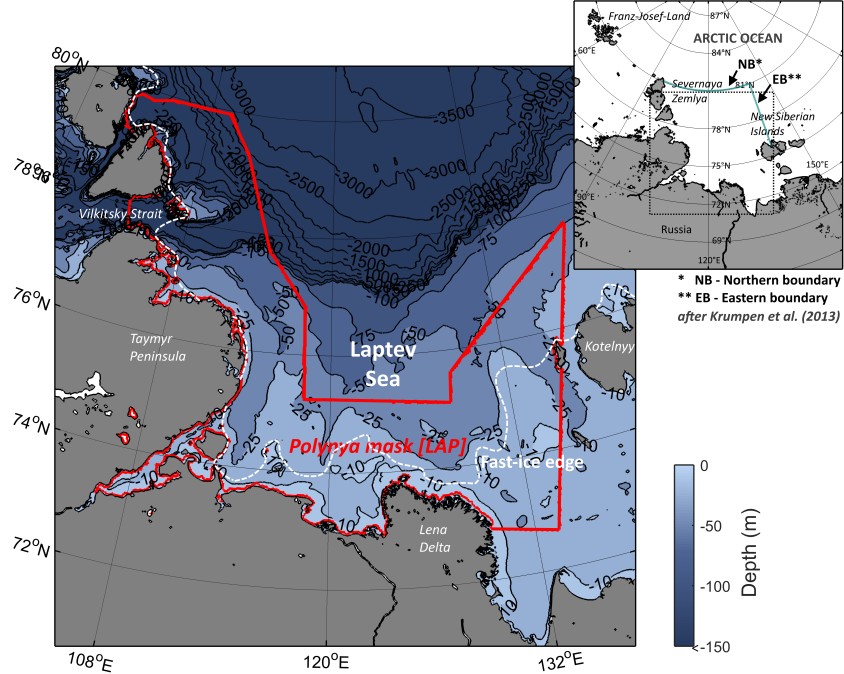

**Figure 9.** The geographical location of the Laptev Sea in the eastern Arctic. The applied polynya mask is marked in red, enclosing the locations of typical polynya formation along the coast and fast-ice edge (dashed white line; position derived from long-term thin-ice frequencies in March (Fig. 4)). Flux gates from the study by Krumpen et al. (2013) at the northern (NB) and eastern (EB) boundary of the Laptev Sea are shown in the inset map (cyan solid lines). Bathymetric data by Jakobsson et al. (2012) (IBCAO v3.0).

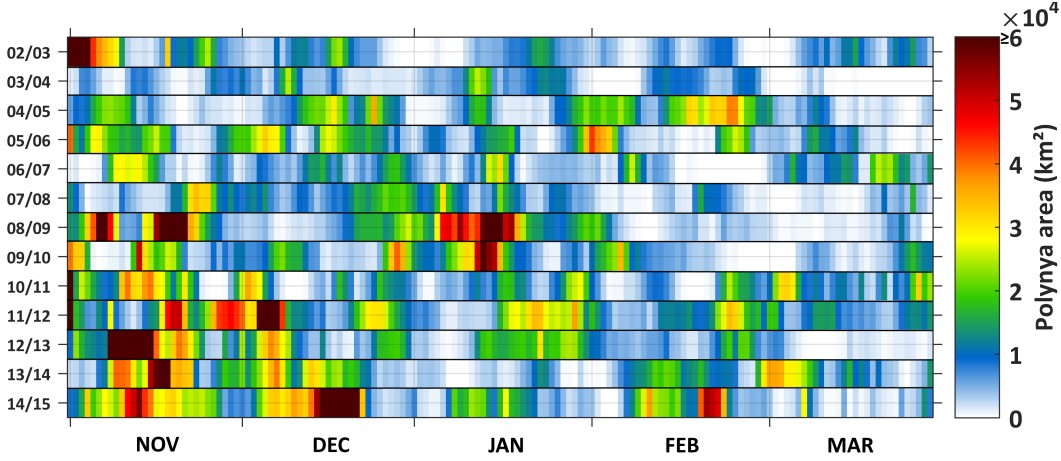

**Figure 10.** The daily polynya area (TIT $\leq$ 0.2 m) in the Laptev Sea region for the winter seasons 2002/2003 to 2014/2015. Values are calculated within the margins of the applied polynya mask (Fig. 1) and saturated at a level of 6 x $10^4$ km$^2$ for a better discrimination of lower values.

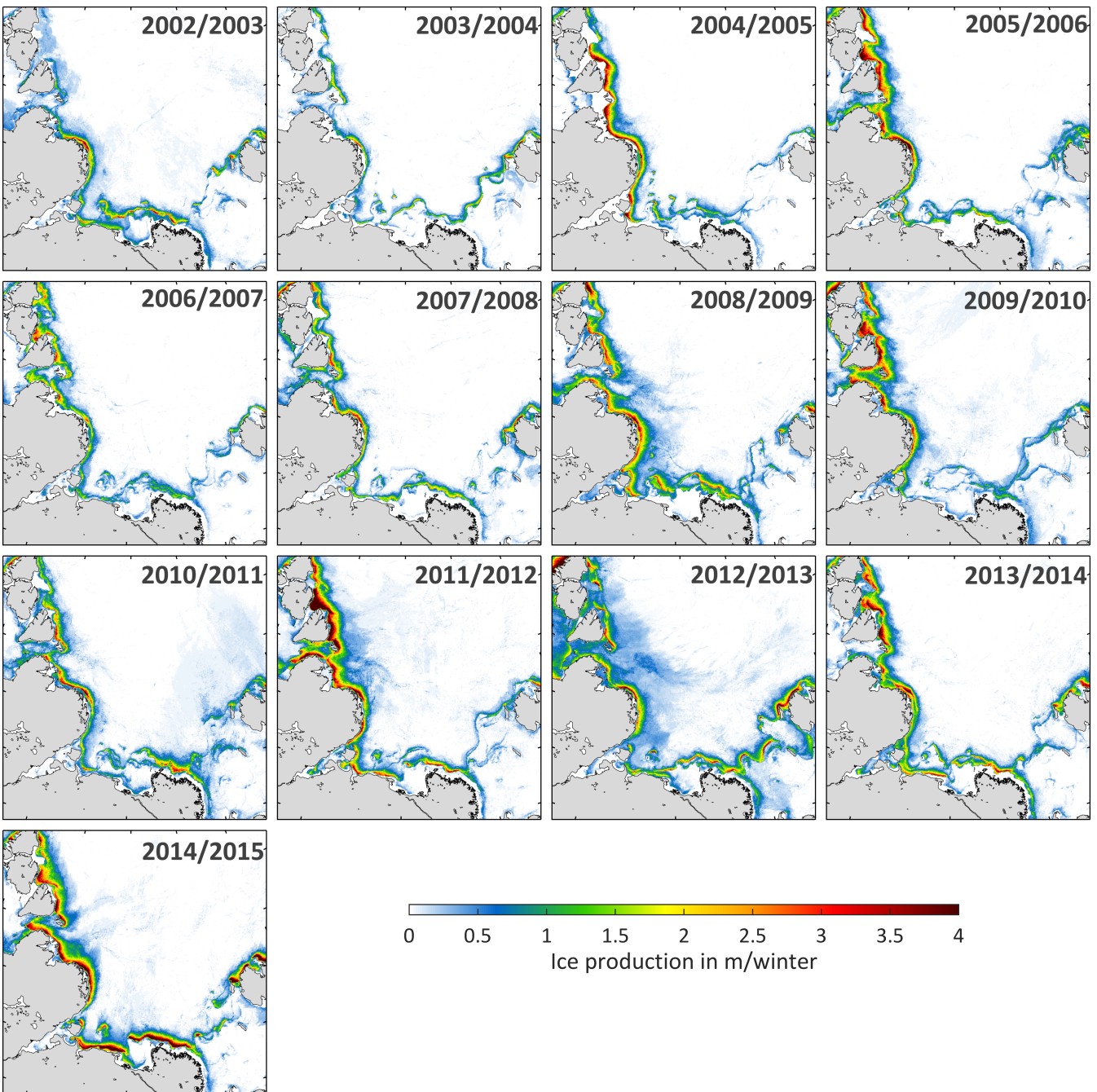

**Figure 11.** Overview of wintertime (November to March) accumulated ice production (m per winter) in the Laptev Sea region between 2002/2003 and 2014/2015.

the winter-seasons 2008/2009 and 2009/2010, large POLA exceeding 50000 $km^2$ are also observed in January and another major polynya event can be noted for mid-February 2015. A pronounced seasonal variation is visible for the winter seasons 2004/2005, 2005/2006 and from 2010/2011 onwards, while the other years show less polynya activity (more lengthy periods with a closed polynya; white color in Fig. 10) and overall smaller polynya extents in February and March.

Fig. 11 shows an annual comparison (2002/2003 to 2014/2015) of accumulated (November to March) ice production (in m per winter) for the Laptev Sea. The highest ice-production rates of sometimes more than 4 m per winter occur predominantly in proximity of the Taymyr Peninsula and Severnaya Zemlya (western Laptev Sea), as well as along the southern fast-ice edge (mainly coastwards of the regions with high ice production). However, ice production in the eastern Laptev Sea (west and north of the New Siberian Islands) shows a greater interannual variability. Furthermore, it is striking that the position of the fast-ice

edge in Fig. 11 is highly variable over the 13-yr record (as in Sect. 4.1, Fig. 8 (a)). However, it has to be noted that certain bands of higher ice production, especially in the south-eastern Laptev Sea, reflect the wintertime evolution of fast ice (compare Selyuzhenok et al., 2015) and are hence primarily related to the early winter period from November to December. Another interesting observation can be made in the Vilkitsky Strait, which is located in the western Laptev Sea south of Severnaya Zemlya (Fig. 9). The distribution of thin-ice areas contributing significantly to the total sea-ice production in that area seems

to shift westwards towards the Kara Sea in several years (2005/2006 to 2012/2013 and 2014/2015). In some cases, the shape of these areas resembles an arch-type/ice-bridge pattern/mechanism, a feature that is commonly appearing e.g. in Nares Strait between Ellesmere Island and Greenland (Williams et al., 2007).

Krumpen et al. (2013) discovered that most of the ice being incorporated in the Transpolar Drift originates from the western and central part of the Laptev Sea. Further, it was indicated that the contribution from polynyas, while being generally small,

is limited to events in proximity of the Laptev Sea boundaries. As noted before, the north-western Laptev Sea shows by far the largest contribution to the total wintertime ice production in the Laptev Sea polynyas, which implies a potential significant influence on the interannual variability of the ice export during winter. In order to check this hypothesis, we compare annual accumulated IP values to independently derived ice-area export (IAE) values (both presented as anomalies and normalized with their standard deviation) in Fig. 12 for 2002/2003 to 2014/2015. IAE values are taken from the updated time series of Krumpen

et al. (2013), where they were calculated as the integral of the product between the eastward and northward component of the ice drift velocity and ice concentration at the northern boundary (NB) and eastern boundary (EB) of the Laptev Sea, respectively. Likewise to a high agreement between polynya area and across-boundary ice export (Krumpen et al., 2013), there is also a significant correlation between calculated ice production and the areal ice export (r = 0.69 with p = 0.009).

The spatial overview of annual ice production (Fig. 11) is supplemented by the previously shown time series of the average

wintertime POLA and accumulated IP per winter (Fig. 5 and Fig. 6, respectively). Both time series of POLA and IP in the Laptev Sea show an overall positive trend (significant with p $\leq$ 0.01), which can for the most part be traced back to larger thin-ice areas during the freeze-up period in November and December (as decribed above; Fig. 10). This is underlined by Tab. 2 and 3, which both reveal largest average values of POLA / IP and most significant trends during that period of winter. The average ice production from November to March in the Laptev Sea is estimated with about 96 $\pm$ 33 $km^3$ (2002/2003 -

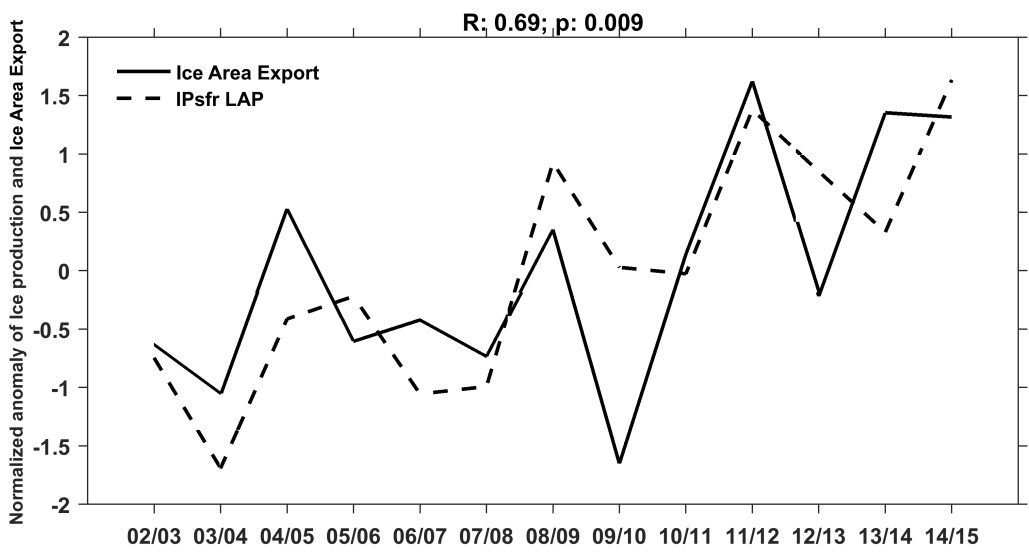

**Figure 12.** Normalized anomalies of accumulated wintertime ice production (IP of the present study; dashed line) and accumulated Ice Area Export (IAE; solid line) for winter seasons 2002/2003 to 2014/2015. IAE data is based on an updated time series by Krumpen et al. (2013).

2014/2015), with a positive trend of 6.8 $km^3$ per year. Compared to other Arctic polynyas (compare Tab. 3), this corresponds to a share of about 5% of the total ice production in polynya regions.

As the relative strength of the Transpolar Drift is dependent on atmospheric dynamics, it has previously been linked to atmospheric indices like the Arctic Oscillation (AO) Index (Rigor and Wallace, 2004). For the period from 1982 to 2009, the study by Kwok et al. (2013) presented indicators for a net-strengthening of both the Transpolar Drift and the Beaufort Gyre as well as a general increase of the Arctic ice drift-speed, which is presumably related to a decreasing fraction of thick multi-year (MY) ice. As mentioned before (Sect. 4.1), the latter is thought to be connected to an increased fragility and mobility of the Arctic sea-ice cover, which may have implications for polynya and lead dynamics not only in the eastern Arctic. According to Rigor et al. (2002), a positive winter AO promotes both an increased ice transport out of the Arctic Ocean through Fram Strait and an increased ice transport away from the Siberian coastal areas, thereby leaving open water and thin ice that foster new ice formation. Hence, positive trends in both POLA and IP not only fit well to the previously estimated positive trend in IP from Iwamoto et al. (2014) but also to the positive trend of 0.85 x $10^5$ $km^2$ per decade in the Laptev Sea ice area flux (Krumpen et al., 2013). Other linkages and dependencies with the Arctic sea-ice extent in September (annual minimum), the timing of the freeze-onset and further connections to large-scale atmospheric circulation patterns are very likely and have been proposed by various previous studies (e.g. Alexandrov et al., 2000; Deser et al., 2000; Rigor et al., 2002; Willmes et al., 2011; Krumpen et al., 2013). Particularly a significant lengthening of the melt season in recent years and hence a later freeze-up in autumn already seems to imprint on the derived POLA (i.e. thin-ice area) and IP estimates in the early winter period (Markus et al., 2009; Parkinson, 2014; Stroeve et al., 2014). In that context, increasing atmosphere- and ocean-temperatures in

autumn and winter have recently been reported by Boisvert and Stroeve (2015) that comprise the potential to alter/shift vertical temperature gradients with consequences for the surface energy balance and ultimately IP. Further, a shortened fast-ice duration and enhanced variability of the fast-ice edge in early winter (Yu et al., 2014; Selyuzhenok et al., 2015) presumably influences the location of flaw leads and consequently high ice production / brine release. Frankly, all these (potential) interconnections are rather complex and would require more detailed investigations that go beyond the scope of the present study. In the context of other reported changes during the spring and summer period (Janout et al., 2016), it may emerge that the overall set-up for atmosphere-ice-ocean interactions in the Laptev Sea is gradually changing towards a new state.

## 5 Conclusions

In the present study we analyzed circumpolar polynya dynamics and ice production in the Arctic based on high-resolution MODIS thermal infrared imagery and atmospheric reanalysis from the ERA-Interim data set. Pan-Arctic and (quasi-) daily thin-ice thickness distributions were calculated using a 1D-energy balance model for the period from 2002/2003 to 2014/2015 (November to March). After applying a necessary and well-working gap-filling approach to compensate for cloud and data gaps, the thermodynamic ice production was derived by assuming that all heat loss at the ice surface contributes to the growth of sea ice. We presented results for 17 prominent polynya regions, whereby the a strong focus was set on the Laptev Sea region in the eastern Arctic. Despite existing limitations originating from the use of thermal infrared remote sensing data during winter, we think that this new data set of 13 consecutive winter seasons is a huge step forward for a spatially accurate characterization of Arctic polynya dynamics and the associated sea-ice budget related to winter-time sea-ice production. Our main findings and conclusions are the following:

(1) The use of high-resolution MODIS data enables the detection of thin ice much closer to the coast or fast-ice edges, mitigates land spill-over effects efficiently and generally increases the capability to resolve small scale (> 2km) thin-ice features such as narrow polynyas and leads, which therefore contribute to our ice production estimates. This represents an advantage compared to other (passive microwave) data sets.

(2) The average wintertime accumulated ice production in all 17 polynya regions is estimated with about $1811 \pm 293 \text{ km}^3$. The largest contributions originate from the western proximity of Novaya Zemlya (20%), the Kara Sea region and the North Water polynya (both 15%) as well as scattered smaller polynyas in the (eastern) Canadian Arctic Archipelago (all combined around 12%). By relying on predefined and fixed polynya masks, these IP estimates can include both thermodynamic ice growth within detected polynya margins (TIT $\leq 0.2$ m) as well as ice production in open ocean / MIZ areas. However, our estimate on the average total ice production exceeds that of Iwamoto et al. (2014) by about 52-54%. We note that differences in the regarded time frame, reference areas, sensor-specifics as well as a potential bias due to cloud cover and/or the exclusive assumption of clear-sky conditions certainly contribute to this discrepancy.

(3) Positive trends in ice production can be detected for several regions of the eastern Arctic (most significant in the Laptev Sea region with an increase of 6.8 km$^3$/yr and the North Water polynya, while other polynyas in the western Arctic show a more pronounced interannual variability. These regionally different trends could potentially originate from changes in the

overall sea-ice mobility (i.e. sea-ice drift), a temporal shift of the freeze onset in autumn (leading to larger thin-ice areas in November and December) or distinct large-scale atmospheric set-ups that promote an increased ice export and enhanced ice production in the Siberian shelf regions during winter.

(4) The Laptev Sea region was chosen as a special focus in our study as it is one of the core areas for ice production in the Arctic with a distinct connection to Transpolar Drift characteristics and showing a strong positive trend. Ice production in the Laptev Sea was mapped with enhanced spatial detail, which is especially valuable in this region with narrow and elongated flaw leads close to the fast-ice edge. Our results showed that polynyas in the Laptev Sea contribute with at least 5% to the total potential sea-ice production in Arctic polynyas. While the interannual variability in terms of location and extent seems to be rather high, the positive trends in both POLA and IP time series fit well to results and observations from other recently published studies in the Laptev Sea. A clear relation between increasing sea-ice area export (Krumpen et al., 2013) and positive trends in IP could be demonstrated, and future comparisons with recently derived volume-flux estimates in the Transpolar Drift (Krumpen et al., 2016) certainly promise further insights on the absolute contribution of polynyas to the volume ice export out of the Laptev Sea and adjacent seas.

(5) Compared to the MODIS-derived lead product by Willmes and Heinemann (2016), the SFR-algorithm used in the present study is not able to adequately reconstruct leads with a low spatial and temporal persistence. A thoughtful combination of both concepts is therefore a goal worth to achieve for future investigations on thin-ice regions in the polar regions using thermal infrared data from MODIS or other comparable satellite sensors, allowing for estimates of IP by leads also for the central Arctic ocean.

*Author contributions.* Andreas Preußer carried out the complete analysis and drafted the manuscript. Sascha Willmes and Stephan Paul supported the satellite data analysis and further processing. All co-authors contributed to the writing of the manuscript. The final draft of the manuscript was revised and approved by all of the authors.

*Acknowledgements.* The authors want to thank the National Snow and Ice Data Center (NSIDC) as well as the European Center for Medium-Range Weather Forecasts (ECMWF) for providing the MODIS Sea Ice product (ftp://n5eil01u.ecs.nsidc.org/SAN/) and the ERA-Interim atmospheric reanalysis data. The University of Bremen is also kindly acknowledged for providing the AMSR2 passive microwave sea ice concentration data. The provision of the ice area export data set by Dr. Thomas Krumpen (Alfred Wegener Institute, Bremerhaven, Germany) as well as related discussions are highly acknowledged. Additional thanks to the three referees and the editor Dr. Dirk Notz, who helped to improve the manuscript with their highly valuable comments and suggestions during the review. This work was funded by the Federal Ministry of Education and Research (Bundesministerium für Bildung und Forschung - BMBF) under Grant 03G0833D.

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
