# Peer review of "Circumpolar polynya regions and ice production in the Arctic: Results from MODIS thermal infrared imagery for 2002/2003 to 2014/2015 with a regional focus on the Laptev Sea"

_The Cryosphere, 2016_

## Referee Comment (RC1) · S. Kern (Referee) · 22 Aug 2016

Review of Circumpolar polynya regions and ice production in the Arctic: Results from MODIS thermal infrared imagery for 2002/2003 to 2014/2015 with a regional focus on the Laptev Sea by Preußer, A., et al.

Summary: The great potential of the combined ice surface temperature (IST) data sets derived from TERRA and AQUA MODIS infrared surface temperature observations is utilized to derive a pan-Arctic view of polynya area with unprecedented spatial grid resolution for such a long period of winters (Nov.-Mar.) spanning 2002/03 through

2014/15. Polynya area is derived by means of combining the IST with meteorological information provided by ERA Interim re-analysis data to estimate thin ice thickness (TIT). To overcome gaps due to cloud coverage an innovative, recently published approach is further developed and applied to the derived time-series of quasi-daily TIT maps. The final results: time series of distributions of polynya area, TIT and ice production are presented and discussed. The average polynya area and ice production are within the range of previous studies. Polynyas in the Eastern Arctic are found to have an increase in ice production for Nov.-Mar. over the time period considered.

General comments: 0) The paper is very well written and it reads fluently. The figures are mostly excellent. The paper presents the retrieval and discussion of a polynya area and ice production data set of yet unprecedented spatial resolution and hence for sure warrants publication. In the current version of the manuscript a few critical definitions and questions remain unanswered, though, which I feel are required to not misinterpret this very nicely written article. The discussion of potential uncertainties and biases in the retrieved data should be improved for the same reason. Finally, the inter-comparison to other studies and discussion of the differences to other studies by means of the material the authors already have in hands could be improved.

1) The abstract and conclusion write: "most accurate characterization of ..." I would rate it as important that the authors clearly state that they speak about spatial accuracy and not about retrieval accuracy of the thin ice thickness and ice production. In addition to that, as I write further down (in the context of the discussion with results about the polynya area from other authors), the authors could elaborate on the question whether the net effect of a finer grid resolution is solely an increase in the derived total polynya area, or whether the reduced smearing / smoothing for larger size thin ice areas when using MODIS data doesn't mean that derived polynya sizes could be also smaller.

2) Tied with accuracy is that, to my feeling, the retrieval accuracy of the method is discussed not enough. The only notion I found about the accuracy of the thin ice thickness retrieval is the one cited by Adams et al. (2013). It does not seem that the

authors did carry out accuracy investigations on their own.

This starts with the validity of using coarse resolution ERA-Interim data (coarse compared to MODIS) in a pan-Artic sense. Yes, for the Laptev Sea investigations published in the literature have shown that re-analysis data fit observations quite nicely, but this is an "easy" area in terms of topography. Areas around Greenland (NEW, NOW) and the Canadian Arctic Archipelago are less "easy" and I would have hoped for a notion how good or bad ERA-Interim data might be in these, topographically more complex regions. This applies particularly to temperature and wind speed.

This continues with not picking the potential latent heat effects of some of the polynyas (e.g. NOW) in the discussion of the results; ice production values could biased positive when oceanic heat fluxes are neglected.

And this finalizes in a, to me, not satisfying demonstration that the cloud gap filtering approach is indeed resulting in physically realistic results plus a lack of the potential uncertainty of this approach. I have the feeling that the approach as presented here potentially misses short-lived (1-2 days) polynya closing or opening events coinciding with the passage of low-pressure systems (which are ususaly associated with changing wind directions and clouds). While this might not change the average polynya area it might have an impact on the overall ice production and in the variability of both, polynya area and ice production. I would have appreciated either an analysis which demonstrates that biases due to missed polynya closing or opening events are unlikely to occur, or a theoretical analysis which estimates the uncertainty in polynya area and hence ice production due to such cases.

3) Not clear to me (and this refers again to comment 2) is how the metrics used in Table 1 (COV2 and COV4) works and why a fraction < 0.5 seems to be "bad" and why it seems to be "good" to have a polynya fraction close to 1. I am sure this is simply based by a misunderstanding and that reformulating sentences will clarify this issue.

4) The authors could clarify better that an observed increase in polynya area and/or ice

production for the period November through March over the winters 2002/03 through 2014/15 could have one main reason: a later freeze-up. It seems as if parts of the regular fall freeze-up are included in the analysis of the authors. And since the fall freeze-up has the tendency to occur later and later it impacts the derived polynya area and associated ice production. Currently I don't see that the authors make an effort to discriminate between regular fall freeze-up and a "real" polynya event - which one could consider as a methodological hic-up. It would be, however, difficult to find a definition between the end of fall freeze-up and the beginning of the "regular" winter-time polynya-opening.

5) Into the same direction as 4) goes my final general comment. While the authors state in Figure 4 that they excluded the marginal ice zone facing the Nordic Seas I could not find a notion how this was done. The marginal ice zone could overlap with NOW, with the polynya regions facing the Bering Sea, and with SZN, KAR, FJL, and SVA and I am wondering how the authors separated events where the marginal ice zone extented into these regions from "real" polynya events.

Specific comments: I note that some of the specific comments might read as a repetition of my general comments. This is caused by the fact that I usually first go for the specific comments and afterwards decide which I rate as a general and/or major comment without deleting the specific comment. Often there are more details given in the latter as well anyways.

Abstract: Page 1, line 4: I suggest to add "MODIS" in front of "swath-data". Line 7: Acronym "POLA" is not further used in the abstract and can therefore be deleted. It needs to be introduced for the main body of the manuscript anyways. Line 13: Because the manuscript focuses on polynyas I suggest to re-formulate "thin-ice features such as large leads" into "polynyas and also large leads"

Introduction: Page 2, line 2: Why "large". I would have considered polynyas and leads as small open water and thin ice areas - at least small compared to the entire Arctic

[Figure]

Ocean. Perhaps "Areas of open water and thin ice, i.e. polynyas and leads, are ..." would also be an appropriate formulation?

Page 3, line 1: I agree with the authors that wind-induced stress is the main driver for most polynyas and also leads. I am wondering, however, whether the authors might also want to comment on tidal currents, which could play a role for essentially all polynyas on the shelf. In addition, entrainment and/or upwelling of warmer / saltier water masses from below or from riverine input (here just warmer and not saltier of course) could also play a role in keeping open polynyas and/or leads, and in supporting their formation. Since the authors are after sea-ice thickness retrieval using the heat-flux method and are focusing on thermodynamic sea-ice growth assuming that oceanic heat fluxes are neglected it might be worth to at least mention that this assumption could be violated (partly) for those polynyas which are not solely a latent heat polynya but which have a substantial sensible heat polynya component.

Line32/33: What about information about meteorological parameters and heat transfer coefficients. Aren't particularly the latter quite variable and isn't it challenging to apply the correct coefficient for the different thin ice areas encountered in this manuscript? Also, I would have thought that a correct surface-to-near surface air temperature and moisture gradient as well as the correct near surface wind speed need to be known as detailed as possible. Perhaps the authors could either explain in the manuscript why these are not important or, if in fact these are, also add these here.

Data: Page 5, line 6: Is MOD35 also used for MYD29 or does a separate cloud mask exist (and is applied) for MODIS aboard AQUA?

Line 9: Could the authors perhaps motivate the grid-cell size chosen? As this to do with the decrease in spatial resolution of the MODIS pixels towards off-nadir?

Line 19: Please check whether you have introduced the acronym "TIT" in the text already. So far I only see it in the caption of Figure 2.

Line 26: Please note the average and maximum time difference between MODIS swath data and ERA-Interim data.

Methodology: Page 6, line 7: I encourage the authors to add a statement about the ice type which their method is able to derive the thickness for. Is is frazil / grease ice or are we talking about nilas and thicker sheet ice types like grey ice?

Line 14/15: I understand that the authors mention March here as this month contains the spring equinox. However, November is almost as close to the winter solstice as February is. Could it be that in November the cloud coverage is the problem?

Page 7, Figure 3: In the case shown there were good TIT maps on January 14 and 16 (i.e. from 2 days of the surrounding 6 days used), i.e. directly adjacent in time to the TIT map from which the MCC filtering removed artificial but also correct TIT areas. I am assuming that this is a very good example. How often did the authors not find appropriate adjacent TIT maps? Caption, lines 7/8. I am not sure that Spreen et al. (2008) is the only reference you should use here because that paper is adressing AMSR-E while the data you used stem from AMSR2. Hasn't there been a paper by Beitsch et al., Remote Sensing, 2014, about applying the ASI algorithm to AMSR2 89 GHz data for sea ice concentration retrieval? The same comment applied to page 8, lines 15/16.

Page 8, line 11: I have difficulties to understand Table 1 and the statement of "with certain regions performing better ... and some other regons noticable worse" If I understood the COV2 and COV4 correctly, then this is giving the fraction of the predefined area (Figure 1) covered by thin ice as retrieved by the authors's method. What seems strange to me is that some of these show a COV4 close to 1, which would mean that the entire predefined area is covered with thin ice. I doubt that KAR is really covered to 95% by thin ice. Possibly I did misunderstand something here. I encourage the authors to clarify this issue and to better explain what their metrics is to decide which is "better" or "worse".

Line 21-23: I suggest that the authors refer more to their own earlier results (Brunt ice shelf, etc.) because I find it a bit dangerous to conclude that the correction works fine from just one example shown here.

Page 8, line 32 through page 9, line 5: This discussion about the correct sea ice salinity comes back to my previous comment about which ice type the approach can consider. I guess it is worth mentioning whether the approach primarily retrieves TIT in the frazil / grease ice domain until that area where this "unstable" ice starts to collect at the leeward side of the lead/polynya to form nilas and subsequently thicker ice types, or whether the approach primarily considers the nilas and thicker sheet ice types. Actually, if it would be frazil ice, the sea ice salinity might have chosen to be larger; studies focussing on frazil ice use salinities of 917 kg/mˆ3 (delaRosa and Maus, The Cryosphere, 2012) or 920 kg/mˆ3 (Jordan et al., Journal of Physical Oceanography, 2015).

Page 9, Table 1: The "plus/minus" values in the column TIT are one standard deviation over all winters considered. How about the respective values in columns COV2 and COV4?

Page 9, lines 8-10: "We do not consider an ocean heat flux ..." I agree with the authors that this would complicate the TIT retrieval substantially. I am curious, however, whether your discussion of uncertainties will reflect that fact that some areas might have substantial oceanic heat fluxes. The authors might want to consider one further reference in this respect: Yao and Tang, The formation and maintenance of the North Water polynya, Atmosphere-Ocean, 41(3), 2003, and also cite Melling et al., 2015 here.

Page 10, line 5: I guess the authors wanted to refer to either "optical and infrared" or even only "infrared" instead of "optical" here.

Line 10: "falls below 0.5" I have difficulties to understand the authors' concept of using the fraction of the predefined polynya regions shown in Figure 1 as a quality measure. I commented on that already in the context of table 1. Here, the authors limit the

fraction of thin ice in these predefined areas to be above 0.5 - if I have understood this correctly. Or, in other words, it reads as if a thin ice fraction of the predefined polynya regions in Figure 1 needs to be above 0.5, otherwise it is regarded faulty. I probably misunderstood something?

Results and Discussions:

Page 10, line 16/17: The trend in TIT mentioned in these lines are not summarized in any of the tables, am I correct? Perhaps the authors could spend a "(not shown)" or something?

Lines 29-33: I suggest the authors cite work which is related to the derivation of fast-ice extent in, e.g. the Laptev Sea like for instance: Selyuzhenok et al., J. Geophys. Res., 2015.

Page 12, Table 2: I am wondering whether the trends given are "per year" as indicated or "per decade"? If these are indeed per year, then in region ESF the increase in POLA would be 1.095 km^2 in 10 years which equals the average POLA value given. The same applies to region SZN. Perhaps the authors could check which reference period they used for their trend calculations? The authors might also consider to write how the p-values were derived, i.e. which statistical test was carried out.

Page 12, line 11: Stylistically I would say "the large POLA values" is enough here (instead of "these") because the authors refer to NOW in the remainder of the sentence. I note in this context, that the increase in NOW POLA is not significant in the authors' study.

Page 15, lines 1-8: The authors inter-compare their results with Kern (2008), who only focused in the Kara Sea. Aren't there other studies about polynya area which results would be worth to compare the authors' results with?

Lines 8/9: "increases for" Do the authors refer to an increase in POLA or to an increase in POLA variability?

Lines 9-20: I absolutely agree with the authors' observations written down in this part. The only concern I have here is: Where do the authors differentiate between IP during regular fall freeze-up and IP within polynyas and leads. Or in other words, when do the authors define an open water / thin ice areas to be belonging to a polynya and when is this still considered fall freeze-up? In this context: in the caption of Figure 4 the authors make a note that they discarded the regions of high TIT frequency along the marginal ice zones facing the Nordic Seas from further analysis. Wouldn't it make sense to do the same for the northern Baffin Bay (in November) and also the southern Chukchi and Beaufort Seas (in November)? Also: What was the criterion to exclude areas with a high TIT frequency. I could not find a notion how exactly these regions were defined. Did the authors used a TIT frequency threshold?

Line 23: "slight decrease" I suggest the authors add that these decreases are far from being significant.

Line 26: "plus/minus 258 kmˆ3" Is this an uncertainty, or is this the standard deviation from computing the average IP of the 13 winters?

Line 30-31: I suggest that the authors comment more on this comparison. Tamura and Ohshima's results are based on SSM/I data while Iwamoto et al. base their study on AMSR-E data. The authors' study is based on MODIS data. This implies different spatial resolutions which effect on the results could be discussed here. Actually, in the next paragraph starting in line 32 the authors carry out this discussion but without linking it to the statement in lines 30-31 and without trying to investigate (and discuss theoretically) whether 2 km instead of 6.25 km grid resolution would allow to explain the larger IP found in this study compared to Iwamoto et al. Yes, I agree, with a finer grid resolution one is able to identify smaller scale thin ice features. There is no doubt about that and this has been demonstrated in previous papers of the leading author. But at the same time POLA of larger polynyas could become smaller because the polynya edge is better defined at 2 km than at 6.25 km. Therefore there could be competing effects with the net effect being zero. In addition the period of Iwamoto et al. is much

closer to the one used by the authors. By looking at the winters 2011/12 through 2014/15 the authors could check whether their larger value compared to Iwamoto et al. could be explained by considerably larger IP in these winters compared to the winters before 2011/12.

Page 17, Figure 8: I am wondering why the map showing the significance is smaller than the one showing the trends. I suggest to make both maps the same size or, alternatively, to overplot significance levels on an even enlargened version of image a) using, e.g. dots and crosses to denote areas of >95 and >99% significance or isolines. However, what is a bit unfortunate here - as well as already in Figures 4 and 7 is the fact, that the marginal ice zone (MIZ)facing the Nordic Seas is visually dominating the Figure and distracts the eye from those regions which are really relevant for the present study. In the context of the yet unexplained way how these MIZ areas are excluded (according to the caption of Figure 4), I encourage the authors to find a way to make these areas to appear less prominent, perhaps by grey shading or similar, so that the reader can focus on the relevant areas.

Line 2: "diminishing fast ice extent over the recent 13 years." I am not sure that the extent of the fast ice can be mentioned as the reason here - at least not solely. I recommend that the authors take a look at the paper by Selyuzhenok et al., Seasonal and interannual variability of fast ice extent in the southeastern Laptev Sea between 1999 and 2013, J. Geophys. Res.-Oceans, 2015 and of Yu, Y., et al., Interannual variability of Arctic landfast ice between 1976 and 2007, J. Climate, 2014 to underline or perhaps change their statement here.

Page 18, line 11: I suggest the authors cite the two other studies at the end of this sentence (i.e. Tamura and Ohshima 2011, and Iwamoto et al. 2014) I further suggest that the authors clarify that by "more accurate" they solely refer to the spatial accuracy and not to an accuracy of the TIT and IP computation approaches. Perhaps this could be done by replacing "and for more accurate" with "and therefore spatially more accurate"?

Page 18, Figure 9: I have a late comment to the choice of the regions LAP and SZN. I am wondering why these two regions were defined as they are. Why does the western part of region LAP extends well into the Severnaya Zemlja area and with that well beyond the shelf break? Wouldn't it be more consistend to let region LAP and shortly south of the Vilkitsky Strait?

Page 19, Figure 10: What is the motivation to interpolate / smooth the POLA in this figure? Wouldn't similar conclusions be reached by simply showing the daily POLA as is?

Lines 9/10: "largest POLA values appear on average in November and ..." Is this perhaps still fall freeze-up?

Lines 12/13: "polynya activity" Are the authors referring to the sheer occurrence of a polynya or to the POLA? If the authors talk about the former then one could conclude that the activity is as large today as it was in the past. The main difference is that the POLA tends to be larger recently.

Line 18: "position of the fast-ice edge": I suggest the authors include a note that Figure 11 of course integrates over the full winter season from November to March. That is, periods of polynya activity exchange with quiet periods during which the fast ice potentially extends northwards. This is just to avoid a readers' conclusion that the fast ice breaks up; the fact that there can be bands of higher ice production within the area which should be fast-ice covered can also (if not merely) be associated with the episodic nature of fast ice development, particularly during early winter.

Page 19, line 19 until page 21, line 3: Did the authors check whether winters with a characteristic "ice arc" feature can be related to years where the sea ice did not melt completely in that region of the Kara Sea? Also: While in the Nares Strait the dominant wind direction and hence formation of the ice arc is clear, how is this in the Vilkitzky Strait?

Page 21, lines 8-12: While one could have a look at the paper by Krumpen et al. (2013) the authors could also, in one of their images in Figure 11, draw a line which marks the gate across which the IAE is computed.

Line 12: "significant" With which p-value?

Lines 16-19 and page 22, lines 10-14: While I am not doubting that the IP of the LAP has indeed increased for November-March for 2002/03 through 2014/15 I am wondering whether the authors could also include a critical comment of these numbers and take into account that freeze-up has been commencing later recently over many regions of the Arctic Ocean (Markus et al., Recent changes in Arctic sea ice melt onset, freezeup, and melt season length, J. Geophys. Res., 2009; Parkinson, C., Spatially mapped reductions in the length of the Arctic sea ice season, Geophys. Res. Lett., 2014) and that this could be the main driver for the increase in IP observed in the present study - in addition to a thinner, more easily to be deformed and pushed away by offshore winds sea ice cover. Yes, the authors mention the "length of the freezing period", among other reasons, but remain not conclusive enough to my taste. In particular, it is not the length of the freezing period but the onset of freeze-up. Unmentioned remains also a potential air-temperature increase particularly during winter which would counterbalance an increase in IP during November-March.

Conclusions: Page 22, line 24: "and the sea-ice budget in general". I suggest that the authors remain more specific here and write: "and the associated sea-ice budget related to winter-time sea-ice production." Even though the polynyas for sure make a substantial contribution to the Arctic Ocean sea ice budget which is certainly mainly determined via the annual freeze-up and ice thickening underneath existing sea ice due to congelation growth.

Page 23, lines 3-4: I suggest to here only mention those negative trends which are significant. Hence one could end the sentence after "... variability."

Lines 4-6: What the authors write here could be true but certainly deserves more work

to be done. Most importantly, however, this is not a result the authors achieved and I recommend that the authors stay with their own results in the conclusion bullets before they eventually give an idea about what they think could be a possible reason for the changes observed in their data set.

Lines 13-14: Is the paper by Boisvert and Stroeve, 2015, focusing on the Laptev Sea? I cross-read the paper and had difficulties to find evidence for the link presented here. Yes, air- and skin-temperatures seem to have a positive trend in the LAP - especially in October and November but no further information about the winter is given. Increasing temperatures at first glance point to a decrease in IP, though. It is important that the authors clearly state how the causal links are and not only list a number of potentially relevant papers. The same applies to the "significant lengthening of the melt season". I suggest to be more specific here as well, because the melt-onset is not important here but the commence of freeze-up.

---

## Referee Comment (RC2) · G. Björk (Referee) · 23 Aug 2016

General comments

This work gives a comprehensive description of Arctic polynyas based on high resolution surface temperature (MODIS) data. This data set has the advantage of a higher spatial resolution compared with satellite products used in earlier investigations. It gives a 22 % higher total polynya ice production than recent results which shows that the development of satellite products, algorithms and analyses is still an important issue in order to follow the past and future development of the Arctic Ocean ice cover.

[Figure]

It reveals significant positive trends of the polynya ice production in the eastern arctic which can be further utilized for analyses of the effect on dense water mass formation on the shelves which likely have influence on shelf circulation, shelf basin interaction and water chemistry. The paper is generally of a high quality in language and analyses and is therefore well suited for publication.

Specific comments

I'm not perfectly happy with the sentence starting on line 6 page 3 ( "A regular monitoring. . .").  It appears to be somewhat a repetition of the sentence on page 5 line 8 ( "Hence an accurate. . .".

Page 9 Table1 text.  I miss some more explanation of what the "interannual average coverage" means. Coverage of what? It is hard to understand as it stands now.

Page 9 Line 2 and before.  It is hard to follow the logic why fixed values for ice and Lf are used. The arguments regarding frazil ice crystals are not clear to me.

Page 12 Line 2. It is interesting to see persistent leads well off the shelf in the Beaufort Sea.  These must be related to the large scale ice circulation in the area and it is remarkable that they are so persistent that the show up as well defined bands in this type of data (most notable in Feb-Mar). I wonder if this structure has been described before or if it is a new finding. It is worth some more comment anyhow.

Page 12 Line 3. I can't see the leads along the Transpolar drift in figure 4. The central area around the North Pole appears to be without leads in the figure.

Page 15 Line 8. Suggestion: "is especially large " instead of "increases".

Page 19 Line 13. Sentence starting with "A pronounced seasonal. . ." is unclear. I can see that the seasonal variation is largest in the late half of the period, but the last part is confusing.

Page 19 Line 18. I think the reader needs some more help to identify the fast ice edge

in figure 11 and also in earlier figures. It is not clear to me since there are several bands of high ice production from the coast and outward in most of the fields.

P21 Line 9. IAE need to be defined better. Is it the export just outside the Laptev Sea or what?

Technical corrections None

---

## Referee Comment (RC3) · Anonymous Referee #3 · 24 Aug 2016

General comments

Coastal polynyas play a crucial role in altering a variety of physical, biological and chemical processes at the boundary between the atmosphere and the ocean. In the case of Arctic Ocean, polynya ice production is a key component for understanding the maintenance and variability in ocean stratification (cold halocline) and ice-ocean interaction, as well as the seasonal sea-ice mass budget. This paper provides the circumpolar mapping of polynya area (POLA) and its ice production (IP) in the Arctic Ocean, with fine spatial resolution of about 2 km. This resolution is much finer than

the previous mapping with satellite microwaves. The authors have accomplished the creation of the dataset of POLA and IP by treating massive amount of 143000 MODIS data, with well-refined procedures. As well, the paper provides 14-year dataset of POLA and IP, which will be the basic data for understanding of drastically changing Arctic Ocean. The paper is overall logical, well-organized, and the presentation/writing is refined. Although the results might have some bias arising from that the calculations were made only for clear-sky and nighttime conditions, this is mainly because of limitation of satellite (MODIS) data. I think that the authors have done a best to create the circumpolar data set with a high spatial resolution. I believe that the paper surely contributes to the community of Arctic and climate sciences. But there still remains some points that should be improved, all of which are minor ones. Some figures can be a bit improved for clarity (see comments 7, 8, 12, 14, 15 for details). In brief, the paper should be published in Cryosphere after a minor revision. The specific points are the followings.

Specific comments

1. MODIS clear-sky data can be obtained rarely in the polar cloudy condition. Thus most of researchers including me think that it is difficult to obtain seamless (daily) surface dataset from the MODIS data. For example, in investigation of landfast ice (Fraser et al., 2012, J. Climate) from MODIS, data set was made only for 20-day interval because of cloudy condition. At first I could not believe the average coverage fraction of 70-80% per day (Table 1) in this study. However, if the MODIS image can be obtained for one area several tens of times per day, composite of clear-sky portion could offer the daily data. I guess this is the case and explain why such high fraction of coverage is possible. If this is true, the authors should clearly explain why such high fraction of coverage is possible. For example. How many times per day can MODIS cover a certain area? What percentage can we obtain the cloud free scene? I think that such explanation enhances the creditability of this study.

2. Although the MODIS data provide high resolution data set, POLA and IP can be

obtained only in clear-sky condition. The atmospheric condition and accordingly sur-face heat flux in clear-sky condition would be different from those in cloudy condition. Thus it is likely that POLA and IP obtained all from clear-sky condition have some bias compared to those from cloudy condition or pure average irrespective of atmospheric condition. I understand that evaluation of such bias is not easy and no further analysis is needed. But more discussion or clear statement of such bias should be made in the revision. At least such drawback should be stated in conclusion section.

3. Similarly, POLA and IP can be obtained only in nighttime and thus POLA and IP obtained all from nighttime likely have some bias compared to those from daytime or pure average. Although a brief statement was made in page 9, it may be better to evaluate such bias even in a brief way. For example, difference in heat budget on thin ice for nighttime and daytime under a typical wintertime condition can be evaluated.

4. The study does not include the results of October and April when the polynya ac-tivity starts and continues, which is one of the drawback of this study. According to Iwamoto et al. (2014), for example, these two months provide 10-30 % of total annual ice production (IP). Particularly in NEW, Laptev, Archipelago, IP becomes maximum in October. Such drawback should be stated in IP section and conclusion.

5. Abstract: "Overall, our study contains the most accurate characterization of circum-polar polynya dynamics and ice production to date". This statement is ambiguous and overvaluing. The authors should state more specifically in what points this study pro-vides the most accurate characterization? Probably, high spatial resolution is strong selling point. On the other hand, this study still has the drawback of data gap by cloud.

6. P3, L15-16: "west of Novaya Zemlya is excluded in our investigations due to a variety of potential ambiguities originating from ocean heat fluxes": I understand the situation. But, as described in the textbook by Martin (2001, Polynyas. In: Encyclopedia of Ocean Sciences. vol 3. Academic Press,), the Novaya Zemlya polynya is one of the most active polynya, and other studies (e.g., Iwamoto et al., 2014) includes the Novaya

Zemlya polynya in their tables. Similar situation by the effect of ocean heat also occur in the polynyas of Storfjorden, Franz-Josef Land. Why only the Novaya Zemlya polynya is excluded?

7. In some figures (Figs. 4, 7, and 8), coast lines are not visible.

8. P6, Figure 3: The scale in the right bottom should be enlarged.

9. P7, L14-20: I understand the former part of this paragraph, but I do not understand well that why a pixel-wise persistence index (PIX) becomes the ratio between the total number of MODIS swaths that feature thin-ice and the total number of swaths that feature clear-sky conditions.

10. P7, L25-28: "a probability of thin-ice occurrence is derived using a weighted composite of the surrounding days", "by a weighted average of the surrounding six days". Please describe the weight (function) specifically.

11. P9, L19-24: This paragraph is hard to understand. How does the coverage-correction (CC) carry out the extrapolation? What's mean by "the additional SFR areas (COV4)".

12. P15, Figure 7: Most of the IP area are colored by nearly same color, blue, implying the production of 0.8-2.5 m. These high frequency ranges should be better resolved using stronger color gradient to discriminate the difference.

13. P17, L7-9: "a tendency towards a diminishing fast-ice extent", "retreating-behavior of fast ice": Which part corresponds to these features? It is better to describe the location of these areas specifically.

14. P18, Figure 9: The inset map at the upper right is not effective at all. Rather, range of Fig. 9 (Laptev Sea area) should be indicated in Fig.1 with the name of Island.

15. P19, Figure 10: This is not usual Hovmoeller plot. Hovmoeller plot is generally used to see the spatial vs. temporal variations to examine the propagation characteristics. There is no temporal continuation in the vertical direction of Fig.10 and thus the contours should not be used in the vertical direction. This figure should not be drawn as Hovmoeller plot. The contours can be used in the lateral direction (seasonal evolution). I propose the following figure. The polynya area ratio (strength) is represented by the contours and color in the lateral direction (seasonal evolution) separately for each year. Then such horizontal long graphs are lined up in order from 2002 to 2015. Namely the contour procedure is only used for the lateral direction when compared to the original Fig.10.
* * *

---

## Author Comment (AC1) · 11 Oct 2016

**Received from Dr. Stefan Kern (referee) on August 22, 2016**

**Summary:**

The great potential of the combined ice surface temperature (IST) data sets derived from TERRA and AQUA MODIS infrared surface temperature observations is utilized to derive a pan-Arctic view of polynya area with unprecedented spatial grid resolution for such a long period of winters (Nov.-Mar.) spanning 2002/03 through 2014/15. Polynya area is derived by means of combining the IST with meteorological information provided by ERA Interim re-analysis data to estimate thin ice thickness (TIT). To overcome gaps due to cloud coverage an innovative, recently published approach is further developed and applied to the derived time-series of quasi-daily TIT maps. The final results: time series of distributions of polynya area, TIT and ice production are presented and discussed. The average polynya area and ice production are within the range of previous studies. Polynyas in the Eastern Arctic are found to have an increase in ice production for Nov.-Mar. over the time period considered.

**General comments:**

**0)** The paper is very well written and it reads fluently. The figures are mostly excellent. The paper presents the retrieval and discussion of a polynya area and ice production data set of yet unprecedented spatial resolution and hence for sure warrants publication. In the current version of the manuscript a few critical definitions and questions remain unanswered, though, which I feel are required to not misinterpret this very nicely written article. The discussion of potential uncertainties and biases in the retrieved data should be improved for the same reason. Finally, the inter-comparison to other studies and discussion of the differences to other studies by means of the material the authors already have in hands could be improved.

We would like to thank Dr. Stefan Kern (referee #1) for his valuable comments and suggestions that will definitively help to improve the original manuscript, most importantly the discussion of results and the specification of error-margins. We carefully went over the mentioned parts of the manuscript and we will answer remaining general as well as specific comments in the following.

**1)** The abstract and conclusion write: "most accurate characterization of ..." I would rate it as important that the authors clearly state that they speak about spatial accuracy and not about retrieval accuracy of the thin ice thickness and ice production. In addition to that, as I write further down (in the context of the discussion with results about the polynya area from other authors), the authors could elaborate on the question whether the net effect of a finer grid resolution is solely an increase in the derived total polynya area, or whether the reduced smearing / smoothing for larger size thin ice areas when using MODIS data doesn't mean that derived polynya sizes could be also smaller.

You are right about our formulations in the abstract and conclusions. To be more specific at these textpositions, we changed the mentioned parts according to:

"Abstract: Overall, our study **presents a spatially highly accurate characterization** of circumpolar polynya dynamics and ice production which should be valuable for future modeling efforts on atmosphere- sea ice - ocean interactions in the Arctic."

and

"Conclusions: (...) we think that this new data set of 13 consecutive winter seasons is a huge step forward for a **spatially** accurate characterization of Arctic polynya dynamics and the seasonal sea-ice budget in general."

Regarding the other remark, the net effect of the finer grid resolution, i.e. the sign of a possible bias, is really difficult to assess without actual reference / comparison data at hand. Certainly, a more precise delineation of larger polynya areas could also lead to the opposite effect regarding POLA and hence IP differences, but these effects can only be evaluated by looking at the distribution of thin-ice (and consequently heat fluxes) within the larger footprint of the passive microwave data sets.

2) Tied with accuracy is that, to my feeling, the retrieval accuracy of the method is discussed not enough. The only notion I found about the accuracy of the thin ice thickness retrieval is the one cited by Adams et al. (2013). It does not seem that the authors did carry out accuracy investigations on their own. This starts with the validity of using coarse resolution ERA-Interim data (coarse compared to MODIS) in a pan-Artic sense. Yes, for the Laptev Sea investigations published in the literature have shown that re-analysis data fit observations quite nicely, but this is an "easy" area in terms of topography. Areas around Greenland (NEW, NOW) and the Canadian Arctic Archipelago are less "easy" and I would have hoped for a notion how good or bad ERA-Interim data might be in these, topographically more complex regions. This applies particularly to temperature and wind speed. This continues with not picking the potential latent heat effects of some of the polynyas (e.g. NOW) in the discussion of the results; ice production values could biased positive when oceanic heat fluxes are neglected. And this finalizes in a, to me, not satisfying demonstration that the cloud gap filtering approach is indeed resulting in physically realistic results plus a lack of the potential uncertainty of this approach. I have the feeling that the approach as presented here potentially misses short-lived (1-2 days) polynya closing or opening events coinciding with the passage of low-pressure systems (which are usually associated with changing wind directions and clouds). While this might not change the average polynya area it might have an impact on the overall ice production and in the variability of both, polynya area and ice production. I would have appreciated either an analysis which demonstrates that biases due to missed polynya closing or opening events are unlikely to occur, or a theoretical analysis which estimates the uncertainty in polynya area and hence ice production due to such cases.

(1) Regarding retrieval accuracy: As the specific procedure to derive TIT (compare Sect. 3.1) has not changed significantly (besides the use of ERA-Interim instead of NCEP2 reanalysis data), we regard the accuracy assessment by Adams et al. (2013) as a valid characterization of uncertainty ranges.

(2) Regarding ERA-Interim: We appreciate the remark on this topic. In order to address this, we added the following information to Sect. 3.3:

"For topographically complex regions like Greenland and Arctic fjords, recent studies revealed shortcomings of the coarse-resolution ERA-Interim data regarding the representation of mesoscale spatial features in the wind field, such as tip-jets, channeling effects or other topography-induced phenomena related to locally increased wind speeds (e.g. Moore et al., 20161). Thus, ERA-Interim shows a tendency to underestimate peak wind speeds (Moore et al., 2016) which might in some cases induce a negative bias (lower heat fluxes/ less IP) in regions where polynya formation is strongly influenced by the local topography (e.g. CAA, NOW, NEW, SZN). In our study, the usage of ERA-Interim is motivated by ensuring comparability to similar studies (e.g. lwamoto et al., 2014) as well as the

<sup>1 Moore, G.W.K., Bromwich, D.H., Wilson, A.B., Renfrew, I., Bai, L. (2016): Arctic System Reanalysis improvements in topographically-forced winds near Greenland. Quarterly Journal of the Royal Meteorological Society, doi:10.1002/qj.2798.

constraint that higher-resolution atmospheric data sets such as the Arctic System Reanalysis (ASRv1 – 30km; Bromwich et al., 20152) are not available for the complete time period from 2002 to 2015."

Hence, we aim to investigate the potential for a future application of high-resolution (~15km) regional reanalysis /climate models such as the ASRv2 (Bromwich et al., 2015) or COSMO-CLM (Gutjahr et al., 2016) in the here presented TIT retrieval once they become available.

**(3) Regarding oceanic heat fluxes:**

After performing a rough estimation of the effect of an oceanic heat flux (similar to Tamura and Ohshima (2011) and Iwamoto et al. (2014)), we see that the ice production in the North Water polynya (Avg. 276.7 km3) could be reduced by around 22.5% when assuming a constant heat supply from the ocean of 50 W/m2 (Bourke and Paquette [1991] and Darby et al. [1994]). This is approximately the same range as in both referred Japanese studies. However, the effect of oceanic heat on wintertime thin-ice dynamics in the Arctic is to date still not very well documented / understood and obviously a subject of recent scientific discussions (see some quotes below). For instance, the study by Yao and Tang (2003) concluded that, in case of the NOW polynya, the ocean heat flux does not reduce the ice growth rate even though there is evidence of convective mixing and entrainment by ice growth, which might trigger enhanced ocean heat fluxes in northern Baffin Bay.

**Yao and Tang (2003):**

"Salt flux from ice growth is balanced by advection, from which we infer that the **exchange is predominantly horizontal** and not coastal upwelling. It appears that atmospheric heat flux compensates so that the **ocean heat flux does not reduce the ice growth rate.**"

Carmack et al. (2015)3:

"In autumn and winter, ocean sensible heat is transported to the air—ocean and air—ice interfaces by upper-ocean mixing and by conduction through the ice; however, **measurements from recent years show that some of the heat gained by the upper ocean in summer is stored into the winter and can slow the growth of sea ice** (e.g., Jackson et al. 2010, 2012)."

"Through most of the Arctic Ocean, however, **heat input as AW and PW is separated from the surface by a layer of relatively cold and fresh water that reduces the direct impact of these heat sources on sea ice.** One notable exception is the Nansen basin where, near the Fram Strait gateway, near-surface AW heat results in a significant reduction in sea ice thickness along the continental slope north and northeast of Svalbard (Onarheim et al. 2014)."

"However, analyses of ITP records from the central Eurasian basin, away from steep topography, suggest that the delivery of AW heat to the overlying layers in the Eurasian basin interior can be important (Polyakov et al. 2013). Those authors showed that **the transfer of heat from the upper pycnocline to the SML is highest in winter, with an average heat loss of 3–4 W/m2 between January and April.** It is likely that the increased heat loss from the AW layer to the SML in winter is caused by a combination of brine-driven convection that is associated with sea ice formation and larger vertical velocity shear below the base of the SML that is enhanced by winter storms."

(4) Regarding potentially missed short-lived events and SFR uncertainties: The SFR has only to do with the availability of MODIS coverage and is even most effective on short time-scales (Paul et al. 2015a). Polynyas typically appear on time ranges between 1-3 days (high autocorrelation).

<sup>2 Bromwich, D.H., Wilson, A.B., Bai, L.-S., Moore, G.W.K., Bauer, P. (2015): A comparison of the regional Arctic System Reanalysis and the global ERA-Interim Reanalysis for the Arctic. Q. J. R. Meteorol. Soc. 142: 644–658.

<sup>3 Carmack, E.; Polyakov, I.; Padman, L.; Fer, I.; Hunke, E.; Hutchings, J.; Jackson, J.; Kelley, D.; Kwok, R.; Layton, C.; Melling, H.; Perovich, D.; Persson, O.; Ruddick, B.; Timmermans, M.-L.; Toole, J.; Ross, T.; Vavrus, S. and Winsor, P. (2015): Toward Quantifying the Increasing Role of Oceanic Heat in Sea Ice Loss in the New Arctic Bull. Amer. Meteor. Soc., American Meteorological Society, 2015, 96, 2079-2105.

The fraction of days where the use of SFR fails to achieve an IST/TIT coverage > 0.5 is overall very low (less than 2% of all the days in 2002/03-2014/15; except CHU  $\rightarrow$  ~13%). Hence, the probability to miss (or overestimate POLA) short-lived events is generally rather small, but may be higher for more frequently cloud-covered regions such as Chukchi Sea (CHU).

Figure 1 Overview on the interannual (2002/2003 to 2014/2015; Nov.-Mar.) fraction of exclusively interpolated days (POLA/IP values), i.e. with the best possible daily MODIS coverage (COV4) not exceeding 0.5 (50% spatial coverage). Values are given per region. The absolute amount of days is additionally listed in turquoise numbers.

**3)** Not clear to me (and this refers again to comment 2) is how the metrics used in Table 1 (COV2 and COV4) works and why a fraction < 0.5 seems to be "bad" and why it seems to be "good" to have a polynya fraction close to 1. I am sure this is simply based by a misunderstanding and that reformulating sentences will clarify this issue.

You're right in your assumption of a misunderstanding, as we are not writing about polynya fractions. COV2 and COV4 are metrics that refer to the spatial coverage of MODIS data, i.e. the availability of valid (clear-sky, HQ MCP, SFR) IST/TIT value-pairs inside a respective polynya mask area.

We will reformulate and clarify the respective parts in the manuscript, e.g. P.8 L10; Caption Tab.1; ...

**(...) see P.8 L.9.: "Table 1 gives an overview on the **achieved MODIS coverage** before and after application of the SFR algorithm. On a pan-Arctic level, the average (...)"**

**4)** The authors could clarify better that an observed increase in polynya area and/or ice production for the period November through March over the winters 2002/03 through 2014/15 could have one main reason: a later freeze-up. It seems as if parts of the regular fall freeze-up are included in the analysis of the authors. And since the fall freeze-up has the tendency to occur later and later it impacts the derived polynya area and associated ice production. Currently I don't see that the authors make an effort to discriminate between regular fall freeze-up and a "real" polynya event - which one could consider as a methodological hic-up. It would be, however, difficult to find a definition between the end of fall freeze-up and the beginning of the "regular" wintertime polynya-opening.

This is actually one of the critical points when analyzing wintertime polynya dynamics, you are right. But as you also mention, separating between fall-freeze-up and regular polynya events is quite challenging for a number of reasons, especially on such a large scale as the timing varies significantly for each region in the Arctic. However, we think that for many investigated areas throughout the Arctic the complete period between Nov. and Mar. is highly interesting as potentially occurring larger heat fluxes in early winter strongly alter the atmospheric and oceanic boundary layers regardless of fulfilled textbook definitions of a polynya. Hence, we decided to use a fixed reference frame in order to ensure comparability between different regions and winter seasons, as well as to present additional and separately derived values for the period JFM, as can be seen in Fig.5 and 6 (seasonal comparisons ND vs. JFM). You are right however that we could have done a better job in referring to the influence of the freeze-up in certain regions such as STO, CAA, KAR, CHU and FJL. Therefore, in the revised version of manuscript we tried to emphasize this topic more clearly. In addition, we overhauled the former Table 2 to clearly show seasonal differences in derived average values and trends (now Tab.2+3).

**5)** Into the same direction as 4) goes my final general comment. While the authors state in Figure 4 that they excluded the marginal ice zone facing the Nordic Seas I could not find a notion how this was done. The marginal ice zone could overlap with NOW, with the polynya regions facing the Bering Sea, and with SZN, KAR, FJL, and SVA and I am wondering how the authors separated events where the marginal ice zone extended into these regions from "real" polynya events.

We apply polynya masks to exclude unlikely polynya / thin-ice locations throughout the Arctic and focus on likely and known polynya locations. The selection/definition is based on previous studies (e.g. Barber and Massom (2007)) as well as the here derived avg. TIT-frequencies between 2002 and 2015 (compare Fig.4.).

**Specific comments:**

I note that some of the specific comments might read as a repetition of my general comments. This is caused by the fact that I usually first go for the specific comments and afterwards decide which I rate as a general and/or major comment without deleting the specific comment. Often there are more details given in the latter as well anyways.

**Abstract:**

Page 1, line 4: I suggest to add "MODIS" in front of "swath-data".

**Fixed.**

*Line 7: Acronym "POLA" is not further used in the abstract and can therefore be deleted. It needs to be introduced for the main body of the manuscript anyways.*

**Deleted "POLA".**

*Line 13: Because the manuscript focuses on polynyas I suggest to re-formulate "thin-ice features such as large leads" into "polynyas and also large leads"*

We re-formulated the sentence accordingly.

**Introduction:**

Page 2, line 2: Why "large". I would have considered polynyas and leads as small open water and thin ice areas - at least small compared to the entire Arctic Ocean. Perhaps "Areas of open water and thin ice, i.e. polynyas and leads, are ..." would also be an appropriate formulation?

True, this might be irritating so early on in the manuscript as these relative size-relations are depending on the context. We re-formulated the sentence as proposed.

Page 3, line 1: I agree with the authors that wind-induced stress is the main driver for most polynyas and also leads. I am wondering, however, whether the authors might also want to comment on tidal currents, which could play a role for essentially all polynyas on the shelf. In addition, entrainment and/or upwelling of warmer / saltier water masses from below or from riverine input (here just warmer and not saltier of course) could also play a role in keeping open polynyas and/or leads, and in supporting their formation. Since the authors are after sea-ice thickness retrieval using the heat-flux method and

are focusing on thermodynamic sea-ice growth assuming that oceanic heat fluxes are neglected it might be worth to at least mention that this assumption could be violated (partly) for those polynyas which are not solely a latent heat polynya but which have a substantial sensible heat polynya component.

We are aware of the fact that polynyas and leads can also be influenced by tidal currents and/or oceanic heat fluxes. The studies of Hannah et al. (2009) and Melling et al. (2015) described these processes exemplary for the Lancaster and Jones Sound regions in the eastern part of the CAA (compare P.10 L.21). However, tidal-driven polynyas have time and space scales being much smaller than the polynyas listed in Tab.1-3. In Sect. 3.3 (P.9 L8-10) we already listed some studies which described areas (CHU, CAA, NOW), where an oceanic heat influence was either found/measured or assumed/suspected, and pointed to a potential reduction of thermodynamic ice growth. In order to make this part more concise, we added numbers on the potential influence of oceanic heat from the indicated studies.

Line32/33: What about information about meteorological parameters and heat transfer coefficients? Aren't particularly the latter quite variable and isn't it challenging to apply the correct coefficient for the different thin ice areas encountered in this manuscript? Also, I would have thought that a correct surface-to-near surface air temperature and moisture gradient as well as the correct near surface wind speed need to be known as detailed as possible. Perhaps the authors could either explain in the manuscript why these are not important or, if in fact these are, also add these here.

In a recently published study by Gutjahr et al.  $(2016, TCD)^4$ , we included a more detailed overview on the variance of the iteratively calculated heat transfer coefficient (CH) in the Laptev Sea region. To quote the respective section on P.21:

"Heat loss is affected by differences in the surface temperature, vertical temperature gradient, parameterization of the energy balance components, sea-ice thickness and properties, parameterization of the heat flux through the ice, and by the parameterization of atmospheric turbulent fluxes. Particularly important is the horizontal resolution of the atmospheric data set and the assumptions on the turbulent exchange coefficient for heat ( $C_H$ ). [...] The  $C_H$  values based on MODIS data and ERA-Interim are lower than simulated by CCLM with a mean of  $C_H = (2.3 \pm 0.3) \times 10^{-3}$ . A similar PDF was derived by Adams et al. (2013), who combined MODIS and NCEP."

Figure 2 Frequency-distribution (class-width 0.2 x 10-3) of iteratively calculated heat transfer coefficients ( $C_H$ ) in the Laptev Sea polynya (TIT  $\leq$  0.2m) region between November 2007 and March 2008. In this particular winter, the average value of  $C_H$  was estimated with 2.3  $\pm$  0.3 x 10-3.

<sup>4 Gutjahr, O., Heinemann, G., Preußer, A., Willmes, S., and Drüe, C.: Sensitivity of ice production estimates in Laptev Sea polynyas to the parameterization of subgrid-scale sea-ice inhomogeneities in COSMO-CLM, The Cryosphere Discuss., doi:10.5194/tc-2016-83, in review, 2016.

As the heat loss is calculated pixel-wise for each individual MODIS swath (with varying atmospheric parameters,  $C_H$ , etc.), we actually do account for differences among considered thin-ice areas.

We added some more information on this topic in the Introduction.

**Data:**

Page 5, line 6: Is MOD35 also used for MYD29 or does a separate cloud mask exist (and is applied) for MODIS aboard AQUA?

Thank you for this remark. The cloud mask is also generated for MODIS data from Aqua (MOD/MYD35). We added this to the manuscript.

*Line 9: Could the authors perhaps motivate the grid-cell size chosen? As this to do with the decrease in spatial resolution of the MODIS pixels towards off-nadir?*

Yes, we chose the grid-cell size of approx. 2km due to the decreasing spatial resolution off-nadir, resulting from panoramic distortion effects of the MODIS sensor (rotating scan-mirror; constant focal length). The study of Fraser et al. (2009)5 referred to increase-factors of 2.01 (along-track direction) and 4.93 (across-track direction) for the marginal pixels of each MODIS scan-line.

*Line 19: Please check whether you have introduced the acronym "TIT" in the text already. So far I only see it in the caption of Figure 2.*

The acronym is introduced in Sect. 2.1 (P.5 L.3).

*Line 26: Please note the average and maximum time difference between MODIS swath data and ERA-Interim data.*

The maximum time difference can be 3 hours, as we do not perform an additional temporal interpolation as in Paul et al. (2015b). Motivated by your comment, we extracted the time difference for each single MODIS swath and the respective ERA-Interim time step (00.00, 06.00, 12.00, 18.00UTC) for all years considered. The overall average time difference amounts to **89.5 ± 52.3 minutes**, which is exactly within the range of what could have been expected when assuming normally distributed MODIS swaths around each time step.

We added this information as proposed in L.26.

**Methodology:**

Page 6, line 7: I encourage the authors to add a statement about the ice type which their method is able to derive the thickness for. Is is frazil / grease ice or are we talking about nilas and thicker sheet ice types like grey ice?

There were similar remarks in previous reviews of studies from the authors (STO, Weddell Sea). Our response stays the same: The presented thin-ice algorithm does not explicitly discriminate between different ice types. It follows the assumption that a linear temperature profile can be used to calculate the heat conduction through the ice. Hence, we added this information to the manuscript. Regarding the choice of constant values for the ice density and latent heat of fusion (Lf), we followed earlier studies (e.g. Willmes et al. (2011), Tamura and Ohshima (2011), Iwamoto et al. (2014)) to ensure

<sup>5 Fraser, A. D., Massom, R. A. and Michael, K. J. (2009): A Method for Compositing Polar MODIS Satellite Images to Remove Cloud Cover for Landfast Sea-Ice Detection. *IEEE Transactions on Geoscience and Remote Sensing*, vol. 47, no. 9, pp. 3272-3282. doi: 10.1109/TGRS.2009.2019726

comparability of achieved results. These studies followed an even earlier characterization of sea-ice formation mechanisms by Martin (1981).

Section 3.1 has been complimented to now read: "(...) and the lower boundary of the ice (constant; freezing point of sea water) is linear. Consequently and following this assumption, the approach does not explicitly discriminate between different ice types within a polynya, as TIT are solely derived from calculating the heat conduction in/through the ice (aside from subsequent gap-filling; see Sect.~3.2)."

Line 14/15: I understand that the authors mention March here as this month contains the spring equinox. However, November is almost as close to the winter solstice as February is. Could it be that in November the cloud coverage is the problem?

Including the months of October and April would be problematic since the amount of suitable clearsky and nighttime MODIS scenes decreases with increasing amounts of solar radiation.

Page 7, Figure 3: In the case shown there were good TIT maps on January 14 and 16 (i.e. from 2 days of the surrounding 6 days used), i.e. directly adjacent in time to the TIT map from which the MCC filtering removed artificial but also correct TIT areas. I am assuming that this is a very good example. How often did the authors not find appropriate adjacent TIT maps?

You are certainly right that we picked a good example to illustrate the basic principle of our approach at this point of the paper, which combines both a meaningful correction from the MCC filter (which can often be quite subtle) as well as bounding days with a good MODIS coverage in the cloud covered/influenced/spurious regions. Frankly speaking, it is hard to quantify how frequently this "ideal" combination can be found, as it not only varies depending on the location, but also the temporal distance of available pixels for the SFR approach can vary between 1 and 3 days.

Caption, lines 7/8. I am not sure that Spreen et al. (2008) is the only reference you should use here because that paper is addressing AMSR-E while the data you used stem from AMSR2. Hasn't there been a paper by Beitsch et al., Remote Sensing, 2014, about applying the ASI algorithm to AMSR2 89GHz data for sea ice concentration retrieval? The same comment applied to page 8, lines 15/16.

This is correct. We will add the study by Beitsch et al. (2014) to the list of references and quote it, respectively.

Page 8, line 11: I have difficulties to understand Table 1 and the statement of "with certain regions performing better ... and some other regions noticeable worse" If I understood the COV2 and COV4 correctly, then this is giving the fraction of the predefined area (Figure 1) covered by thin ice as retrieved by the authors's method. What seems strange to me is that some of these show a COV4 close to 1, which would mean that the entire predefined area is covered with thin ice. I doubt that KAR is really covered to 95% by thin ice. Possibly I did misunderstand something here. I encourage the authors to clarify this issue and to better explain what their metrics is to decide which is "better" or "worse".

 $\rightarrow$  Please refer to our response under general comment (3).

Line 21-23: I suggest that the authors refer more to their own earlier results (Brunt ice shelf, etc.) because I find it a bit dangerous to conclude that the correction works fine from just one example shown here.

Rewritten to read:

"All derived quality attributes (MCC-filter, cloud-cover information, PIX) are utilized in the Spatial Feature Reconstruction (SFR) algorithm (Paul et al., 2015a), which was recently successfully applied on

a regional scale in both the Antarctic and Arctic to increase the information about otherwise cloudcovered areas (Paul et al., 2015b; Preußer et al., 2015a). The basic principle is that cloud-induced gaps in the daily TIT composites are compared with the TIT of the surrounding six days. In doing so, a probability of thin-ice occurrence is derived using a weighted composite of the days surrounding an initial day of interest (DOI). As in previous studies, we applied the following set of weights:  $w_3 = 0.02$ (DOI  $\pm$  3), w2 = 0.16 (DOI  $\pm$  2) and w1 = 0.32 (DOI  $\pm$  1). The probability threshold remains fixed at th = 0.34 and needs to be surpassed in order to assign 'new' polynya pixels. Paul et al. (2015a) showed that this combination is less restrictive in terms of missing coverage in close proximity of the initial day of interest. The procedure is applied on all areas with identified low-quality data (low persistence, cloud-covered), so that indicated gaps can be filled with new information on potential thin-ice occurrences. For these areas, new TIT and IST values are pixel-wise allocated using a weighted average of the surrounding six days (Paul et al., 2015b; Preußer et al., 2015a). Table 1 gives an overview on the achieved IST and TIT coverage before and after application of the SFR algorithm. On a pan-Arctic level, the average (2002/2003 to 2014/2015) coverage is increased from around 0.75 (ccs and high-quality mcp) to 0.93 (including SFR areas), with certain regions performing better (e.g. CBP, LAP, NEW, SZN) and some other regions noticeably worse (CHU, GLN, WNZ).

A total of 66 case studies in the Brunt Ice Shelf region of Antarctica demonstrated the generally good performance of the algorithm in comparison to more intelligible approaches by realistically reproducing artificially cloud covered thin-ice areas with an average spatial correlation of 0.83 (Paul et al., 2015a). When compared to reference runs based on equally-weighted and in some cases shorter time intervals, the SFR procedure featuring above listed weights  $w_3$  to  $w_1$  (DOI ± 3 days) yielded superior results both in spatial correlation and reconstructed POLA-values, regardless of the temporal polynya-evolution (e.g. opening/closing event).

(...) while maintaining the increased spatial detail at the same time. **Based on this example and above mentioned previous works by the authors (Paul et al. 2015a, Paul et al. 2015b, Preußer et al. 2015b),** we conclude that the applied schemes to compensate and correct cloud-effects work reasonably well **on a pan-Arctic scale** and allow for a fair comparison to other commonly used remote sensing approaches to infer polynya characteristics, with limitations regarding the reconstruction of leads. "

Page 8, line 32 through page 9, line 5: This discussion about the correct sea ice salinity comes back to my previous comment about which ice type the approach can consider. I guess it is worth mentioning whether the approach primarily retrieves TIT in the frazil / grease ice domain until that area where this "unstable" ice starts to collect at the leeward side of the lead/polynya to form nilas and subsequently thicker ice types, or whether the approach primarily considers the nilas and thicker sheet ice types. Actually, if it would be frazil ice, the sea ice salinity might have chosen to be larger; studies focussing on frazil ice use salinities of 917 kg/m3 (de la Rosa and Maus, The Cryosphere, 2012) or 920 kg/m3 (Jordan et al., Journal of Physical Oceanography,2015).

Please refer to our earlier response. As we do not explicitly differentiate between ice types, we chose to stick to a sea ice density of 910 kg/m3 (Timco and Frederking, 1996) for the sake of comparison to earlier studies using the same value for fresh ice.

*Page 9, Table 1: The "plus/minus" values in the column TIT are one standard deviation over all winters considered. How about the respective values in columns COV2 and COV4?*

They also refer to the standard deviation over all winters considered. We augmented the caption accordingly.

Page 9, lines 8-10: "We do not consider an ocean heat flux ..." I agree with the authors that this would complicate the TIT retrieval substantially. I am curious, however, whether your discussion of

uncertainties will reflect that fact that some areas might have substantial oceanic heat fluxes. The authors might want to consider one further reference in this respect: Yao and Tang, The formation and maintenance of the North Water polynya, Atmosphere-Ocean, 41(3), 2003, and also cite Melling et al., 2015 here.

Thank you for this remark (please compare general comment (4)). Including an appropriate parametrization for a varying influence of ocean heat fluxes is certainly challenging, as you correctly write above. Information on respective numbers and orders of magnitudes are sparse, and even more so during wintertime. We briefly mentioned this topic in our paper on the North Water polynya in Northern Baffin Bay (Preußer et al., 2015) with reference to the study by Yao and Tang (2003). We added this study at the referred part of the manuscript. In the same manner, the study by Melling et al. (2015) (quoted in Sect.4.1 when referring to Fig.4) dealing with 'Invisible polynyas' in the Canadian Arctic Archipelago is certainly a welcome addition at this point of the manuscript.

Page 10, line 5: I guess the authors wanted to refer to either "optical and infrared" or even only "infrared" instead of "optical" here.

**Fixed, thank you for this suggestion.**

Line 10: "falls below 0.5" I have difficulties to understand the authors' concept of using the fraction of the predefined polynya regions shown in Figure 1 as a quality measure. I commented on that already in the context of table 1. Here, the authors limit the fraction of thin ice in these predefined areas to be above 0.5 - if I have understood this correctly. Or, in other words, it reads as if a thin ice fraction of the predefined polynya regions in Figure 1 needs to be above 0.5, otherwise it is regarded faulty. I probably misunderstood something?

 $\rightarrow$  Please refer to our response under general comment (3).

**Results and Discussions:**

Page 10, line 16/17: The trend in TIT mentioned in these lines are not summarized in any of the tables, am I correct? Perhaps the authors could spend a "(not shown)" or something?

You are correct, trend are not summarized/listed at any point in the manuscript. We added a '(not shown)' to avoid confusion.

*Lines 29-33: I suggest the authors cite work which is related to the derivation of fast-ice extent in, e.g. the Laptev Sea like for instance: Selyuzhenok et al., J. Geophys. Res., 2015.*

We added three references here, so that it now reads: "(...) for a regular Arctic-wide mapping of monthly fast-ice extents and could thereby compliment currently existing approaches from earlier studies (e.g. Yu et al., 2014; Mahoney et al., 20146; Selyuzhenok et al., 2015)."

Page 12, Table 2: I am wondering whether the trends given are "per year" as indicated or "per decade"? If these are indeed per year, then in region ESF the increase in POLA would be 1.095 km2 in 10 years which equals the average POLA value given. The same applies to region SZN. Perhaps the authors could check which reference period they used for their trend calculations? The authors might also consider to write how the p-values were derived, i.e. which statistical test was carried out.

<sup>6 Mahoney, A. R., Eicken, H., Gaylord, A. G., and Gens, R. (2014): Landfast sea ice extent in the Chukchi and Beaufort Seas: The annual cycle and decadal variability, Cold Regions Science and Technology, 103, 41–56, doi:10.1016/j.coldregions.2014.03.003.

Derived and indicated trends do indeed refer to 'per year', i.e. winter-period from November to March. POLA values for the East Siberian Fast-ice mask range between ~ 0 and 3000 km2, which is resulting in the average value of as depicted in Fig.5 and Tab.2.

The p-values are based on a **two-sided t-test**. We added this information at appropriate parts of the MS (e.g. Table 2; Fig.8; Text Sect. 4)

Page 12, line 11: Stylistically I would say "the large POLA values" is enough here (instead of "these") because the authors refer to NOW in the remainder of the sentence. I note in this context, that the increase in NOW POLA is not significant in the authors' study.

Thank you for these remarks. We changed the formulation and added information on the (in-) significance. It now reads:

"The study of \citet{preusser2015b} demonstrated that the large POLA values in the NOW-region are part of a (non-significant) long-term increase of average polynya extents between 1978 and 2015".

Page 15, lines 1-8: The authors inter-compare their results with Kern (2008), who only focused in the Kara Sea. Aren't there other studies about polynya area which results would be worth to compare the authors' results with?

There are certainly some other studies with information on POLA, such as the often referred Pan-Arctic studies by Tamura & Ohshima (2011) and Iwamoto et al. (2014) or many local studies. At this point of our submitted manuscript, we want to focus on one of the major regions (Kara Sea) that was not featured in our previous regional studies (STO, NOW; LAP in Sect. 4.2). Hence the comparison to Kern (2008).

Lines 8/9: "increases for" Do the authors refer to an increase in POLA or to an increase in POLA variability?

 $\rightarrow$  We refer to an increase in POLA variability.

Lines 9-20: I absolutely agree with the authors' observations written down in this part. The only concern I have here is: Where do the authors differentiate between IP during regular fall freeze-up and IP within polynyas and leads. Or in other words, when do the authors define an open water / thin ice areas to be belonging to a polynya and when is this still considered fall freeze-up? In this context: in the caption of Figure 4 the authors make a note that they discarded the regions of high TIT frequency along the marginal ice zones facing the Nordic Seas from further analysis. Wouldn't it make sense to do the same for the northern Baffin Bay (in November) and also the southern Chukchi and Beaufort Seas (in November)? Also: What was the criterion to exclude areas with a high TIT frequency? I could not find a notion how exactly these regions were defined. Did the authors used a TIT frequency threshold?

 $\rightarrow$  Please refer to our response under general comment (4).

Line 23: "slight decrease" I suggest the authors add that these decreases are far from being significant.

We changed the mentioned part to read: "a slight, yet insignificant decrease..."

*Line 26: "plus/minus 258 km^3" Is this an uncertainty, or is this the standard deviation from computing the average IP of the 13 winters?*

**As in Tab.2/3, "± 258 km3" refers to the standard deviation over the 13-yr period.**

Line 30-31: I suggest that the authors comment more on this comparison. Tamura and Ohshima's results are based on SSM/I data while Iwamoto et al. base their study on AMSR-E data. The authors'

study is based on MODIS data. This implies different spatial resolutions which effect on the results could be discussed here. Actually, in the next paragraph starting in line 32 the authors carry out this discussion but without linking it to the statement in lines 30-31 and without trying to investigate (and discuss theoretically) whether 2 km instead of 6.25 km grid resolution would allow to explain the larger IP found in this study compared to Iwamoto et al. Yes, I agree, with a finer grid resolution one is able to identify smaller scale thin ice features. There is no doubt about that and this has been demonstrated in previous papers of the leading author. But at the same time POLA of larger polynyas could become smaller because the polynya edge is better defined at 2 km than at 6.25 km. Therefore there could be competing effects with the net effect being zero. In addition the period of Iwamoto et al. is much closer to the one used by the authors. By looking at the winters 2011/12 through 2014/15 the authors could check whether their larger value compared to Iwamoto et al. could be explained by considerably larger IP in these winters compared to the winters before 2011/12.

**$\rightarrow$ Please refer to our response under general comment (1).**

Further, we took a closer look at the numbers from Iwamoto et al. (2014) and our numbers up until 2010/2011. It shows, that the winter seasons from 2011/2012 onwards vary considerably between low (2011/2012) IP and the largest (2012/2013) IP in our 13-yr time series. Thereby, the average value for 2002/2003 to 2010/2011 is not affected very much by leaving out the last 4 winter seasons and amounts now (incl. new WNZ region) to around 1789 km3/winter (~ -1-2%).

We added some more comments on that comparison at the mentioned part in Sect. 4.1.

Page 17, Figure 8: I am wondering why the map showing the significance is smaller than the one showing the trends. I suggest to make both maps the same size or, alternatively, to overplot significance levels on an even enlargened version of image a) using, e.g. dots and crosses to denote areas of >95 and >99% significance or isolines. However, what is a bit unfortunate here - as well as already in Figures 4 and 7 is the fact, that the marginal ice zone (MIZ) facing the Nordic Seas is visually dominating the Figure and distracts the eye from those regions which are really relevant for the present study. In the context of the yet unexplained way how these MIZ areas are excluded (according to the caption of Figure 4), I encourage the authors to find a way to make these areas to appear less prominent, perhaps by grey shading or similar, so that the reader can focus on the relevant areas.

We changed the overall appearance of several figures (3, 4, 7, 8, 10, and 11) according to the feedback of all three reviewers. Thereby, relevant regions are now additionally marked using the polynya masks, as can be seen below.

---

## Author Comment (AC2) · 11 Oct 2016

*Received from Prof. Göran Björk (referee) on August 23, 2016*

**General comments**

*This work gives a comprehensive description of Arctic polynyas based on high resolution surface temperature (MODIS) data. This data set has the advantage of a higher spatial resolution compared with satellite products used in earlier investigations. It gives a 22 % higher total polynya ice production than recent results which shows that the development of satellite products, algorithms and analyses is still an important issue in order to follow the past and future development of the Arctic Ocean ice cover. It reveals significant positive trends of the polynya ice production in the eastern arctic which can be further utilized for analyses of the effect on dense water mass formation on the shelves which likely have influence on shelf circulation, shelf basin interaction and water chemistry. The paper is generally of a high quality in language and analyses and is therefore well suited for publication.*

We would like to thank the Prof. Göran Björk (referee #2) for his valuable comments and remarks. We carefully went over the mentioned parts of the manuscript. Specific comments will be addressed in the following.

**Specific comments**

*I'm not perfectly happy with the sentence starting on line 6 page 3 ("A regular monitoring..."). It appears to be somewhat a repetition of the sentence on page 5 line 8 ("Hence an accurate...").*

Thank you for this remark, but in our sense the sentence on page 6 refers to the general monitoring of thin-ice areas using remote sensing data, while the section on page 5 refers to the determination of sea-ice production.

*Page 9 Table1 text. I miss some more explanation of what the "interannual average coverage" means. Coverage of what? It is hard to understand as it stands now.*

The referee is right regarding the somewhat misleading / confusing formulation of the caption here. We changed it accordingly to read:

**"Areal extents (i.e. total ocean area) of all applied polynya masks in km². Further, the interannual average amount of MODIS swaths that could be used for calculating daily composites in a given region is indicated, together with the interannual average daily MODIS coverage (decimal cover fraction ranging from 0 to 1 with their respective standard deviations) before (COV2) and after (COV4) application of the Spatial Feature Reconstruction (SFR) for each polynya region from 2002/2003 to 2014/2015 (November to March). (…)"**

*Page 9 Line 2 and before. It is hard to follow the logic why fixed values for ice and Lf are used. The arguments regarding frazil ice crystals are not clear to me.*

The presented thin-ice algorithm does not explicitly discriminate between different ice types. It follows the assumption that a linear temperature profile can be used to calculate the heat conduction through the ice. Hence, we added this information to the manuscript. Regarding the choice of constant values

for the ice density and latent heat of fusion ($L_f$), we followed earlier studies (e.g. Willmes et al. (2011), Tamura and Ohshima (2011), Iwamoto et al. (2014)) to ensure comparability of achieved results. These studies followed an even earlier characterization of sea-ice formation mechanisms by Martin (1981).

*Page 12 Line 2. It is interesting to see persistent leads well off the shelf in the Beaufort Sea. These must be related to the large scale ice circulation in the area and it is remarkable that they are so persistent that the show up as well defined bands in this type of data (most notable in Feb-Mar). I wonder if this structure has been described before or if it is a new finding. It is worth some more comment anyhow.*

Thank you for this interesting remark. Indeed, these broad lead-structures in the Beaufort Sea (related to the clockwise rotation of the Beaufort Gyre) have been previously described e.g. by Willmes and Heinemann (2015, Remote Sens., doi:10.3390/rs8010004), who also used MODIS TIR data, and also by Röhrs et al. (2012, TC, doi:10.5194/tc-6-343-2012), who used coarser resolution AMSR-E passive microwave data for their analysis.

What is interesting in the present study though, is the relatively high persistence of these leads (so that they are not discarded from our daily thin-ice distributions) together with apparently distinct favorable locations of appearance so that they appear in the these interannual frequencies of TIT ≤ 0.2m. Therefore, we added the following statement *"(…) leads are mainly located in the area of the Beaufort Sea and north of Greenland (shear zones)* **which can be attributed to their relatively high spatial and temporal persistence. (…)**".

*Page 12 Line 3. I can't see the leads along the Transpolar drift in figure 4. The central area around the North Pole appears to be without leads in the figure.*

The referee is correct with this remark, as we were aiming to highlight enhanced TIT frequencies in the Atlantic sector of the Transpolar Drift (~Fram Strait region; see above for FEB). However, as frequencies are quite low and mainly located outside our indicated regions of interest (i.e. polynya margins; Fig.1), we decided to remove this sentence to avoid confusion.

*Page 15 Line 8. Suggestion: "is especially large " instead of "increases".*

Fixed, thank you for this suggestion.

*Page 19 Line 13. Sentence starting with "A pronounced seasonal..." is unclear. I can see that the seasonal variation is largest in the late half of the period, but the last part is confusing.*

We slightly changed the sentence in order to make it less confusing, so that it now reads:

"***A pronounced seasonal variation is visible for the winter seasons 2004/2005, 2005/2006 and from 2010/2011 onwards, while the other years show less polynya activity (more lengthy periods with a closed polynya; white color in Fig. 10) and overall smaller polynya extents in February and March.***"

*Page 19 Line 18. I think the reader needs some more help to identify the fast ice edge in figure 11 and also in earlier figures. It is not clear to me since there are several bands of high ice production from the coast and outward in most of the fields.*

The referee is right that characteristics of the fast-ice edge might be difficult to assess for readers who are unfamiliar with the topic. This is especially true when showing plots that integrate over the period from Nov.-Mar., and therefore inhibit different stages of fast-ice development. However, a complete mapping / marking of these areas is a quite challenging task and definitively not feasible for this present study. In order to address your remark, we decided to keep most of the figures as they are (except for Fig.9 where we inserted the approximate average position of the fast-ice edge at the end of March; see below) and put more effort on describing the characteristics.

*P21 Line 9. IAE need to be defined better. Is it the export just outside the Laptev Sea or what?*

To quote Krumpen et al. (2013), the ice area flux is calculated as the integral of the product between the U and V component of the ice drift velocity and ice concentration at the northern boundary (NB) and eastern boundary (EB) of the Laptev Sea. In their study, a positive flux (given in km²) is referred to an export out of the Laptev Sea into the Transpolar Drift and East Siberian Sea, while a negative flux denotes to an import into the Laptev Sea. Please refer to the mentioned study for more details on the data sets, calculation procedure and outcomes.

The geographical locations of these two boundaries (NB/EB) on which meridional and zonal ice area flux estimates were based in Krumpen et al. (2013) are now additionally illustrated in Fig.9 (see below, cyan solid lines in the inset) and **some further explanation on the IAE values are now given in the manuscript.**

[Figure]

*Figure 1 The geographical location of the Laptev Sea in the eastern Arctic. The applied polynya mask is marked in red, enclosing the locations of typical polynya formation along the coast and fast-ice edge (dashed white line; position derived from long-term thin-ice frequencies in March (Fig. 4)). Flux gates from the study by Krumpen et al. (2013) at the northern (NB) and eastern (EB) boundary of the Laptev Sea are shown in the inset map (grey solid lines). Bathymetric data by Jakobsson et al. (2012) (IBCAO v3.0).*

**Technical corrections**

*None*

---

## Author Comment (AC3) · 11 Oct 2016

**Received from Anonymous Referee #3 on August 24, 2016**

**General comments**

Coastal polynyas play a crucial role in altering a variety of physical, biological and chemical processes at the boundary between the atmosphere and the ocean. In the case of Arctic Ocean, polynya ice production is a key component for understanding the maintenance and variability in ocean stratification (cold halocline) and ice-ocean interaction, as well as the seasonal sea-ice mass budget. This paper provides the circumpolar mapping of polynya area (POLA) and its ice production (IP) in the Arctic Ocean, with fine spatial resolution of about 2 km. This resolution is much finer than the previous mapping with satellite microwaves. The authors have accomplished the creation of the dataset of POLA and IP by treating massive amount of 143000 MODIS data, with well-refined procedures. As well, the paper provides 14-year dataset of POLA and IP, which will be the basic data for understanding of drastically changing Arctic Ocean. The paper is overall logical, well-organized, and the presentation/writing is refined. Although the results might have some bias arising from that the calculations were made only for clear-sky and nighttime conditions, this is mainly because of limitation of satellite (MODIS) data. I think that the authors have done a best to create the circumpolar data set with a high spatial resolution. I believe that the paper surely contributes to the community of Arctic and climate sciences. But there still remains some points that should be improved, all of which are minor ones. Some figures can be a bit improved for clarity (see comments 7, 8, 12, 14, 15 for details). In brief, the paper should be published in Cryosphere after a minor revision. The specific points are the followings.

We highly appreciate the valuable and constructive comments and suggestions from Referee #3 and would like to thank her/him for her/his efforts. The overall quality of our submitted manuscript will certainly benefit from the listed specific comments, all of which we will respond to in the following.

**Specific comments**

1. MODIS clear-sky data can be obtained rarely in the polar cloudy condition. Thus most of researchers including me think that it is difficult to obtain seamless (daily) surface dataset from the MODIS data. For example, in investigation of landfast ice (Fraser et al., 2012, J. Climate) from MODIS, data set was made only for 20-day interval because of cloudy condition. At first I could not believe the average coverage fraction of 70-80% per day (Table 1) in this study. However, if the MODIS image can be obtained for one area several tens of times per day, composite of clear-sky portion could offer the daily data. I guess this is the case and explain why such high fraction of coverage is possible. If this is true, the authors should clearly explain why such high fraction of coverage can we obtain the cloud free scene? I think that such explanation enhances the creditability of this study.

It is understandable that this circumstance might be surprising at first glance, as thermal infrared data is strongly influenced / limited by the presence of clouds – especially in the polar-regions. We are certainly aware of these difficulties. However, the calculation and usage of daily median composites of IST / TIT enables a vastly increased spatial coverage of these quantities, based on the principle/assumption that clouds move over sub-daily timespans. Of course this assumption can be

violated as clouds also tend to behave rather stationary. In those cases, cloud-gaps cannot be avoided completely.

As already written in the manuscript (P.5, L.11), an average of around 73 MODIS swaths per day is available for the Arctic domain. The absolute amount of overpasses for a certain region increases with latitude due to the polar-orbit configuration of the MODIS sensors onboard Terra and Aqua. Therefore, regarding the request of the referee, we **included an additional column in Table 1 that features the average amount of MODIS scenes per day and per region**.

2. Although the MODIS data provide high resolution data set, POLA and IP can be obtained only in clearsky condition. The atmospheric condition and accordingly surface heat flux in clear-sky condition would be different from those in cloudy condition. Thus it is likely that POLA and IP obtained all from clear-sky condition have some bias compared to those from cloudy condition or pure average irrespective of atmospheric condition. I understand that evaluation of such bias is not easy and no further analysis is needed. But **more discussion or clear statement of such bias should be made in the revision**. At least such drawback should be stated in conclusion section.

The exclusive usage of clear-sky pixels is a prerequisite of our approach and can't be avoided using TIR data. The evaluation of such a potential bias is certainly an interesting aspect for further improvements to our TIT retrieval scheme, but at the same time (as you already mentioned) quite challenging. In this regard, a potential bias might originate from both sub-daily as well as daily timescales.

On a sub-daily timescale, the POLA retrieval can be assumed to be only little affected by the bias to clear-sky and nighttime conditions, since the TIT (taken as the daily composite) will not change that much, if clouds are present. This is different for the IP, which is computed from energy fluxes. While the turbulent fluxes of sensible and latent heat over polynyas are relatively insensitive to cloudiness, the increase in longwave downward radiation will cause a lower IP in reality. This will lead to a systematic overestimation of IP in our method. However, this is the case mainly for low-level clouds, which emit at a relatively high temperature. Heinemann and Rose (1990)1 show that this effect can amount up to 50 W/m2. König-Langlo and Augstein (1994)2 show that the effective emissivity ( $\epsilon_{atm}$ ) increases from 0.765 (clear sky) to 0.985 (fully cloudy), taking the 2m-temperature in the Stefan-Boltzmann law for the computation of the downward longwave radiation (L $\downarrow$ ). For typical L $\downarrow$  values of about 200 W/m2, the increase by 0.22 for the emissivity would also result in an increase of around 50 W/m2, thereby impacting the total energy balance considerably. For an estimation of the actual error on a sub-daily basis, L $\downarrow$  would have to be weighted with the percentage of cloudy overpasses.

Regarding a bias originating from the SFR-approach/cloud-interpolation, further sophisticated comparisons with cloud-insensitive active or passive microwave remote sensing data could be helpful, but certainly go beyond the scope of the here presented manuscript.

We augmented the **conclusions** to read: "Compared to the most recent study on ice production in Arctic polynyas by \citet{iwamoto2014}, our estimate on the average total ice production is about 52-54\% larger, although differences in the regarded time frame, reference areas, sensor-specifics **as well as a potential biases due to cloud cover and/or the exclusive assumption of clear-sky conditions** certainly contribute to this discrepancy."

<sup>1 Heinemann, G., Rose, L., 1990: Surface energy balance, parameterizations of boundary layer heights and the application of resistance laws near an Antarctic ice shelf front. Boundary Layer Meteorol. 51, 123-158.

<sup>2 König-Langlo, G. and Augstein, E. (1994): Parameterization of the downward long-wave radiation at the Earth's surface in polar regions , Meteorologische Zeitschrift, N.F.3, 343-347.

Further, also in **Sect.3.3** it now reads: "(...) Since low-level clouds reduce the net radiative loss by about 50 \unit{W/m^2} in polar regions (Heinemann and Rose 1990, König-Langlo and Augstein 1994), the restriction to cloud-free conditions in the daily composites results in a positive bias in IP. Considering the fraction of average MODIS coverage of 75\% (COV2; Tab.~\ref{tab:tab01}) and assuming that not all clouds are low-level, the overestimation of net energy loss by our method can be estimated to be less than 10 \unit{W/m^2}, which corresponds to less than 0.4 \unit{m} IP per winter."

3. Similarly, POLA and IP can be obtained only in nighttime and thus POLA and IP obtained all from nighttime likely have some bias compared to those from daytime or pure average. Although a brief statement was made in page 9, it may be better to evaluate such bias even in a brief way. For example, difference in heat budget on thin ice for nighttime and daytime under a typical wintertime condition can be evaluated.

We appreciate the referee's remark on a possible bias due to the exclusive analysis of nighttime conditions. While we do agree that the influence of shortwave radiation and albedo effects could be rather significant, we do not think that a bias evaluation would contribute in a meaningful way to the here presented study due to the following reasons: Currently, our method does not feature a shortwave radiation parametrization, as (our) previous studies showed that the implementation of such can be rather problematic and connected with ambiguities. In addition, the implementation of such would introduce further error-sources that would make it even more complicated to evaluate a possible bias.

4. The study does not include the results of October and April when the polynya activity starts and continues, which is one of the drawback of this study. According to Iwamoto et al. (2014), for example, these two months provide 10-30 % of total annual ice production (IP). Particularly in NEW, Laptev, Archipelago, IP becomes maximum in October. Such drawback should be stated in IP section and conclusion.

Including the months of October and April would be problematic since the amount of suitable clearsky and nighttime scenes decreases with increasing amounts of solar radiation.

5. Abstract: "Overall, our study contains the most accurate characterization of circumpolar polynya dynamics and ice production to date". This statement is ambiguous and overvaluing. The authors should state more specifically in what points this study provides the most accurate characterization? Probably, high spatial resolution is strong selling point. On the other hand, this study still has the drawback of data gap by cloud.

We admit that this statement might have been too unspecific in that context. This was also a remark from Referee #1. Therefore, we changed this part and a similar formulation in the conclusions-section to read:

"Abstract: Overall, our study **presents a spatially highly accurate characterization** of circumpolar polynya dynamics and ice production which should be valuable for future modeling efforts on atmosphere- sea ice - ocean interactions in the Arctic."

and

"Conclusions: (...) we think that this new data set of 13 consecutive winter seasons is a huge step forward for a **spatially** accurate characterization of Arctic polynya dynamics and the seasonal sea-ice budget in general." Regarding the mentioned drawback of our applied method, we refer to the limitations of thermal infrared data at several parts of the manuscript (e.g. Sect.2, Sect.3, Conclusions), as we are absolutely aware of them.

6. P3, L15-16: "west of Novaya Zemlya is excluded in our investigations due to a variety of potential ambiguities originating from ocean heat fluxes": I understand the situation. But, as described in the textbook by Martin (2001, Polynyas. In: Encyclopedia of Ocean Sciences. vol 3. Academic Press,), the Novaya Zemlya polynya is one of the most active polynya, and other studies (e.g., Iwamoto et al., 2014) includes the Novaya Zemlya polynya in their tables. Similar situation by the effect of ocean heat also occur in the polynyas of Storfjorden, Franz-Josef Land. Why only the Novaya Zemlya polynya is excluded?

In recent years, the area at the northern tip and western coast of Novaya Zemlya was rarely fully enclosed by sea ice during winter. Initially, this was one of the main reasons why we decided to exclude this region as seemed to more fulfill MIZ characteristics in our opinion.

Nevertheless, motivated by the reviewers comment we took a closer look at this region and decided to include it in an updated / revised version of the manuscript. Following Årthun et al. (2012), at the least the influence of the eastern branch of Atlantic water spreading into the Barents Sea seems to be lower as expected in this region. It remains up for debate if those regions with changing ice conditions in recent years can be considered as a polynya region in a textbook sense, but in order to increase consistency regarding the considered polynya regions to Tamura and Ohshima (2011) and Iwamoto et al. (2014) the manuscript was changed accordingly with an additional polynya mask "**Western Novaya Zemlya**" (WNZ). Necessary changes can be found to the marked up version of the manuscript.

Figure 1 Map of all investigated areas of interest located in the Arctic, north of 68 ° N. Except for the Laptev Sea (red frame), all other applied polynya masks are marked in blue and enclose the typical location of each polynya in wintertime.

Årthun et al. (2012)3: "The inflow of Atlantic water between Norway and Bear Island [the Barents Sea Opening (BSO); e.g., Ingvaldsen et al. 2002] is the Barents Sea's main oceanic heat source. The inflow consists of several branches (Fig. 1a; Loeng 1991) but mainly follows a counterclockwise circulation before exiting the Barents Sea between Novaya Zemlya and Franz Josef Land (Schauer et al. 2002). During its passage through the Barents Sea, the Atlantic water loses most of its heat to the Arctic atmosphere (Häkkinen and Cavalieri 1989; Årthun and Schrum 2010), and the heat transport through the northern exit is consequently small (Gammelsrød et al. 2009). The dominant role of the Atlantic inflow on the Barents Sea heat budget and its intimate link to surface heat fluxes are further evident from the close correspondence between observed volume transport through the BSO and thermal water mass transformation in the western Barents Sea (Segtnanet al.2010)."

**7. In some figures (Figs. 4, 7, and 8), coast lines are not visible.**

It is understandable that the line-width of the coast-lines might seem a tad small/narrow, although still visible in several test-print outs. The small width was chosen on purpose for these pan-Arctic overviews, in order to not distract the readers view from polynya-activities close to the coastline.

**8. P6, Figure 3: The scale in the right bottom should be enlarged.**

Fixed, thank you for this remark.

---

## Author Response (AR2)

Report #1
Submitted on 03 Nov 2016
Referee #1: Stefan Kern, stefan.kern@zmaw.de

Anonymous during peer-review:                              Yes **No**
Anonymous in acknowledgements of published article:   Yes **No**

Recommendation to the Editor

**1) Originality (Novelty)**                                    Excellent  **Good** Fair Poor
Within the scope of The Cryosphere, does the manuscript represent
substantial progress beyond current scientific understanding (new
insight, concepts, methods, or data)?

**2) Scientific Quality (Rigour)**                              Excellent **Good** Fair Poor
(A) Is the purpose of the work clearly articulated, reflected in an
adequate methodology, and its achievement compellingly
underpinned by the evidence presented?
(B) Are the applied methods and techniques valid and suitable?
(C) Are the results discussed in an appropriate and balanced way
(consideration of related work, including appropriate references)?

**3) Significance (Impact)**                                    Excellent **Good** Fair Poor
Does the manuscript contribute to changing our scientific
understanding of a subject substantially or to introducing new practical
applications of broad relevance?

**4) Presentation Quality**                                     Excellent **Good** Fair Poor
Are the scientific results and conclusions presented in a clear, concise,
and well-structured way (number and quality of figures/tables,
appropriate use of English language)?

For final publication, the manuscript should be
accepted as is
accepted subject to technical corrections
**accepted subject to minor revisions**
reconsidered after major revisions
       I would like to review the revised paper
       I would NOT be willing to review the revised paper
rejected

Please note that this rating only refers to this version of the manuscript!

We thank you once again for your comprehensive report on our revised submission. You point out
some important issues, all of which we will address in the following.
Please refer to the new revised version of the manuscript for all applied changes and/or corrections.

Suggestions for revision or reasons for rejection (will be published if the paper is accepted for final publication)

Review of

Circumpolar polynya regions and ice production in the Arctic: Results from MODIS thermal infrared imagery for 2002/2003 to 2014/2015 with a regional focus on the Laptev Sea - Revision 01

by
Preußer, A., et al.

I thank the author team for their careful consideration of the reviewer's comments and concerns. Reading the rebuttal letter doesn't leave anything open except one thing:

I wrote: Page 6, Line 14/15: I understand that the authors mention March here as this month contains the spring equinox. However, November is almost as close to the winter solstice as February is. Could it be that in November the cloud coverage is the problem?

You replied: Including the months of October and April would be problematic since the amount of suitable clearsky and nighttime MODIS scenes decreases with increasing amounts of solar radiation.

My comment (again): In the 1st version of the manuscript you wrote:

Because of the restriction to nighttime scenes, a less frequent IST coverage is present in the beginning (November) and at the end (March) of each winter-season.

Winter solstice is on December 21, right? Therefore sun elevations and hence day lengths are similar on November 21 and January 21, on October 21 and February 21, and so on. From that I'd expect that the number of nighttime cases is similar for the month pairs December/January, November/February, and October/March. Therefore I felt that mentioning November and March in one sentence explaining the limitation to fewer valid cases does not fit and was suggesting in my former review that clouds might have had a larger impact in November - which is something found in the literature as well.

We apologize for not being precise enough in our previous reply, as you are right about this irritating formulation. What we had in mind was the decreasing amount of available MODIS swaths before (so also November) and after the winter solstice, but of course especially March suffers from the increasing amount of shortwave radiation. This is exemplary illustrated in the Fig. below, which shows the daily amount of MODIS swaths (north of 68°N) with valid IST information in 2013/2014. Overall, we don't see a significantly increased effect of clouds in November.

[Figure]

We rephrased this part to make it more concise. It now reads:
"(…) Because of the restriction to nighttime scenes, a less frequent MODIS coverage is present especially towards the end (February to March) of each winter-season."

##################################

General comment:

The manuscript has improved substantially and is almost ready to go.

However, I still have a concern about the usage of polynya area and the associated ice production. The fact that previous studies used similar regions as are used in this study should not prevent the authors from pointing out potential shortcomings of those definitions.
- I strongly suggest therefore that the authors clarify their understanding of a polynya (in the context of how WMO defines it) and their usage of polynya area.
- I suggest further that the authors think one more time about the way they include the results of regions WNZ and KAR into their analysis and results and state even more clearly that their results of particularly these two regions are heavily influenced by ice formation in a MIZ rather than in a polynya.

We understand that especially at the beginning of a winter season, text-book / WMO definitions of a polynya might not be fulfilled in every polynya (i.e. thin ice) region, first and foremost those in the Atlantic sector of the MIZ as you correctly mention (KAR, WNZ, SVA). [*However, this somewhat more frequently appearing "MIZ behaviour" evolved gradually over the investigated period and became more common just in the last 4-5 winter seasons.*]

In the revised MS, we already mention the MIZ influence for the indicated regions (from which the WNZ was specifically requested by Referee #3), but we could highlight this more clearly as you correctly point out.

Although we share some concerns regarding MIZ characteristics, we nevertheless decided to include the WNZ region to ensure comparability to earlier studies. To quote our response to Referee #3:
*"In recent years, the area at the northern tip and western coast of Novaya Zemlya was rarely fully enclosed by sea ice during winter. Initially, this was one of the main reasons why we decided to exclude this region as seemed to more fulfill MIZ characteristics in our opinion. Nevertheless, motivated by the reviewers comment we took a closer look at this region and decided to include it in an updated / revised version of the manuscript. Following Årthun et al. (2012), at the least the influence of the eastern branch of Atlantic water spreading into the Barents Sea seems to be lower as expected in this region. **It remains up for debate if those regions with changing ice conditions in recent years can be considered as a polynya region in a textbook sense, but in order to increase consistency regarding the considered polynya regions to Tamura and Ohshima (2011) and Iwamoto et al. (2014) the manuscript was changed accordingly with an additional polynya mask "Western Novaya Zemlya" (WNZ).** Necessary changes can be found to the marked up version of the manuscript."*

Further down, you highlight this year's unusually late freeze-up as an example of large-scale freeze up conditions. We expect that this development will continue to extend further into Nov./Dec. over the next years/decades which will eventually lead to some necessary adaptions in our approach for future investigations.

To address this topic in the MS, we now added the following part to Sect. 1:

"The marginal ice zone (MIZ) in Fram Strait and northern Barents Sea is **mostly** excluded in our investigations due to a variety of potential ambiguities originating from ocean heat fluxes and a high interannual variability of the MIZ in terms of location and extent. **However, in order to ensure consistency to previous studies, the MIZ is to some extent included in the CHU, STO, NOW, WNZ and KAR areas. For those regions, this implies that the here derived characteristics may contain periods with extensive ice-free conditions, first and foremost in early winter.**"

And the following part to Sect. 3.3:

"The complete period from November to March each winter is considered for the calculation of POLA / IP, which implies that the here derived values are potentially influenced by shifts in the timing of freeze onset during the early freezing season (November / December). **For potentially MIZ-influenced regions (CHU, SVA, NOW, WNZ, KAR), this has to be considered when comparing metrics derived for the full winter period (November to March).**"

*These two concerns plus the specific comments led me to the suggestion that the authors should have the chance to for a minor revision which are to the authors' discretion, i.e., there is no need for me to have another review.*

Specific comments / suggestions / typos:

P4, L31: Please check meaning of "inhibit". I would go for "contain" or "exhibit".
Replaced by "contains".

P5, Figure 2, Caption, 3[rd] line: "'IST' denotes to …" Please either change "to" to "the" or omit completely.
Fixed.

P6, Lines 6-12: I suggest to add the information that the approach assumes a sheet of ice being present, i.e. that is not able to include the frazil / grease ice stadium.
Slightly changed to: "(…) the approach does not explicitly discriminate between different ice types within a polynya, as TIT are solely derived from calculating the heat conduction in/through **an assumed layer of ice** (aside from subsequent gap-filling; see Sect.~3.2)."

P6, Line 23: The authors could motivate the usage of the median instead of the mean in one additional sentence.
It now additionally reads: "The median is preferred over a simple average in order to reduce the potential risk of erroneously high or low values in single swaths, originating e.g. from unidentified clouds."

P6, Line 31: "completely uncovered pixels" means that there are no MODIS data available?
This is exactly the case.

P8, Line 16: "weighted average" Does this averaging use the same weights as mentioned above in Lines 11-12?
You are right, the same weights are used. We added this to L.16.

P8, Line 26: Please define "POLA".
As "POLA" is introduced a bit later in Sect. 3.3, the part here is slightly altered to read "(…) yielded superior results both in spatial correlation and **reconstructed polynya extent**, regardless of (…)"

P10, L9: How is Q_ice computed? I assume it is computed per pixel and it is computed as a function of the daily composite TIT of that respective pixel? I guess an additional sentence clarifying the procedure would help.
The conductive heat flux through the ice (Q_ice) and with that the equal atmospheric heat flux Q_atm is indeed calculated pixel-wise. Q_ice is defined as

$$Q_{\text{ice}} = \kappa_{\text{ice}} \times \frac{(T_{\text{surf}} - T_{\text{f}})}{h_{\text{ice}}} \qquad \text{(Eq.1)}$$

To quote Preußer et al. 2015a (Sect. 2.3):

*"To obtain the ice thickness (h_ice), the total atmospheric flux Q_atm (Eq. 2) is set equal to the conductive heat flux through the ice (Q_ice; Eq. 1), and Eq. (1) is solved for h_ice using a value for the thermal conductivity of κ_ice = 2.03 Wm^−2 K^−1 (Drucker et al., 2003)."*
However, we do not think it is necessary to include this once again, as the reader is already referred to the indicated references for further details on the (well-known) procedure (Yu and Rothrock (1996), Yu and Lindsay (2003), Drucker et al. (2003), Willmes et al. (2010/2011), Adams et al. (2013), Preußer et al. (2015a,b), Paul et al. (2015b), etc.).

P10, L24: "areal extent of each pixel" Would it be fair to assume that this is 4 km²?
This is correct, the average extent of a pixel is around 4 km². However, due to the currently used equirectengular grid the resolution in longitudinal direction is slightly varying with latitude, so that it increases south and decreases north of 79°N.

P10, L25: "extrapolated to daily rates" One could also write instead: "multiplied by 86400 to obtain IP / day."
Despite this reasonable remark, we decided to keep the current formulation at this point of the MS.

P11, Line 3: "stable cloud cover": In line 5 you write "persistent" instead of stable. I suggest to use "persistent" throughout.
Fixed accordingly, thank you.

P11, Lines 6-9 + Figure 4: While these 3 lines are a step into the correct direction I would suggest to take into account **two more important aspects** here:
i) Starting early in the freezing season might imply that **the definition of a polynya is simply not yet fulfilled**. Hence what is termed "POLA" is not POLA because it is not related to a polynya by definition. A look at Figure 4, November, reveals this clearly: In the eastern Kara Sea TIT frequencies are low because the thick ice has arrived. Polynyas at Islands and along the coast can be delineated. Towards the southwest TIT frequency first increases and then decreases again towards the Kara Strait. **The fact that TIT frequency is low close to the Kara gate is due to the fact that sea ice is still absent there (on average) in November and that the Kara Strait is presumably still open**. Kern et al. (2005, Geophys. Res. Lett.) stated:
*"In each season the PSSM (e.g. polynya area retrieval) analysis starts once the entire Kara Sea is ice covered or an ice bridge has formed in the Kara Strait so that the remaining open water area inside the Kara Sea can be regarded as a polynya. The such defined starting date varies from mid-November to the end of December."*
Because of this the results of Kern et al. (2005) and Kern (2008) are based on the period January to April – to ensure that what is targeted in the conclusions and interpretations is in fact associated with a polynya.
Please note also that the results of Kern (2008) are based on coarse resolution (25 km and 12.5 km, Backus-Gilbert interpolated to 12.5 km and 5 km) SSM/I data with the respective limitations detailed already by the authors.
ii) The **same applies to the IP**. If one would choose 2016 (we are at the beginning of November right now) more or less the entire Kara Sea is still ice free. The same would apply (currently) to regions WNZ, STO, BSH, CHU, half of ESS, and NOW. **The entire ice production happening in the Kara Sea would be included into an estimate of the ice production which according to your paper is associated with ice production in a polynya. What one would in fact include is open ocean ice production**. Yes, I agree that the advance of the ice-edge can be rather quick and the time between still open water and a closed sea ice cover of thickness > 0.2 m can be a few days only **and the associated IP will be like noise**. I tend to say though that **freeze-up of region KAR and WNZ takes longer**.
One would also assume that the entire ice production in these still large open water areas is due to thermodynamics. This is certainly a fair assumption for TIT < 0.2 m. However, one cannot consider effects such as rafting (due to wind but also due to tidal forcing) and one cannot consider the fact

that a substantial part of the sea ice might be of type pancake ice which has a substantial dynamic formation component.

I guess the same **applies to region WNZ where particularly the southern part might not be ice covered at all during November through March** (at least when I browse through sea-ice concentration maps of the period 2002/03 through 2014/15 I'd say that in half of the winters region WNZ was ice free most of the time). For the region SVA only the south-eastern part is influenced by MIZ activities. The Whaler's Bay polynya north of Svalbard, which is known as a sensible heat polynya is included into region SVA, however.

I suggest to, in the context of Figure 4 and its interpretation, **A) describe how the TIT frequency is computed** (does this include open water cases?) and **B) stress that areas with low TIT frequency can be both sea ice or open water**.

I am asking for A) because I am quite surprised to see the fracture event occurred in Feb 2013 in the Beaufort Sea (see Beitsch et al., 2014) to pop up in the TIT frequency map of FEB while the fact that large parts of the Kara Sea where essentially ice free in February 2013 seems not to have added to the TIT frequency. Therefore I am curious to learn from which measure the TIT frequency is computed (all native clear-sky IST cases – or all cases where a TIT value is present no matter whether it is native or interpolated) and what is the basis to make it a relative measure.

Thanks for this extensive remark which will hopefully help to clarify some potential interpretation-difficulties.

Regarding the computation of TIT frequencies (Remark A+B) in Fig.4, the presented numbers are the relative appearance rate, **i.e. the fraction of TIT≤ 0.2m detections per pixel (based on daily composites; including interpolated TIT), relative to the total amount of days between 2002/2003 and 2014/2015** (example for one polynya-pixel in January: e.g. 150 thin-ice detections in total in a maximum period of 13winters*31days (=403) days → TIT frequency = 0.37).

The thickness range of TIT≤ 0.2m theoretically includes all TIT between 0 and 20 cm, but due to the circumstance that a TIT of exactly 0cm is only possible in theory using the calculation with Q_ice (the IST would have to be exactly equal to Tf = 271.35K), open water in the sense of TIT=0cm is if at all only rarely included in our results. However, very thin ice (0cm < TIT < ~1-2cm) is frequently calculated.

Concerning your specific example for the Kara Sea in February (did you mean 2012?), (prolonged) ice-free conditions imply that there is potentially no IST available (temperature above freezing) at those locations and consequently – no TIT. But, as you correctly remark, thin ice can appear under those circumstances as kind of a noisy pattern for pixels where IST and hence TIT are available.

In case of the lead-event in the Beaufort Sea, we assume that thin ice in this broad, enduring and extensive area was correctly detected by our algorithm and is therefore also visible in Fig.4.

We added the following sentence when introducing Fig.4:
"Monthly thin-ice frequencies, calculated per pixel as the fraction of days with a TIT $\leq$ 0.2\unit{m} relative to the 13-yr investigation period, are presented in Fig.~4."

In addition, the caption of Fig.4 is slightly modified to read:
"Average wintertime (November to March) frequencies of TIT ≤ 0.2 m in the Arctic between winters 2002/2003 and 2014/2015. **For each month, frequencies are calculated per pixel as the fraction of days with a TIT $\leq$ 0.2\unit{m} relative to the 13-yr investigation period. (…)"**

P11, Lines 21/22: Please stress that the average TIT is computed only from pixels with TIT < 0.2 m and not from the fraction (COV4) of the open ocean area. This is important to avoid misinterpretation of your results. Perhaps you could put the TIT values into a separate table together with the POLA?

It seems likely to us that you might still be a bit confused by the "COV4" numbers. As we wrote in our first response letter, this measure refers to the **spatial coverage of MODIS data**, i.e. the availability of valid (clear-sky, HQ MCP, SFR) IST/TIT value-pairs inside a respective polynya mask area. Hence, it is not a fraction of open ocean area.

You are right however, that these average TIT values are calculated only for TIT ≤ 0.2m. We indicated that both in the caption of Tab.1 and in the MS (Sect. 4.1: "*Interannual average values for TIT ≤ 0.2 m are listed in Tab. 1 for each polynya region.*") and think that this should be sufficient, considering that this thickness-range is the basis of all results presented throughout the MS. Regarding the remark on a separate additional table, we would also sincerely dismiss this suggestion, as this basic information fits well into the context of more or less general information on each region.

Page 13, Line 8: Please check usage of "compliment". Perhaps "complement" would fit better?
It sure fits better, compliment to you for this suggestion/remark.

Page 13, Line 10: "this" → "these"
Fixed, thank you.

Page 16, Line 11 to Page 17, Line 9: I am glad the authors do mention here the fact that freeze-up has an impact on both POLA and IP particularly during Nov./Dec. To my opinion it is not only the freeze-up which has an impact but also the fact that some of the regions simply remain ice free has an impact. I'd tend to say that the decreasing trend of IP in region WNZ, for example, is simply a cause of a general decrease of the sea-ice cover in that region. In Line 6 on page 17, I suggest to refer to region SVA instead of region Storfjorden polynya.
Regarding the open-water influence, we supplemented the respective part accordingly. However, the Storfjorden / Svalbard area is now removed in that context, as (a) the seasonal contrast is more pronounced in the other named regions and (b) it shortens this exemplary list, thereby enhancing the reading flow in our opinion.
It now reads: "In case of e.g. the Kara Sea, Franz-Josef-Land, the Chukchi Sea and the Canadian Arctic Archipelago**, large thin-ice and potential open-water areas** during the early freezing period in November and December (…)"

Page 17, Line 7: The fact that region NOW stands out that much is again caused by a substantial fraction of IP during freeze-up. This is indicated partly by the high IP values in the fjords which become covered with fast ice in December/January (see Preusser et al., 2015, Remote Sensing, 7).
This effect certainly contributes to the observed distribution of high IP areas.

Page 19, line 11/12: Instead of "the net effect … here" the authors could write: "would require a theoretical study where an artificial polynya is investigated using several different spatial resolutions." As this would point towards a future direction of research.
Despite this good suggestion, we decided to keep the current formulation at this point of the MS.

Page 20, Lines 3-4 versus Line 7: I find that the notions of "suggests a southward shift of the fast-ice edge" and "shape and location of the fast-ice edge did not vary significantly" contradict each other and could lead to a misunderstanding. Perhaps the authors could reformulate this part. A suggestion would be: "… and (2) we observe opposing negative / positive IP-trends along the coasts of the Laptev and Kara Seas which could be due to changes in fast-ice extent. Decreasing …"
We modified the respective part as suggested to read:
"(…) and (2) we observe opposing negative / positive IP-trends along the coasts of the Laptev and Kara Seas which could be due to changes in fast-ice extent over the 13-year period."

Page 20, Lines 11-17: This is a quite global statement which could be strengthened. I suggest *to i) refer to when in particular 2m-air temperatures have been increasing (which season?), to ii) stress*

*that delayed freeze-up and longer open water seasons may easily lead to increased POLA and IP which could counterbalance smaller IP caused by warmer air temperatures*, and to iii) *be more specific how and why a downward trend in sea-ice volume can have an influence on POLA and IP.*
A downward trend in sea-ice volume (and hence ice thickness) would theoretically result in a more fragile and more mobile sea ice cover in the Arctic with an increased sensitivity for external forcing mechanisms responsible for thin ice / polynya-openings (e.g. wind, ocean currents). However, this part of the manuscript was meant to highlight the processes and factors that contribute or interact to/with a shortening of the freezing season at high latitudes, thereby indirectly influencing POLA and IP though longer open-ocean periods. One of the recent studies on year-round increasing temperatures (~Arctic Amplification) in most regions of the Arctic was recently published e.g. by Cohen et al. (2014)[1].

We modified the part at the end of Sect.4.1 to now read:

"(…) could be connected to an overall later appearing fall freeze-up (Markus et al., 2009; Stroeve et al., 2014) in recent years, which itself is thought to result from a complex mixture/interplay of steadily and **year-round** increasing (2m-) air temperatures (**e.g. Cohen et al., 2014**), distinct large-scale atmospheric patterns (e.g. Rigor et al., 2002) and the overall downward trend of total sea-ice extent and volume in the Arctic (e.g. **Schweiger et al., 2011; Laxon et al., 2013**). **The latter implies a tendency towards a more fragile, thus mobile, sea-ice cover in the Arctic, with a potentially increased sensitivity for external forcing mechanisms (i.e. strong winds and/or ocean currents) that are responsible for thin-ice formation in polynyas and leads.**"

In addition, due to similar explanations, we had to slightly reformulate a short part in Sect.4.2, which now reads:

"For the period from 1982 to 2009, the study by Kwok et al. (2013) presented indicators for a net-strengthening of both the Transpolar Drift and the Beaufort Gyre as well as a general increase of the Arctic ice drift-speed, **which is presumably related to a decreasing fraction of thick multi-year (MY) ice. As mentioned before (Sect. 4.1), the latter is thought to be connected to an increased fragility and mobility of the Arctic sea-ice cover, which may have implications for polynya and lead dynamics not only in the eastern Arctic.**"

Page 22, Line 11: "it has to noted" → "it has to be noted"

Fixed, thank you.

Page 24, Figure 12: The annotation in the upper left corner of the image and the y-axis annotation says: "Ice Export Area" while the caption speaks of "Ice Area Export". Which is correct?
"Ice Area Export". We fixed **Fig.12** accordingly.

Page 26, bullet (1): You could add that with this spatial resolution you can go much closer to the coast and land spill over effects are efficiently mitigated.
Bullet (2): Given the concerns formulated above I suggest the authors stress that the IP mentioned here is not necessarily from ice production in a polynya but includes open ocean ice production. Further in this bullet you could formulate more concisely instead of "Compared to …": "Our estimate of 
[revised manuscript text omitted]